# Accelerated DNA replication fork speed due to loss of R-loops in myelodysplastic syndromes with *SF3B1* mutation

David Rombaut [1,2,3,4,18], Carine Lefèvre [1,2,3,18], Tony Rached [1,2], Sabrina Bondu[1,2], Anne Letessier [1], Raphael M. Mangione [5], Batoul Farhat [1,2], Auriane Lesieur-Pasquier[1,2], Daisy Castillo-Guzman[6], Ismael Boussaid [1,2,4], Chloé Friedrich [1,2,4], Aurore Tourville [1,2], Magali De Carvalho[1,2], Françoise Levavasseur [1,2], Marjorie Leduc [1,7], Morgane Le Gall [1,7], Sarah Battault[1,2], Marie Temple [4], Alexandre Houy [8], Didier Bouscary[1,9], Lise Willems[1,9], Sophie Park [10], Sophie Raynaud[11], Thomas Cluzeau [12], Emmanuelle Clappier[13], Pierre Fenaux[14], Lionel Adès [14], Raphael Margueron [15], Michel Wassef[15], Samar Alsafadi [8], Nicolas Chapuis [1,4], Olivier Kosmider [1,2,4], Eric Solary [16], Angelos Constantinou [17,19], Marc-Henri Stern [8], Nathalie Droin [16], Benoit Palancade [5], Benoit Miotto [1], Frédéric Chédin [6] & Michaela Fontenay [1,2,3,4] ✉

Myelodysplastic syndromes (MDS) with mutated *SF3B1* gene present features including a favourable outcome distinct from MDS with mutations in other splicing factor genes *SRSF2* or *U2AF1*. Molecular bases of these divergences are poorly understood. Here we find that *SF3B1*-mutated MDS show reduced R-loop formation predominating in gene bodies associated with intron retention reduction, not found in *U2AF1*- or *SRSF2*-mutated MDS. Compared to erythroblasts from *SRSF2*- or *U2AF1*-mutated patients, *SF3B1*-mutated erythroblasts exhibit augmented DNA synthesis, accelerated replication forks, and single-stranded DNA exposure upon differentiation. Importantly, histone deacetylase inhibition using vorinostat restores R-loop formation, slows down DNA replication forks and improves *SF3B1*-mutated erythroblast differentiation. In conclusion, loss of R-loops with associated DNA replication stress represents a hallmark of *SF3B1*-mutated MDS ineffective erythropoiesis, which could be used as a therapeutic target.

Myelodysplastic syndromes (MDS) are hematopoietic stem cell neoplasms with heterogeneous outcome and limited therapeutic options in which somatic mutation in splicing factor (SF) genes is a cardinal feature[1]. MDS with splicing factor3b subunit 1 mutation (*SF3B1*[MUT]) are commonly associated with bone marrow (BM) ring sideroblasts and ineffective erythropoiesis[2,3]. The response rate to erythropoiesis-stimulating agents (ESA) is achieved in as many as 50-60%, of low-

risk MDS, but patients with *SF3B1*[MUT] MDS could be more often primary resistant with a trend to shorter median duration of response[4,5]. The transforming-growth factor beta family ligand trap, luspatercept has been initially approved to treat the anemia of transfusion-dependent patients with MDS with ring sideroblasts after ESA failure. Forty-five percent achieve transfusion independency with a 30-week median duration of response and patients could continue to benefit from long-

term treatment[6–8]. Deciphering the mechanisms of anemia is needed to generate treatments.

*SF3B1* mutation causes multiple alterations in mRNA processing. The use of alternative 3′ or 5′ splice site produces transcripts containing short intronic sequences that are degraded by the non-sense mediated decay (NMD) or are translated into a variant protein[9–11]. These splicing changes drive transcriptional reprogramming that shapes disease phenotype. For example, down-regulation of Fe-S cluster transporter *ABCB7* by transcript isoform-specific degradation, and reduced translation efficiency of mitochondrial iron transporter *TMEM14C*, contribute to mitochondrial iron accumulation[12,13]. Overproduction of alternative and canonical transcripts of *ERFE* gene encoding hepcidin transcriptional repressor erythroferrone, leads to systemic iron overload[11]. *SF3B1* mutation also targets mitochondrial respiration and serine synthesis pathway[14]. While mutations in serine and arginine-rich splicing factor 2 (*SRSF2*) and U2 small nuclear RNA auxiliary factor 1 (*U2AF1*), predominantly alter cassette exon[15,16], *SF3B1*^MUT splicing pattern is dominated by intron retention reduction (IRR)[17].

An increasing number of genomic alterations associated with MDS progression to acute myeloid leukemia (AML) suggests a genomic instability of stem and progenitor cells[18]. DNA damage and activation of ataxia telangiectasia and Rad3-related protein (ATR) pathway were detected in *SF*-mutated MDS[19,20]. More specifically, *SRSF2* and *U2AF1* mutations induce the formation of unscheduled RNA:DNA hybrids or R-loops, triggering DNA replication stress[19–21]. Mechanistically, mutant *SRSF2* impairs the RNA polymerase II transcription pause release, allowing nascent RNA forming a R-loop at promoter[21]. SF3B1 has been involved in the pathways of DNA repair[22,23]. However, *SF3B1*^MUT MDS patients have a lower risk of AML than other MDS[24,25] suggesting that *SF3B1*^MUT MDS are less prone to genomic instability.

In the present study, we report that, on the contrary to *SRSF2*^MUT or *U2AF1*^MUT cells, *SF3B1*^MUT erythroblasts demonstrate a significant loss of R-loops. These cells endure a DNA replication stress consisting in accelerated fork progression and single-stranded (ss)DNA exposure, and correlating with increased erythroid cell proliferation and impaired differentiation. The ability of low doses of histone deacetylase inhibitor (HDACi) vorinostat to restore R-loops without DNA damage, and to improve erythroid differentiation could serve as a therapeutic approach.

## Results

### Reduction of intron retention correlates with transcriptomic changes of *SF3B1*-mutated bone marrow mononuclear cells

A total of 143 subjects were enrolled in this study including 70 MDS with *SF3B1* mutation, 49 *SF3B1*^WT MDS and 24 healthy controls (Table 1). To investigate the molecular pathways whose deregulation drives the phenotype of *SF3B1*^MUT-MDS, BM mononuclear cells (MNC) RNA-sequencing data available from 21 *SF3B1*^MUT-MDS and 6 *SF3B1*^WT-MDS were re-analyzed[11]. With a mean number of 87 to 97 million reads per sample, DESeq2 analysis identified 1764 differentially expressed genes (DEGs) including 812 up- and 952 down-regulated genes (log$_2$ fold-change (FC) > |1|, Benjamini-Hochberg (BH)-adjusted *P* value < 0.05) (Fig. 1a; Supplementary Data 1). Gene ontology (GO) enrichment analysis showed that up- and down-regulated genes were involved in several pathways such as DNA replication, DNA repair, chromatid segregation, and cell cycle checkpoint signaling (Fig. 1b). These pathways were over-represented among up-regulated genes (Supplementary Fig. 1a). Eighty genes associated with these pathways allowed the clustering of *SF3B1*^MUT-samples (Fig. 1c; Supplementary Data 1). Using KisSplice with a variation of percent splice in (ΔPSI) > |0.10| and a BH-adjusted *P* value < 0.05, we detected 3937 differential splicing events (DSEs) in *SF3B1*^MUT samples, including 1256 abnormal intron retention events, consisting in a majority of IRRs (n = 1027) in *SF3B1*^MUT-samples (Fig. 1d). IRRs were the most frequent event in *SF3B1*^MUT-MDS,

**Table 1 | Clinical and biological characteristics of patients according to *SF3B1* gene status**

| Parameters | *SF3B1*^WT n = 49 | *SF3B1*^MUT n = 70 | P value |
|---|---|---|---|
| **Age** | | | |
| Median. years (range) | 73 (44–94) | 76 (46–91) | 0.210 |
| **Sex** | | | |
| Male no. (%) | 29 (59.2) | 38 (54.3) | 0.708 |
| **WHO-Subtype 2016 no. (%)** | | | |
| 5q- syndrome | 1 (2.0) | 1 (0.1) | <0.0001 |
| MDS-SLD | 8 (16.3) | 0 (0.0) | |
| MDS-MLD | 23 (46.9) | 0 (0.0) | |
| MDS-RS-SLD | 0 (0.0) | 25 (35.7) | |
| MDS-RS-MLD | 1 (2.0) | 40 (57.1) | |
| MDS-EB1 | 16 (32.6) | 4 (5.7) | |
| **Hemogram, median (IQR 25-75)** | | | |
| Hemoglobin (g/dL) | 9.7 (8.7–10.6) | 8.8 (8.1–9.6) | 0.001 |
| Neutrophil count (G/L) | 2.3 (1.4–3.9) | 2.8 (1.8–4.0) | 0.617 |
| Platelet count (G/L) | 136 (82–243) | 258 (205–575) | <0.0001 |
| **Bone marrow, median (IQR 25-75)** | | | |
| Blasts (%) | 4 (2–6) | 2 (1–3) | <0.0001 |
| Erythroblasts (%) | 24 (18–33) | 40 (29–49) | <0.0001 |
| Ring sideroblasts (%) | 0 (0–4) | 41 (25–54) | <0.0001 |
| **Karyotype IPSS-R no. (%)** | | | |
| Very good | 1 (2.0) | 6 (8.6) | 0.033 |
| Good | 29 (59.2) | 51 (72.9) | |
| Intermediate | 8 (16.3) | 9 (12.9) | |
| Poor | 1 (2.0) | 0 (0.0) | |
| NA | 10 (20.4) | 4 (5.7) | |
| **IPSS-R no. (%)** | | | |
| Very low | 7 (14.3) | 12 (15.9) | 0.0002 |
| Low | 13 (26.5) | 44 (62.9) | |
| Intermediate | 11 (22.4) | 8 (11.4) | |
| High | 6 (12.2) | 2 (2.9) | |
| Very high | 2 (4.1) | 0 (0.0) | |
| NA | 10 (20.4) | 4 (5.7) | |

Unpaired t-test to compare quantitative values and Fisher 's exact test to compare categorical variables.
*SLD* single lineage dysplasia, *MLD* multilineage dysplasia, *RS* ring sideroblasts, *EB1* excess of blasts type 1 (5 to 9% bone marrow blasts), *IPSS-R* International Scoring System-Revised, *no* number, *IQR* interquartile range, *NA* not available.

uncommon in the *SRSF2*^MUT or *U2AF1*^MUT-MDS of a large cohort of 189 lower-risk patients (Supplementary Fig. 1b). A GO analysis of the 822 genes affected by IRR revealed their over-representation in DNA replication, DNA repair, cell cycle regulation, and mRNA splicing (Supplementary Fig. 1c). Combined DEG and DSE analyses showed that 296 DEGs targeted by 384 IRR events referred to DNA repair, DNA replication, cell cycle process and regulation of chromosome separation (Supplementary Data 1; Fig. 1e). Thus, IRR might contribute to gene expression changes of *SF3B1*^MUT-BM MNC.

### *SF3B1*-mutated erythroid precursors demonstrate characteristic transcriptomic signatures

To decipher how transcriptional reprogramming driven by *SF3B1* mutation affected erythroid lineage, we expanded erythroblasts from CD34$^+$ hematopoietic stem and progenitor cells (HSPCs) collected from 14 *SF3B1*^MUT, 10 without *SF* (*SF3B1*, *SRSF2*, *U2AF1*) mutation (*SF*^WT) MDS and 6 healthy controls (Fig. 2a). May-Grünwald-Giemsa (MGG)-stained cytospins showed the predominance of basophilic erythroblasts (basoE) at days 11-13 and polychromatophilic erythroblasts

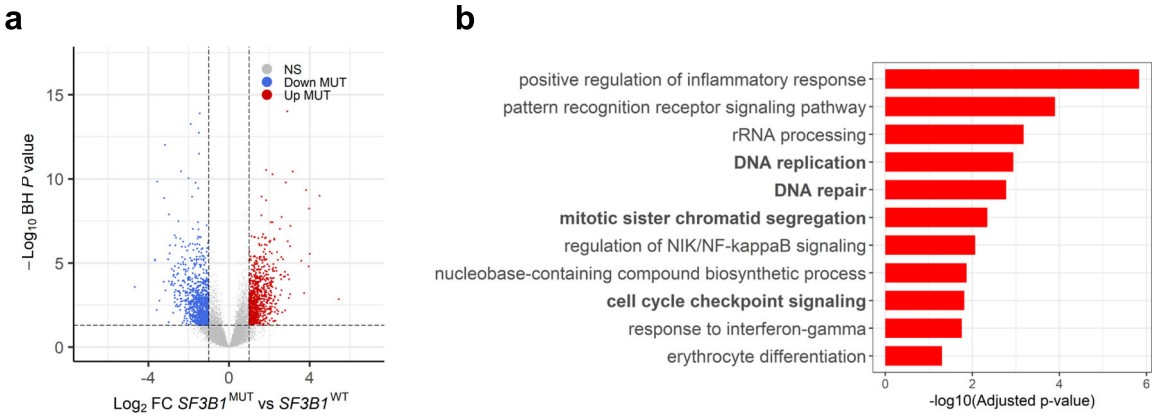

(polyE) at days 14-15. Compared to healthy controls, the proportion of immature erythroblasts was significantly higher in MDS samples (Fig. 2b). Despite a significant increase of apoptotic cells, the rate of proliferation of $SF3B1^{MUT}$ erythroblasts was similar to control and $SF3B1^{WT}$ erythroblasts (Supplementary Fig. 2). In $SF3B1^{MUT}$-cultures, the range of $SF3B1$ variant allele frequency was 23–54% at d11-13 and 17–51% at d14-15 (Fig. 2c).

RNA-sequencing provided an equivalent mean number of million reads per sample in basoE ($116 \pm 18$) and polyE ($97 \pm 19$) (t-test; $P = 0.543$). With a $\log_2(FC) > |1|$ (BH-adj $P$ value $< 0.05$), a total of 675 DEGs (514 up and 161 down) separated mutant and wild-type basoE. The number of DEGs was 765 (390 up, 375 down) in polyE with an overlap of 179 genes with basoE (Fig. 2d, Fig. 2e; Supplementary Data 2). GO enrichment analysis showed that up- and down-regulated

**Fig. 1 | Intron retention reduction correlates with transcriptomic changes of human *SF3B1*^MUT bone marrow mononuclear cells.** RNA-sequencing data of 21 *SF3B1*^MUT and 6 *SF3B1*^WT (4 *SRSF2*^MUT and 2 *SF*^WT) bone marrow mononuclear cell samples were re-analyzed. **a** Volcano plot showing 1764 up or down-regulated genes in *SF3B1*^MUT samples (Log₂(FC) <|1|; Two-sided Wald test and Benjamini-Hochberg (BH)-adjusted *P* value < 0.05). **b** Gene Ontology (GO) enrichment analysis of the up and downregulated genes showing significantly enriched terms according to -log10(adjusted *P* value). Fisher's exact test corrected by false discovery rate (FDR) < 0.05. Terms of interest are in bold. **c** Heatmap representing the clustering of samples by the variations of expression of a subset of 80 genes belonging to GO

terms highlighted in (**b**). Genes affected by 1 to 8 differential splicing events are marked with an asterisk. **d** Barplot representing the number and types of differential splicing events in *SF3B1*^MUT in comparison to *SF3B1*^WT with ΔPSI >|0.10| using two-sided Wald test and BH-adjusted *P* value < 0.05. The bars over 0 indicate the events upregulated in mutant cases and the bars under 0 indicate the events downregulated in wild type cases. **e** GO over-representation analysis of 296 significantly deregulated genes affected by 383 intron retention reductions. Fisher's exact test corrected by FDR < 0.05. FC: fold-change. Source data are provided as a Source Data file.

genes in *SF3B1*^MUT-basoE or -polyE were involved in several pathways such as DNA repair, regulation of MAP kinase cascade, ubiquitin-dependent protein catabolism, cellular response to DNA damage and oxygen-containing compound (Fig. 2f; Supplementary Fig. 3a). In basoE, a GeneSet Enrichment Analysis (GSEA) refined these results showing the deregulation of specific DNA repair pathways such as base excision repair, and a trend for nucleotide excision repair, but neither homologous recombination nor non-homologous end-joining or mismatch repair. Genes involved in DNA replication and G1/S phase checkpoint were also significantly deregulated (Fig. 2g; Supplementary Fig. 3b, c). The DSE profiles identified in basoE and polyE were similar to that of BM MNC (Fig. 2h). IRR represented 194/829 (23.4%) DSEs in *SF3B1*^MUT-basoE and 383/1182 (32.4%) DSEs in *SF3B1*^MUT-polyE, respectively, showing that IRR frequency increased with cell differentiation (Fig. 2i; Supplementary Data 2). Genes affected by IRR regardless of the stage of erythroid differentiation were involved in the response to DNA damage, mitotic cell cycle regulation, nucleocytoplasmic transport and in the positive regulation of histone deacetylation (Fig. 2j). As the nuclear retention or cytoplasmic degradation of IR-containing transcripts participate in the differentiation and specialization of normal erythroid cells[26–28], we reasoned that IRR-transcripts detected in *SF3B1*^MUT-erythroblasts may reshape the transcriptome contributing to defective maturation.

## *SF3B1*-mutated erythroid precursors demonstrate characteristic proteomic signatures

We concomitantly performed a proteomic analysis of proE, basoE and polyE with label-free quantification (LFQ). Principal component analysis showed that proE and basoE clustered together and separately from polyE. In each group, the second dimension discriminated *SF3B1*^MUT from *SF3B1*^WT-samples (Supplementary Fig. 4a). In subsequent analyses, proE and basoE were grouped. The mean number of identified proteins with LFQ values per sample was 4231 (± 263), without significant difference between *SF3B1*^MUT and *SF3B1*^WT-samples, or between proE/basoE and polyE samples of each genetic group. A total of 443 and 290 differentially expressed proteins were detected between *SF3B1*^MUT and *SF3B1*^WT-samples in proE/basoE and in polyE, respectively (t-test, Log₂(LFQ intensity)>|0.20|, *P* value < 0.05) (Fig. 2k; Supplementary Fig. 4b, c).

In *SF3B1*^MUT-proE/basoE proteome, mitochondrial ABCB7 (ATP-binding cassette B member 7), SUCLA2 (succinyl coA ligase), NNT (NAD(P) transhydrogenase), PPOX (protoporphyrinogen oxidase) were decreased, while SOD2 (superoxide dismutase) and ATP5A1 (ATP synthetase subunit alpha) were increased, further indicating mitochondrial dysfunction. Several key components of DNA replication pathway, such as DNA ligases LIG1 and LIG3 were decreased (Supplementary Fig. 4d; Supplementary Data 3). In *SF3B1*^MUT-polyE, FEN1 (5'FLAP-endonuclease and gap endonuclease) and POLE (polymerase ε) expression was decreased while XRCC3 (X-ray repair cross-complementing 3), and ORC1 (Origin recognition complex subunit 1) expression was increased (Supplementary Data 3). Ingenuity Pathway Analysis showed that changes of *SF3B1*^MUT-proE/basoE and polyE proteomes overlapped with changes of their transcriptomes (Fig. 2f, g) such as base excision repair, nucleotide excision repair,

oxidative stress response, protein ubiquitination pathways. MAP kinase pathway, pyrimidine de novo biosynthesis and iron metabolism were specifically deregulated in proE/basoE, while heme biosynthesis, cell cycle checkpoint control and cell cycle control of replication were deregulated in polyE (Fig. 2i). Together, proteomic changes, notably those affecting components of DNA damage response kept the imprint of IRR-transcripts associated with *SF3B1* mutation in erythroid precursors (Fig. 2j).

## *SF3B1* mutation induces a significant loss of R-loops in erythroid cells

RNA splicing may attenuate the probability of forming R-loops by reducing the homology between nascent RNAs and their DNA templates and/or by recruiting splicing factor that antagonize RNA:DNA hybrid formation[29–33]. We examined the profiles of R-loops genome-wide in human primary basoE by performing DNA-RNA immunoprecipitation (DRIP) using S9.6 antibody followed by sequencing in 5 *SF3B1*^MUT, 6 *SF3B1*^WT including 3 with low variant allele frequency, 3 with high risk-mutations (1 *SRSF2* mutation, 1 *SRSF2* and bi-allelic *TET2* (bi*TET2*) co-mutations, 1 *NRAS* and bi*TET2* co-mutations) and 4 controls. DRIP specificity was assessed by a pre-treatment with RNase H1 (RNH1). Stringent calling identified true-positive peaks (BH-adj *P* value < 0.05) and we considered shared peaks between samples of each group. The number of shared peaks in *SF3B1*^MUT erythroblasts was significantly lower than *SF3B1*^WT or control cells (Fig. 3a). Visualization of R-loop profiles of a 50 kb region on chromosome 7 demonstrated the overall reduction of the peaks in *SF3B1*^MUT sample compared to *SF3B1*^WT and controls (Fig. 3b). We then compared the localization of the peaks to gene features between *SF3B1*^MUT and *SF3B1*^WT cells. In *SF3B1*^MUT, the proportion of R-loops decreased at 5'UTR, promoter-transcription start site (TSS) and gene body (Fig. 3c). R-loops remaining in *SF3B1*^MUT cells, were more frequently detected in intergenic regions and at 3'UTR of the genes (Fig. 3d). As an example, the *SUZ2* gene exhibited a reduced R-loop near its promoter in a *SF3B1*^MUT-sample (Fig. 3e). By contrast, at mitochondrial DNA loci, rDNA repeats, or at some loci like *CPNE7* gene, peak intensity was found elevated in *SF3B1*^MUT-samples showing that their R-loop profiles were selectively changed (Supplementary Fig. 5a).

R-loops assemble dynamically at TSS and TTS where they contribute to the regulation of transcription[34–36]. Thus, we compared the mRNA level of genes overlapping shared R-loops at TSS, gene body, TTS and 3'UTR in *SF3B1*^WT and *SF3B1*^MUT-samples. The expression of genes in which R-loops were detected at TTS or 3'UTR in *SF3B1*^WT samples was similar in *SF3B1*^WT and *SF3B1*^MUT-cells while the expression of genes in which R-loops were detected at gene body in *SF3B1*^WT-samples increased in *SF3B1*^MUT-samples. By contrast, the expression of genes in which R-loops were detected at TSS in *SF3B1*^WT but not in *SF3B1*^MUT samples dramatically decreased in *SF3B1*^MUT-samples (Fig. 3f).

To look for differential peaks between the groups, the count of restriction fragments overlapping with significant peaks was normalized using DESeq2. We confirmed a global decrease in the number of R-loops in *SF3B1*^MUT versus controls or *SF3B1*^WT basoE. In these analyses, the number of differential peaks (with log₂(FC) >|1| and BH-adj *P* value < 0.05), was 4589 between *SF3B1*^MUT and controls with only 52 up

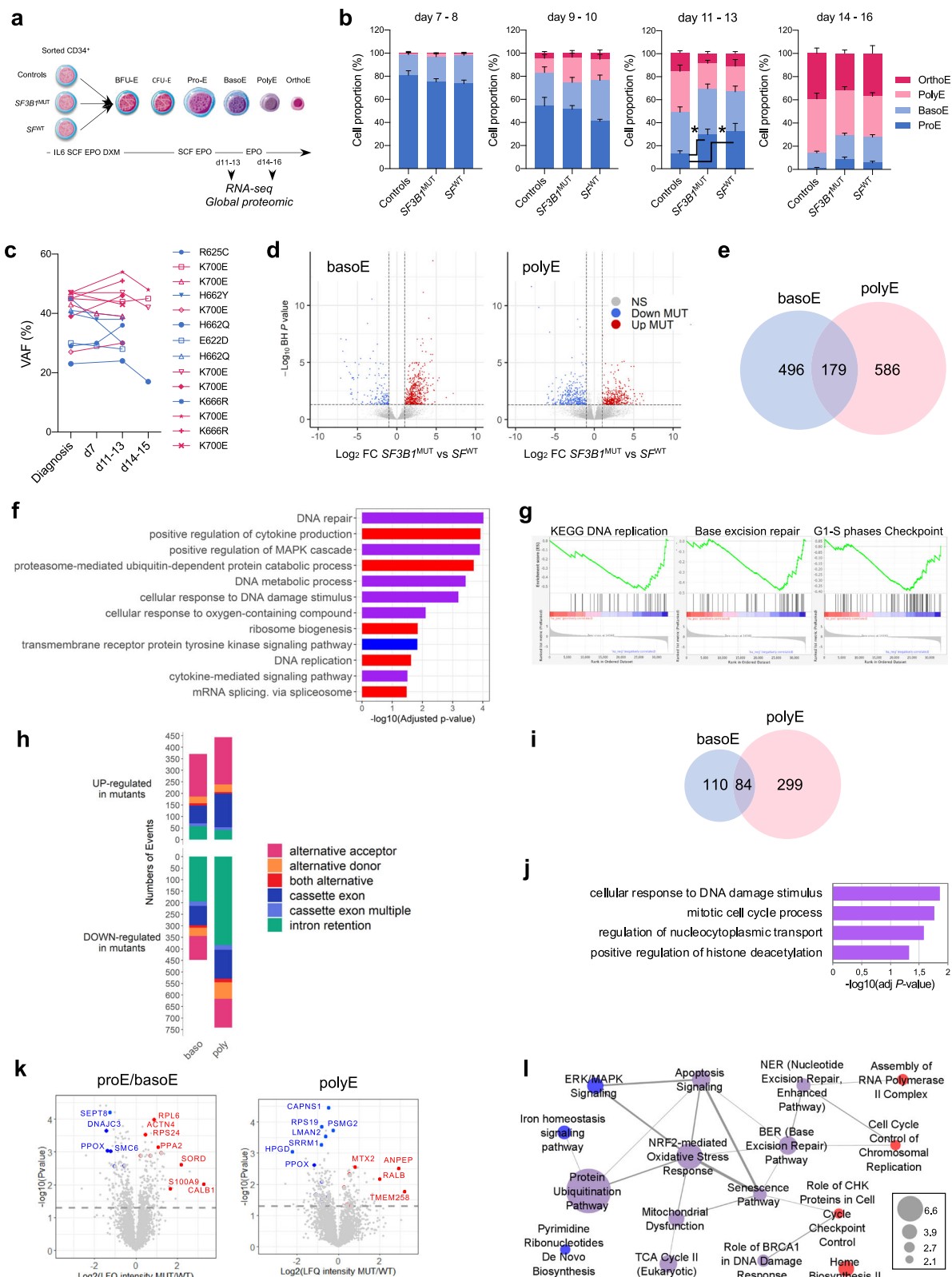

in $SF3B1^{MUT}$ samples, and 19,394 between $SF3B1^{MUT}$ and $SF3B1^{WT}$ samples, of which only 33 were up-regulated in $SF3B1^{MUT}$ cells (Fig. 3g; Supplementary Data 4). In comparison to controls, 53% and 20% of the peaks lost in $SF3B1^{MUT}$ cells were located in gene bodies (mostly centered on intronic sequences), and promoter-TSS, respectively (Supplementary Fig. 5b). Similarly, in comparison to $SF3B1^{WT}$ cells, 63% and 12% of the peaks lost in $SF3B1^{MUT}$ cells were located in gene bodies and

promoter-TSS (Fig. 3h). In this latter comparison, we identified 7,039 unique genes affected by losses of R-loops either in their UTRs, TSS, TTS or gene bodies, which overlapped 345/498 (69%) of the recurrent IRRs shared by $SF3B1^{MUT}$ samples (Fig. 3i). For example, at $RAD9A$ and $IQGAP3$ loci (Fig. 3j) or at $COASY$ locus (Supplementary Fig. 5c), all occupied by an IRR, R-loops were significantly reduced in $SF3B1^{MUT}$ cells. To validate these findings, we selected two genes ($ABCC5$ and

**Fig. 2 | Transcriptomic and proteomic features of *SF3B1*$^{MUT}$ bone marrow-derived basophilic (basoE) and polychromatophilic (polyE) erythroblasts.** Erythroid precursors were expanded in culture from *SF3B1*$^{MUT}$, *SF*$^{WT}$ MDS and controls samples. **a** Schematic representation of the protocol. **b** Erythroid differentiation evaluated on May-Grünwald Giemsa-stained cytospins. Histograms representing the proportion of erythroid precursors in up to 7 controls, 11 *SF3B1*$^{MUT}$ and 7 *SF*$^{WT}$ independent samples at days (d)7-8, 9-10, 11-13 and 14-16. Results are expressed as means ± standard error of the mean. 2-way ANOVA for multiple comparisons. Controls versus *SF3B1*$^{MUT}$, *P* = 0.017; controls versus *SF*$^{WT}$, *P* = 0.012. **c** Variant allele frequencies of *SF3B1* mutation in erythroblasts at d7, d11-13 and d14-15 of 14 independent *SF3B1*$^{MUT}$ samples. **d** Volcano plot representing up- and down-regulated transcripts in *SF3B1*$^{MUT}$ basoE and polyE compared to *SF*$^{WT}$ ones. Two-sided Wald-test and BH-adjusted *P* value < 0.05. **e** Venn diagram representing the numbers of differentially expressed genes between *SF3B1*$^{MUT}$ and *SF*$^{WT}$ samples at basoE and polyE stages. **f** Gene Ontology (GO) enrichment analysis of up- and down-regulated genes in *SF3B1*$^{MUT}$ versus *SF*$^{WT}$ erythroblasts. Fisher's exact test corrected by false discovery rate (FDR) < 0.05. Specific terms to basoE or polyE as blue or red bars, respectively, shared terms as violet bars. **g** Gene set Enrichment Analysis (GSEA) showing terms deregulated in *SF3B1*$^{MUT}$ basoE. **h** Barplots representing numbers and types of differential splicing events in *SF3B1*$^{MUT}$ versus *SF3B1*$^{WT}$ basoE and polyE with ΔPSI > |0.10| using two-sided Wald-test and BH-adjusted *P* value < 0.05. Bars over 0 indicate events upregulated and bars under 0 indicate events downregulated in *SF*$^{WT}$ erythroblasts. **i** Venn diagram of intron retention reductions (IRR) in *SF3B1*$^{MUT}$ basoE and polyE. **j.** GO terms overrepresented among genes with IRR in *SF3B1*$^{MUT}$ basoE and polyE. Fisher's exact-test corrected by FDR < 0.05. **k** Volcano plots representing differentially expressed proteins in *SF3B1*$^{MUT}$ versus *SF*$^{WT}$ samples at proE/basoE and polyE stages (Wald-test, BH-adjusted *P* value < 0.05). **l** Cytoscape representation of Ingenuity *P*athway Analysis showing deregulated pathways in *SF3B1*$^{MUT}$ versus *SF3B1*$^{WT}$ samples (*P* values < 0.05 by Student t-test) either basoE-specific (blue dots), polyE-specific (red dots) or shared (violet dots). Scale: dot size proportional to −log10 (adjusted-*P* value). Source data are provided as a Source Data file.

*TCIRG1*) with IRR and two genes (*IREB2* and *TMX2*) without IRR in *SF3B1*$^{MUT}$-erythroblasts. We performed DRIP-qPCR in basoE generated from 4 *SF3B1*$^{MUT}$ MDS, 3 *SF3B1*$^{WT}$ MDS including one MDS with *U2AF1* mutation, and 4 healthy donors. Positive controls for R-loops detection were *RPL13A* and *TFPT* genes and for each sample, the specificity of the signal was assessed by RNH1 pre-treatment (Supplementary Fig. 6a, b). R-loops detected at *ABCC5* and *TCIRG1* loci in healthy donor and *SF3B1*$^{WT}$ cells were not or hardly detected in *SF3B1*$^{MUT}$-samples. By contrast, at *IREB2* and *TMX2* loci, the enrichment signal was faint, suggesting the absence of R-loop, whatever the sample (Fig. 3k).

Altogether, these results validate a link between *SF3B1* mutation, decreased R-loop formation, intron retention reduction and deregulated gene expression.

## SF3B1 mutation promotes a DNA replication stress in human erythroblasts

Since the accumulation of R-loops is reported to slow down the DNA replication fork velocity and cell proliferation[37], loss of R-loops in *SF3B1*$^{MUT}$ erythroblasts may reduce obstacles to fork progression, promote DNA replication and cell proliferation. To explore this, we expanded erythroblasts from CD34$^+$ HSPCs of 5 *SF3B1*$^{MUT}$, 3 *SRSF2*$^{MUT}$ or *U2AF1*$^{MUT}$, 3 *SF*$^{WT}$-MDS and 4 healthy donors. Flow cytometry analysis showed significant increase in the BrdU intensity of S-phase cells (Fig. 4a, b) and percentage of S-phase cells (Fig. 4c) in *SF3B1*$^{MUT}$-basoE.

To further monitor DNA replication, we performed DNA combing in primary erythroblasts. We labelled basoE from 4 *SF3B1*$^{MUT}$, 2 *SRSF2*$^{MUT}$ and 1 *U2AF1*$^{MUT}$ (*SF*$^{MUT}$), 2 *SF*$^{WT}$ MDS and 2 healthy control samples with 5-iodo-2′-deoxyuridine (IdU) and then with 5-chloro-2′-deoxyuridine (CldU) for 30 min each (Fig. 4d). Incorporation of thymidine analogs allowed measurement of DNA fibers length and symmetry. We observed a significant increase of replication fork speed in *SF3B1*$^{MUT}$-erythroblasts (0.75 ± 0.35 kb/min) compared to *SF*$^{MUT}$- (0.60 ± 0.25 kb/min), *SF*$^{WT}$- (0.53 ± 0.21 kb/min) and healthy donor (0.64 ± 0.25 kb/min) erythroblasts while fork symmetry of *SF3B1*$^{MUT}$-cells measured by IdU/CldU ratio remained similar to that in controls. Compared to *SF3B1*$^{MUT}$-cells, *SF*$^{MUT}$-erythroblasts exhibited a slower and asymmetric fork progression (Fig. 4e, f). Altogether, an accelerated replication fork speed defines the DNA replication stress of *SF3B1*$^{MUT}$-erythroblasts, whereas, it is associated with fork stalling in *SRSF2*$^{MUT}$ or *U2AF1*$^{MUT}$ cells[19,21].

To address whether *SF3B1*$^{MUT}$-erythroblasts endure a DNA damage, we investigated the expression by immunofluorescence of phospho(p)-RPA32 serine 33 that is recruited on single-stranded (ss) DNA during DNA replication and phosphorylated by ATR, and of p-RPA32 serine 4/8 that is phosphorylated by DNA-PK to regulate replication stress checkpoint activation[38,39]. We also used pan DNA-damage markers γH$_2$AX or 53BP1 in human basoE from *SF3B1*$^{MUT}$, *SF*$^{MUT}$ (*SRSF2*$^{MUT}$ or *U2AF1*$^{MUT}$), *SF*$^{WT}$ MDS and controls, at d11 of the culture (Fig. 4g-k) or at different timepoints d9, d11, d13 and d15 (Supplementary Fig. 7a-c). p-RPA s33, p-RPA s4/8, γH$_2$AX and 53BP1 foci were detected in control erythroblasts treated with hydroxyurea (HU), *SRSF2*$^{MUT}$ and *U2AF1*$^{MUT}$-erythroblasts. By contrast, γH$_2$AX and 53BP1 foci were undetectable in *SF3B1*$^{MUT}$ erythroblasts, while these cells were positive for p-RPA32s33 and p-RPAs4/s8 indicating ssDNA exposure. Of note, p-RPA32 s33 foci were significantly less abundant in *SF3B1*$^{MUT}$ than in *SRSF2*$^{MUT}$ cells.

Altogether, these results show that accelerated fork velocity was observed when R-loops were lost in *SF3B1*$^{MUT}$-erythroblasts. Exposure of ssDNA without evidence for DNA damage marked by γH$_2$AX/53BP1 indicates that *SF3B1*$^{MUT}$-erythroblasts endured a milder replication stress than *SRSF2*$^{MUT}$ and *U2AF1*$^{MUT}$-erythroblasts.

## Sf3b1$^{K700E/+}$ in murine erythroblasts reproduces DNA replication stress

To validate these findings in another model, we used the murine proerythroblastic cell lines G1E-ER4 CRISPR-Cas9 *Sf3b1*$^{K700E/+}$ and its isogenic G1E-ER4 *Sf3b1*$^{+/+}$, which could differentiate into basoE upon induction with estradiol of GATA1, with no excess of apoptosis (Supplementary Fig. 8a-f)[11]. RNA sequencing of the murine *Sf3b1*$^{K700E/+}$ and *Sf3b1*$^{+/+}$ erythroblasts identified 1226 (719 up and 507 down) and 1434 (574 up and 860 down) DEGs before (t0) and after (t24) induction of GATA1, with log2 (FC) > |1| and BH-adj *P* value < 0.05, respectively (Supplementary Data 5; Supplementary Fig. 9a, b). We detected 1116 and 1301 genes affected by DSEs, mainly IRR, in *Sf3b1*$^{K700E/+}$.proE and basoE, respectively (Supplementary Data 5; Fig. 5a). Despite the substantial species specificity of RNA splicing, the deregulated pathways associated with DSE in murine cells seemed similar to those identified in human cells (Fig. 5b, upper panel; Fig. 2f). Notably, DNA repair, cellular response to DNA damage and nucleic acid metabolic process GO terms gathered genes presenting IRR in *Sf3b1*$^{K700E/+}$-proE (n = 272) or -basoE (n = 355) (Fig. 5b, bottom panel). Such deregulated pathways were conserved at protein level with deregulated expression of Lig1, Lig3, Pnkp, Parp1 in murine and human cells while DNA damage checkpoint proteins like Atm, Gmnn, Tp53bp1 were specifically deregulated in murine cells highlighting some differences between the two models (Fig. 5c).

To address the functional consequences of these dysregulations, we compared the proliferation of the G1E-ER4 clones. Before GATA1 induction, the proliferation of *Sf3b1*$^{K700E/+}$ proerythroblasts was significantly higher than that of *Sf3b1*$^{+/+}$ cells and remained higher upon induction (*P* < 0.0001, Fig. 5d). The differentiation to basoE was significantly lower in *Sf3b1*$^{K700E/+}$ cells (Fig. 5e, f). In accordance with this, *Sf3b1*$^{K700E/+}$ cells showed a higher BrdU incorporation and a higher G1/S fraction than *Sf3b1*$^{+/+}$ cells, before and after induction (Fig. 5g). Concomitantly to GATA1 induction, when a mild replication stress was imposed by inducing a cell cycle arrest in early S-phase with

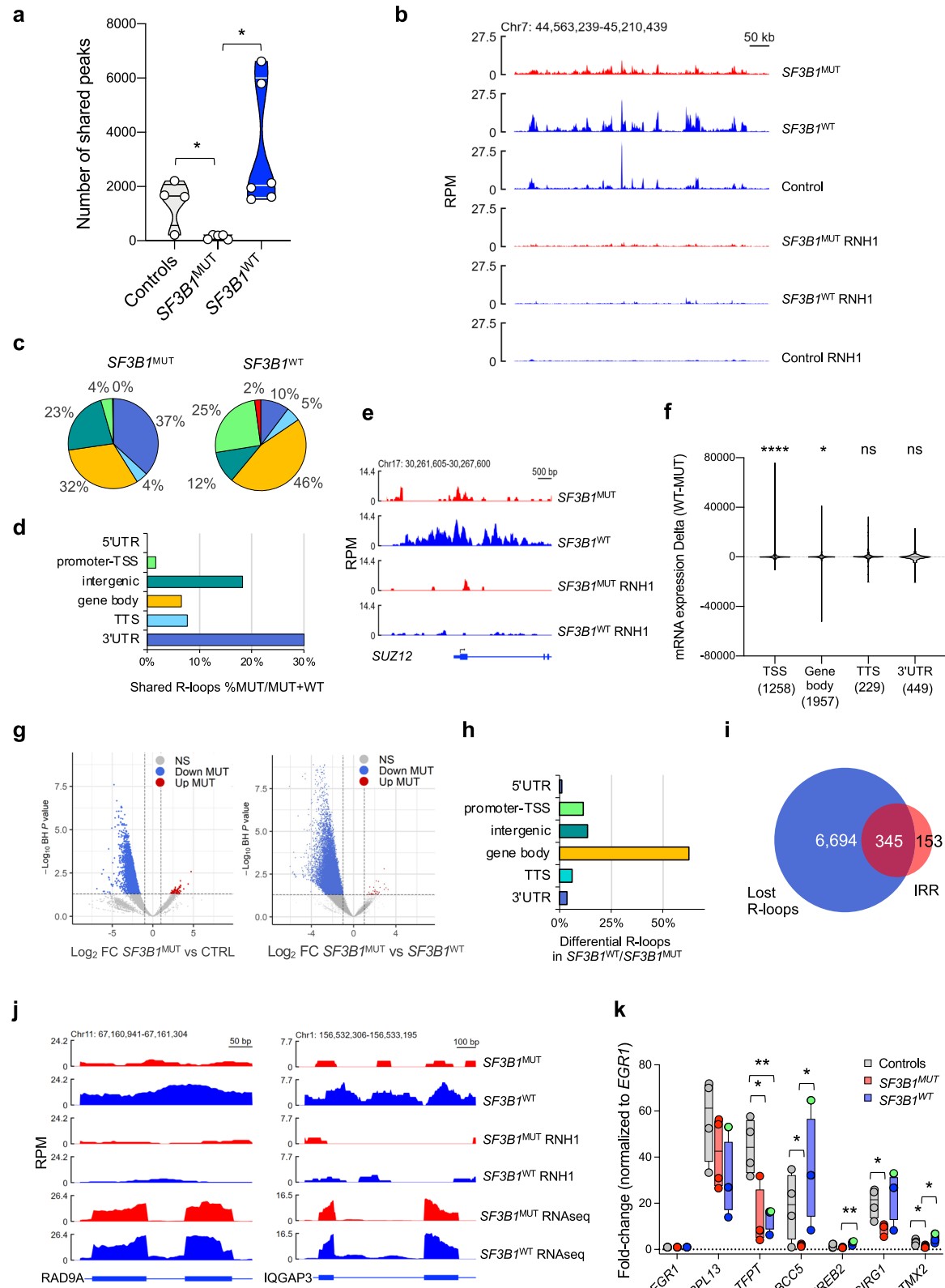

aphidicolin, $Sf3b1^{K700E/+}$ compared to $Sf3b1^{+/+}$ cells exhibited a significantly higher G1/S phase fraction (Fig. 5h, right panel). Inhibition of dNTP biosynthesis by 0.2 mM hydroxyurea (HU) for 16 h, normalized BrdU incorporation in $Sf3b1^{K700E/+}$ cells after induction (Fig. 5i). In metabolomic analysis, the quantities of dATP, dCTP and dTTP before induction were equivalent in $Sf3b1^{K700E/+}$ to $Sf3b1^{+/+}$ cells, showing that DNA replication stress was not related to nucleotide pool depletion

(Fig. 5j). After induction of differentiation with estradiol, dNTP quantities increased in $Sf3b1^{+/+}$ cells, consistent with a slowdown of DNA synthesis. In $Sf3b1^{K700E/+}$ cells, lower dNTP quantities argued for persistent DNA synthesis (Fig. 5j; Supplementary Fig. 9c).

Upon induction of differentiation, more $Sf3b1^{K700E/+}$ cells were positive to p-Rpa32s4/s8 labelling with a higher sensitivity to HU treatment than $Sf3b1^{+/+}$-cells (Fig. 5k). In estradiol-treated cells,

**Fig. 3 | Loss of R-loops overlaps intron retention reductions in *SF3B1*^MUT^ human primary erythroblasts.** DRIP-sequencing. **a** Violin plots showing the shared peak numbers in 4 controls, 5 *SF3B1*^MUT^ and 6 *SF3B1*^WT^ (4 *SF*^WT^, 2 *SRSF2*^MUT^). Two-sided unpaired t-test. Controls versus *SF3B1*^MUT^, $P = 0.011$; *SF3B1*^WT^ versus *SF3B1*^MUT^, $P = 0.015$. **b** DRIP-seq profiles ± RNaseH1 (RNH1) of a 50-kb region on chr7 showing the distribution of R-loops in reads per million. **c** Pie charts representing localizations of shared R-loops at gene features. **d** Proportion of shared peaks at gene features in *SF3B1*^MUT^ samples relative to *SF3B1*^MUT^ + *SF3B1*^WT^ samples. **e** DRIP-seq profiles showing R-loops near *SUZ12* promoter. **f** Comparison of the expression of genes with overlapping R-loops at TSS, gene body, TTS and 3'UTR in *SF3B1*^WT^ samples and without overlapping R-loops in *SF3B1*^MUT^ samples. Violon plots represent the difference of mean expression intensity between *SF3B1*^WT^ and *SF3B1*^MUT^ samples (d11). Central lines represent the means. Gene numbers in each category are indicated. One sample Wilcoxon signed rank test is used for comparison of actual mean to theorical mean. TSS, $P < 0.0001$; gene body, $P = 0.031$; TTS, $P = 0.179$; 3'UTR, $P = 0.118$. **g** Volcano plot representing differential restriction fragments overlapping with peaks between *SF3B1*^MUT^ and control samples (left panel) and *SF3B1*^MUT^ and *SF3B1*^WT^ samples (right panel) with $\log_2(FC) > |1|$ using two-sided Wald-test and a BH-adjusted $P$ value < 0.05. **h** Distribution to gene features of differential R-loops in *SF3B1*^WT^ samples and lost in *SF3B1*^MUT^ samples. **i** Venn diagram showing overlap between genes that lost one R-loop and genes with intron retention reduction (IRR) in *SF3B1*^MUT^ erythroblasts. **j** DRIP-seq and RNA-seq overlays at *RAD9A* and *IQGAP3* loci showing R-loop losses and IRR events in *SF3B1*^MUT^ erythroblasts. Gene structures using GENCODE GRCh37. **k** DRIP-qPCR analysis of 4 controls, 3 *SF3B1*^WT^ including 1 *U2AF1*^MUT^ designated as green dot and 4 *SF3B1*^MUT^ samples. Enrichment signals (normalized to input) at specific loci were normalized to *EGR1* (no R-loop). *RPL13A* and *TFPT* as positive controls. In box plots, central lines represent medians, bounds represent lower and upper quartiles and whiskers correspond to min-max values. Two-sided unpaired t-test for $P$ values (see Suppl informations). **b**, **e**, **j** RPM: reads per million. * $P < 0.05$; ** $P < 0.01$; **** $P < 0.0001$; ns: not significant. Source data are provided as a Source Data file.

Western blot confirmed the engagement of Rpa, but did not show phosphorylation of Chk1 suggesting that the Atr-Chk1 pathway was not activated (Fig. 5l). Finally, the delayed differentiation of *Sf3b1*^K700E/+^ murine erythroblasts can be partially rescued by lowering their high rate of DNA synthesis with HU at the expense of cell viability (Supplementary Fig. 9d, e).

### Targeting of R-loops improves the differentiation of human *SF3B1*^MUT^ erythroblasts

To establish a link between the loss of R-loops and the phenotypic characteristics of fork velocity and replication stress in human primary *SF3B1*^MUT^ erythroblasts, we thought to modulate the level of R-loops in the cell. Previous works have established that the THO complex which contributes to prevent R-loop accumulation interacts with SIN3A-histone deacetylase complex. Furthermore, inhibiting histone deacetylase activity by depleting SIN3A or treating the cells with trichostatin A stabilizes R-loops[40]. We used pan-HDAC inhibitor Suberoylanilide hydroxamic acid (SAHA)/vorinostat (further denoted HDACi) in DRIP-seq experiment. Human erythroblasts from 3 *SF3B1*^MUT^, 3 *SF3B1*^WT^ (1 *SRSF2*, 1 *SRSF2*/bi*TET2* or 1 *NRAS*/bi*TET2* mutations), and 4 controls samples were pre-treated with HDACi 0.5 μM for 20 h, at day11 of culture. The numbers of shared peaks increased in 3/4 controls, even not significantly. HDACi treatment restored the level of R-loops of 2/3 *SF3B1*^MUT^ erythroblast samples up to normal (Fig. 6a; Supplementary Data 6). Unexpectedly, the numbers of R-loops counted in *SRSF2* or *TET2/NRAS* mutated samples collapsed almost entirely ($P < 0.001$). In *SF3B1*^MUT^ samples, the number of shared R-loops was quantitatively important in the gene bodies. The augmentation of R-loops also affected intergenic regions, 3'UTR and TTS more than 5'UTR or promoter-TSS (Fig. 6b). We visualized the changes in R-loop profiles at specific loci. As shown in Fig. 6c and Supplementary Fig. 10, HDACi treatment produced large R-loops near the promoter of *BCL2L1*, *PTPN11*, *ARPC3* and *NCOA4* genes specifically in *SF3B1*^MUT^ cells. By contrast, HDACi did not change the profile of R-loops in *SF3B1*^MUT^ cells at *HK1* locus (Supplementary Fig. 10). To verify whether, by rescuing R-loops, gene expression may change, we performed RT-qPCR at these 4 loci. Upon treatment with HDACi, the expression of *BCL2L1* increased significantly in *SF3B1*^MUT^. While the expression of *NCOA4* and *PTPN11* also tended to increase, *HK1* did not (Fig. 6d). These data suggest the relationship between R-loops and gene expression in these cells.

Then we wondered whether the effect of HDACi could be at least partly due to a modification of IRR profiles. To explore this hypothesis, we selected 5 genes with known IRR (*PPOX*, *PPM1A*, *COASY*, *S100A4*, *BCL2L1*) and performed RT-PCR to visualize the IRR-transcripts and the spliced isoforms in 4 *SF3B1*^MUT^, 6 *SF3B1*^WT^ and 2 control erythroblast samples. HDACi did not modify the pattern of transcripts in controls and *SF3B1*^WT^ samples. The abundance of IRR-transcripts in *SF3B1*^MUT^ remained similar in the presence or absence of HDACi suggesting that the restoration of R-loops observed under this treatment did not depend on intron retention (Fig. 6e). We confirmed these results for *BCL2L1* and *COASY* transcripts by fluorescent PCR fragment analysis (Supplementary Fig. 11a, b).

To evaluate the impact of R-loop restoration on fork progression, we performed DNA combing of 3 samples of erythroblasts treated or not with 0.2 μM HDACi (1 control, 2 *SF3B1*^MUT^ including one with a frameshift mutation in the histone acetyltransferase *EP300*). The fork velocity in *SF3B1*^MUT^ erythroblasts was high and decreased significantly after HDACi treatment. By contrast, neither the *SF3B1*^MUT^/*EP300*^MUT^ sample nor the control showed variations of fork velocity after HDACi (Fig. 7a). To assess DNA synthesis in a larger number of patients, we performed BrdU assays. The percentage of S-phase cells decreased significantly in *SF3B1*^MUT^ cells. However, the BrdU intensity higher in *SF3B1*^MUT^ samples did not change (Fig. 7b). Then, we verified the impact of HDACi (0.5 μM 20 h) on the frequency of p-RPA32 s33 and γH2AX foci. No increase of positive cells was observed in control, *SF3B1*^MUT^ or *SF3B1*^WT^ erythroblast samples suggesting that HDACi at the concentration used did not provide DNA damage to these cells (Fig. 7c–e).

Finally, we studied the consequences of HDACi treatment on erythroid cell differentiation. At progenitor level, HDACi did not change the number or size of the BFU-E type colonies formed in methylcellulose (Fig. 7f). Looking at mutations in single colonies we did not observe any clonal selection during the 14 days of semi-solid culture (Supplementary Fig. 11c). By contrast, HDACi drastically forced the maturation of erythroblasts at late stage as shown by a significant increase of mature erythroblasts (Fig. 7g, h). This effect was also confirmed by the increase of GPA^+^CD49d^low^ cell proportion corresponding to orthochromatic erythroblasts (Fig. 7i). Interestingly we did not observe any inhibitory effect of HDACi at 0.5 μM on the overall rate of expansion of *SF3B1*^MUT^ erythroid precursors compared to *SF3B1*^WT^ or control erythroblasts (Supplementary Fig. 11d), or increase of cell death in *SF3B1*^MUT^, *SF3B1*^WT^, and control cell cultures (Fig. 7j). Altogether, these results showed that HDACi by producing R-loops, slowed DNA replication and facilitated erythroid cell differentiation without altering cell proliferation.

## Discussion

The present study shows that *SF3B1* mutations causing an increased proliferation of immature erythroblasts with a reduced capacity to terminal differentiation trigger a replication stress with ssDNA exposure in erythroid cells. Accelerated DNA replication fork velocity is observed when R-loops are lost. HDAC inhibition restores R-loops and decreases fork speed, which correlates with erythroid differentiation improvement. These features distinguish *SF3B1* mutation from other splicing factor mutations.

Alternative splicing of pre-mRNA contributes to physiological hematopoiesis[26–28,41]. Intron retentions increase along terminal steps of erythroid differentiation[28]. Intron retention modulates gene

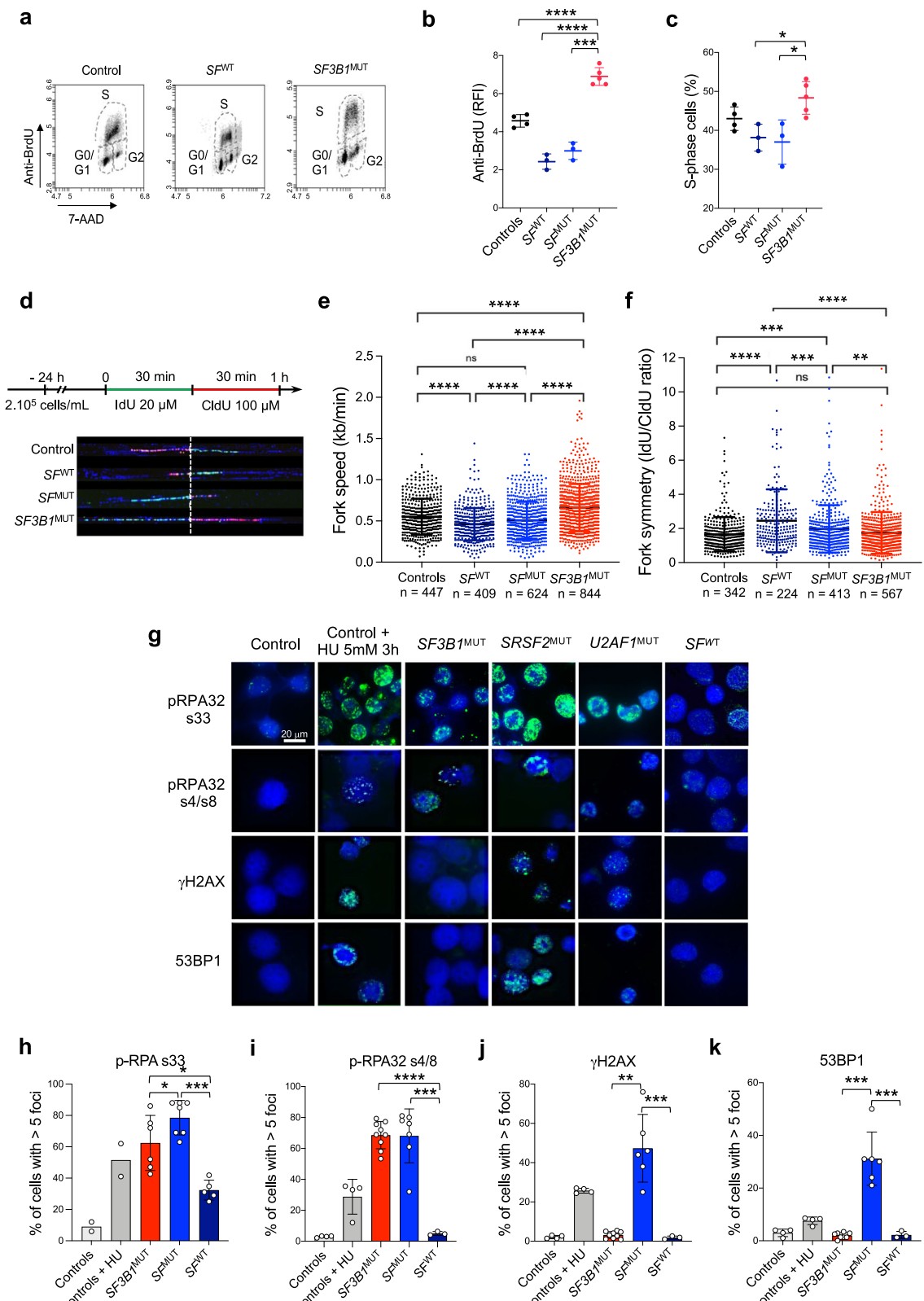

expression by generating transcripts that are either detained in the nucleus or degraded in the cytoplasm by the NMD[41,42]. Here, we show that *SF3B1*[MUT] promotes a reduction of intron retention in erythroblasts and that the number of retained introns lost in *SF3B1*[MUT]-erythroid precursors increases between basoE and polyE. By reversing a physiological process, *SF3B1* mutation changes gene expression profile and reshapes the proteome, affecting several pathways such as

DNA replication, DNA repair mainly base excision repair and nucleotide biosynthesis. *SF3B1*[MUT]-erythroblasts retain proliferative capacities, which contributes to enrichment of the bone marrow in immature erythroblasts and to defective production of mature erythroblasts, defining ineffective erythropoiesis.

To synthesize DNA properly, the replication machinery must overcome several obstacles, including R-loops and transcription

**Fig. 4 | DNA replication stress in *SF3B1*-mutated human primary erythroblasts.**
Erythroblasts were derived in culture from 9 *SF3B1*[MUT], 11 *SF3B1*[WT], 6 controls. **a**–**c**.
Cell cycle analysis by BrdU incorporation. **a** Representative flow cytometry scatter plot of control, *SF*[WT] and *SF3B1*[MUT] samples. **b, c** Scatter plots representing ratios of fluorescence intensity (RFI) anti-BrdU antibody/control Ig (**b**) and percentages of S-phase cells (**c**) in 4 controls, 3 *SF*[WT], 3 *SF*[MUT] and 5 *SF3B1*[MUT] samples. Results are expressed as means ± SD. Two-sided unpaired t-test. **b** Controls vs *SF3B1*[MUT], *P* = 0.003; *SF*[WT] vs *SF3B1*[MUT], *P* < 0.0001; *SF*[MUT] vs *SF3B1*[MUT], *P* < 0.0001. **c** *SF*[WT] vs *SF3B1*[MUT], *P* = 0.012; *SF*[MUT] vs *SF3B1*[MUT], *P* = 0.017. **d**–**g** DNA combing performed in proE/basoE from 4 *SF3B1*[MUT], 3 *SF*[MUT], 2 *SF*[WT] MDS and 2 controls. **d** Upper panel: Schematic representation of IdU/CldU pulse labelling. Bottom panel: Representative microphotographs of DNA fibers. **e** Scatter plot showing fork speed (kb/min) expressed in means ± SD. Numbers of fibers analyzed are indicated. Two-sided Mann-Whitney test. Controls vs *SF3B1*[MUT], *P* < 0.0001; *SF*[WT] vs *SF3B1*[MUT], *P* < 0.0001;

*SF*[MUT] vs *SF3B1*[MUT], *P* < 0.0001; *SF*[WT] vs *SF*[MUT], *P* = 0.0009; *SF*[WT] vs controls, *P* < 0.0001; *SF*[MUT] vs controls, *P* = 0.002. **f** Scatter plot showing fork symmetry as ratios of IdU/CldU length expressed in means ± SD. Two-sided Mann-Whitney test. Controls vs *SF3B1*[MUT], *P* = 0.302; *SF*[WT] vs *SF3B1*[MUT], *P* < 0.0001; *SF*[MUT] vs *SF3B1*[MUT], *P* = 0.005; *SF*[WT] vs *SF*[MUT], *P* = 0.0003; *SF*[WT] vs controls, *P* < 0.0001; *SF*[MUT] vs controls, *P* = 0.0004. **g**–**k** Immunofluorescence experiments in MDS or control erythroblasts treated or not with 5 mM hydroxyurea (HU). **g** Representative images of phospho(p)-RPA32s33, p-RPA32s4/s8, γH2AX and 53BP1 at d11. Nuclei were labeled with DAPI. Magnification 100X (scale: 20 μm). **h**–**k** Quantification of positive cells with >5 intranuclear foci. **h** p-RPA32s33 (2 controls, 6 *SF3B1*[MUT], 7 *SF*[WT], 5 *SF*[MUT]). **i** p-RPA32s4/8. **j** γH2AX. **k** 53BP1 (4 controls, 9 *SF3B1*[MUT], 7 *SF*[WT], 3 *SF*[WT]). Results are expressed as mean percentages of positive cells ± SD. Two-sided unpaired t-tests; * *P* < 0.05; ** *P* < 0.01; *** *P* < 0.001; **** *P* < 0.0001; ns: not significant. Source data are provided as a Source Data file.

complexes[37]. Most of the R-loops located at promoter-TSS, in gene bodies and intergenic regions in *SF3B1*[WT]-erythroblasts were lost in *SF3B1*[MUT]-erythroblasts, in contrast with the augmented R-loops detected in *SRSF2*[MUT] erythroblasts. Previous studies in cell lines or in primary CD34+ *SF*[MUT] progenitors using single-cell imaging with S9.6 antibody have shown that not only *SRSF2*[MUT] or *U2AF1*[MUT], but also *SF3B1*[MUT] cells may produce undesirable R-loops[19–21]. However, S9.6 foci may indicate RNA:DNA hybrids and also double-stranded RNA which are more abundant, making difficult the quantification of R-loops even when RNaseH1 is overexpressed in cell lines to assess the specificity of the signals[19,43,44]. Here we used DRIP-seq to obtain a genome-wide landscape of R-loops. This technique allowed us to identify short type I R-loops located at GC-skewed promoter-proximal regions, or in intergenic regions that may contain active enhancers[21], which are preferentially detected by R-ChIP, and also, large type II R-loops (spanning over 300 bp to 1 kb) distributed along the body of transcribed genes[45,46]. The decreased number of R-loops at promoters-TSS and gene bodies in *SF3B1*[MUT] cells suggest that both large and short R-loops are lost. As the GC skew characteristic of R-loops at promoter progressively diminishes after the first exon/intron junction, the mechanism of R-loop formation may be different in gene bodies[45]. Intron retention, by increasing homology between nascent RNA and its DNA template can initiate the co-transcriptional formation of large R-loops spreading over the gene coding sequence[22,29,47,48]. Conversely, the binding of splicing machinery making intron excision hinders R-loop accumulation[30,33,49]. Increased intron excision occurring when *SF3B1* is mutated may suppress R-loops along the gene bodies. We report here that most of IRR overlap with lost R-loops. However, because R-loops were lost also in promoter-TSS, intergenic regions and TTS, IRR is not the unique mechanism for R-loop loss in these cells.

The presence of unscheduled R-loops at promoter-TSS as a consequence of RNA polymerase II pausing usually correlates with high gene expression[45]. R-loop-positive regions overlap with DNase I-hypersensitive regions, indicative of open chromatin[21,45]. Our integrative analysis of R-loops and gene expression associated the loss of R-loops with low transcript level in *SF3B1*[MUT] erythroblasts. Importantly, we identified *BCL2L1*, a GATA1 and STAT5 target gene, which expression increased, without splicing changes, when R-loops formed near its promoter under HDACi treatment. This is consistent with the interference of R-loops with transcription. In addition, since SF3B1 protein together with U2AF1 and SRSF2 were detected in the vicinity of R-loops at promoters[50], the SF3B1 mutant protein, could notably modify the kinetics of transcription elongation as generally reported for splicing factors[51].

Replication stress that appears at the early stages of malignant cell transformation, has been linked to events that impair DNA synthesis integrity such as reduced or increased origin firing, nucleotides or replication factor depletion and replication-transcription conflicts[52,53]. Replication stress can produce stalled forks, under-replicated DNA, or supra-acceleration of forks[54]. As opposed to *SRSF2*[MUT] or *U2AF1*[MUT] cells in which fork progression is slowed down, we detected an accelerated

fork speed when R-loops were lost in *SF3B1*[MUT]-erythroblast. Alternative causes of increased fork velocity were associated with over-expression of oncogenes, downregulation of mRNA biogenesis, or PARP1 inhibition[55–58]. Moreover, activation of oncogenic HRAS in pre-senescent cells was shown to accelerate forks by inducing over-expression of topoisomerase 1 (TOP1), which is known to resolve unwanted R-loops[59]. TOP1 was heavily expressed in *SF3B1*[MUT] and *SF3B1*[WT] erythroblasts, while senataxin and THO complex proteins, other R-loop modulators were specifically upregulated in *SF3B1*[MUT] erythroblasts and could contribute to fork velocity[60]. Furthermore, if the transcription rate decreases when R-loops are resolved, not only R-loop loss but also the reduction of transcription-replication conflicts could prevent replication fork stalling[61,62].

The detection of p-RPA32 foci without DNA damage marks γH2AX or 53BP1 suggested replicative ssDNA gaps. ATR continuously monitors the recruitment of RPA32 on ssDNA within the replisomes and its phosphorylation on serine 33, independently of CHK1[63]. *SF3B1*[MUT]-erythroblasts endure a mild replication stress without engagement of the CHK1 pathway (Fig. 5l). In cancer cells, the tolerance to replication stress is supported by the overexpression of the upstream components of ATR-CHK1 pathway, Clapsin and Timeless, independently of ATR signaling[64]. As *SF3B1*[MUT]-cells overexpressed Claspin and Timeless genes and Timeless protein, we cannot exclude their role in the mechanism of replication stress tolerance. Alternatively, RPA could bind the displaced ssDNA of R-loops and recruit RNaseH1 to facilitate its resolution[65]. Further studies are needed to test these hypotheses.

Screening for synthetic lethal approaches using *SF3B1*[MUT]-cell lines have revealed potential vulnerabilities by targeting of ATR, PARP1 or NMD[19,65–67]. An alternative, more conservative approach to improve ineffective erythropoiesis could be slightly restoring R-loops. The formation and degradation of R-loops involve multiple actors that include splicing factors, topoisomerases, DNA/RNA helicases, DNA repair molecules (BRCA1, BRCA2 or FANCD2/A/M), RNaseH1 or H2 which degrade R-loops directly, or SAMHD1 (alpha motif and HD-domain containing protein 1) that promotes ssDNA degradation at stalled forks[68,69]. Chromatin accessibility also interferes with R-loop generation[62]. In accordance, R-loops were shown to increase in HDACi-treated or SIN3A-depleted cells[40]. We show here that R-loops reappeared in *SF3B1*[MUT] cells treated with HDACi, which correlated with significant increase of *BCL2L1* expression and improvement of erythroid differentiation without evidence of DNA damage. Since clinical trials have shown a limited hematotoxicity of vorinostat alone compared to its combination with azacitidine[70,71], our results provide a rationale for testing if low doses of vorinostat could improve erythropoiesis of patients with *SF3B1*[MUT] MDS.

## Methods
### Patients
A total of 143 subjects including 70 MDS patients with *SF3B1* mutation, 49 lower-risk (LR)-MDS patients without *SF3B1* mutation (17 *SRSF2*[MUT],

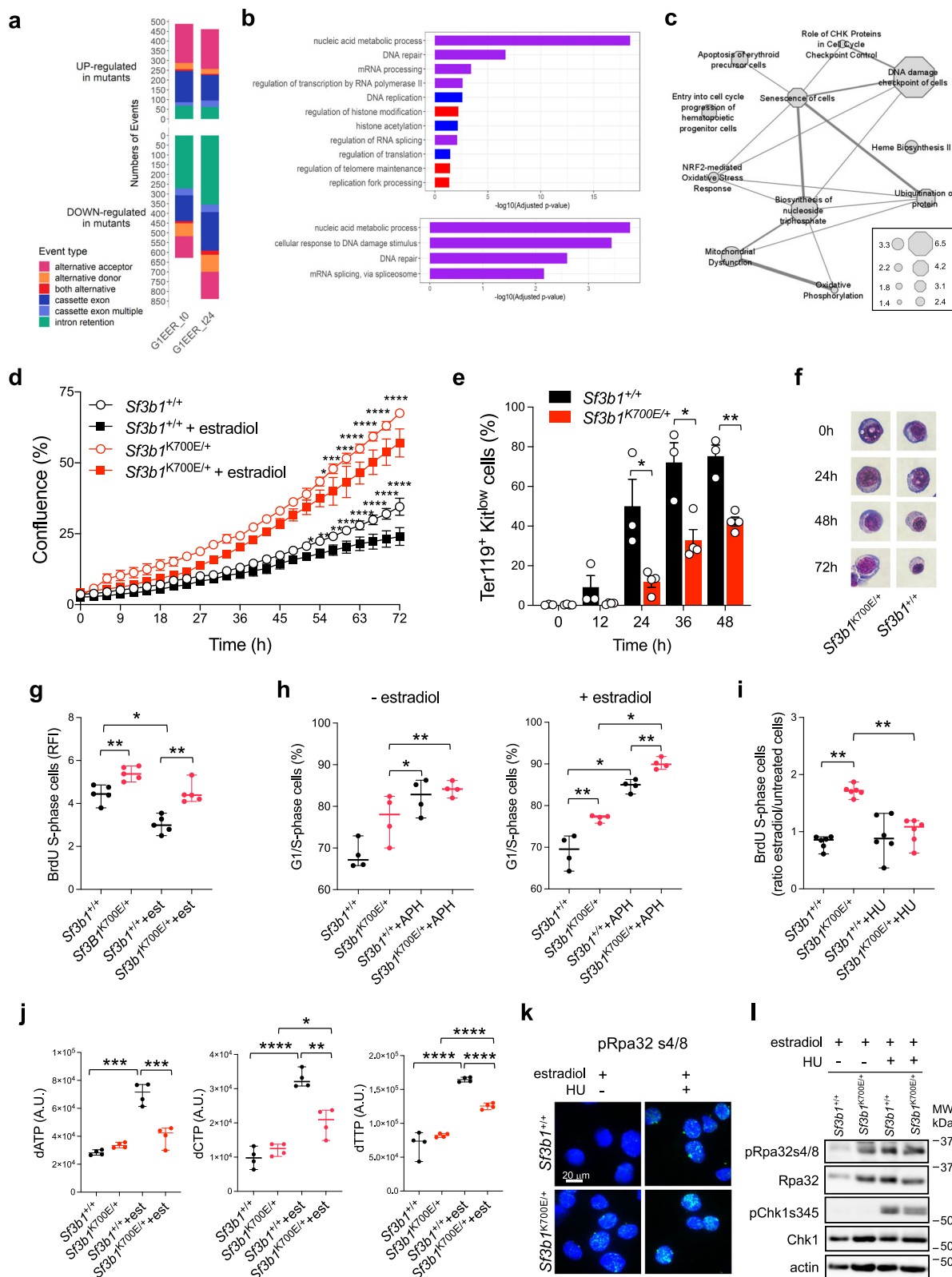

7 $U2AF1^{MUT}$, 25 triple-negative as $SF^{WT}$) and 24 age-matched healthy subjects as controls was enrolled between 2015 and 2023. Bone marrow (BM) aspirates were collected after each patient gave his informed consent for biological investigations according to the recommendations of institutional review board (IRB) and ethics committee (IRB numbers: IdFV 212-A01395-38 EudraCT 2012-002990-7338; OncoCCH 2015-08-11-DC). BM aspirates or femoral head samples were

collected from healthy subjects. Patient characteristics including age, gender, WHO, hemogram, BM blast, erythroblast and ring sideroblast percentages, karyotype, IPSS-R are indicated in (Table 1).

### Genomic studies
BM mononuclear cells (MNC) were purified on Ficoll gradient and were processed for DNA extraction using the DNA/RNA Kit (Qiagen, Hilden,

**Fig. 5 | DNA replication stress in murine G1E-ER4 Crispr-Cas9 *Sf3b1*[K700E/+] proerythroblasts. a** Barplots representing numbers and types of differential splicing events in *Sf3b1*[K700E/+] (clone 5.13) versus *Sf3b1*[+/+] (clone 9.82) cells at t0 (proE) and t24 (basoE) after induction of differentiation with estradiol (ΔPSI > |0.10| using two-sided Wald test and BH-adjusted *P* value < 0.05). **b** Gene Ontology (GO) over-representation analysis. Upper panel: Pathways involving differentially spliced genes in *Sf3b1*[K700E/+] cells shared at t0 and t24 (violet bars), specific to t0 (blue bars) or t24 (red bars). Bottom panel: Pathways involving genes with IRR in *Sf3b1*[K700E/+] cells. Shared GO terms at t0 and t24. Fisher's exact test corrected by false discovery rate (FDR) < 0.05. **c** Ingenuity Pathway Analysis of differential proteins at t0 (Student t-test, *P* values < 0.05. Canonical Pathways (hexagons), Diseases and Functions pathways (circles). **d** Live cell imaging. Mean percentages (± SD) of confluence (n = 3). 2-way ANOVA test for multiple comparisons. **e** Differentiation of *Sf3b1*[K700E/+] (n = 4) and *Sf3b1*[+/+] cells (n = 3) by flow cytometry. Mean percentages of Ter119+Kit[low] cells ± SEM. Unpaired t-test for multiple comparisons. t24h: *q* = 0.033; t36h: *q* = 0.027; t48h: *q* = 0.007. **f** May-Grünwald-Giemsa-stained cytospins. **g** BrdU incorporation in S-phase ± estradiol 24 h. Medians ± 95% confidence intervals (CI) of RFI anti-BrdU antibody/control Ig (5 independent experiments). *Sf3b1*[+/+] vs *Sf3b1*[K700E/+], *P* = 0.002; *Sf3b1*[+/+]+estradiol vs *Sf3b1*[K700E/+]+estradiol, *P* = 0.001; *Sf3b1*[+/+] vs *Sf3b1*[K+/+]+estradiol, *P* = 0.011. **h** Cell cycle analysis by BrdU incorporation ± aphidicolin (APH). Median percentages (± 95% CI) of G1/S-phase cells (4 independent experiments). Left: *Sf3b1*[+/+] vs *Sf3b1*[K700E/+], *P* = 0.035; *Sf3b1*[+/+] vs *Sf3b1*[+/+]+APH, *P* = 0.002. Right: *Sf3b1*[+/+] vs *Sf3b1*[K700E/+], *P* = 0.007; *Sf3b1*[+/+]+APH vs *Sf3b1*[K700E/+]+APH, *P* = 0.002; *Sf3b1*[+/+] vs *Sf3b1*[+/+]+APH, *P* = 0.029; *Sf3b1*[K700E/+] vs *Sf3b1*[K700E/+]+APH, *P* = 0.029. **i** BrdU incorporation in S-phase ± hydroxyurea (HU). Medians ± 95% CI of RFI estradiol-treated/untreated cells (5 independent experiments). **j** dNTP relative quantities. Medians ± 95%CI (4 independent experiments). **g–j** Two-sided unpaired t-test. **k** Immunofluorescence of pRpa32s4/8 (± estradiol 24h, HU 16 h). **l** Western blot of pRpa32s4/s8, Rpa32, pChk1s345 and Chk1. Actin as loading control. **k, l** Representative of 3 independent experiments. **** *P* < 0.0001, *** *P* < 0.001, ** *P* < 0.01, * *P* < 0.05; ns not significant. Source data are provided as a Source Data file.

Germany). Mutations in *SF3B1* were screened by Sanger sequencing or next generation sequencing of a panel of 37 genes (*ASXL1, ATM, BCOR, BCORL1, BRAF, CBL, CEBPA, CUX1, DDX41, DNMT3A, EP300, ETV6, EZH2, FLT3, GATA2, IDH1, IDH2, JAK2, KIT, KRAS, NRAS, MPL, NPM1, PHF6, PTPN11, RAD21, RIT1, RUNX1, SETBP1, SF3B1, SRSF2, STAG2, TET2, TP53, U2AF1, WT1, ZRSR2*).

### Primary cells and cell lines cultures
BM CD34+-derived erythroblasts were expanded from 92 MDS (49 *SF3B1*[MUT] and 43 *SF3B1*[WT] including 24 *SF*[WT], 13 *SRSF2*[MUT] and 6 *U2AF1*[MUT]) and 24 healthy donors as controls. For erythroid cell expansion, CD34+ cells were isolated from the MNC fraction of BM samples using the MidiMacs system (Miltenyi Biotec, Bergisch Gladbach, Germany). CD34+ cells, which purity was higher than 80% were then cultured at 0.8 × 10^6 per mL for 4 days in Iscove's modification of Dulbecco medium (IMDM; Thermo Fisher Scientific, Waltham, MA) containing 15% BIT9500 (Miltenyi Biotech), 100 U/mL penicillin, 100 µg/mL streptomycin, 2 mM L-glutamine (all from Thermo Fisher Scientific), 2 UI/mL recombinant human erythropoietin (rHu Epo; Roche, Basel, Switzerland), 100 ng/mL stem cell factor (SCF; Miltenyi Biotech), 10 ng/mL interleukin 6 (IL6; Miltenyi Biotech) and 2.10^−7 M dexamethasone (Merck, Darmstadt, Germany). Cells were diluted every day in the same medium until day 4. From day 4, IL6 was removed. From days 10 to 16 cells were switched to rHu Epo (2 UI/mL) to obtain terminal erythroid differentiation.

The murine G1E-ER4 cell line expressing a GATA1-estrogen receptor fusion gene[72] was cultured in IMDM, containing 20% fetal calf serum (FCS; GE Healthcare, Chicago, IL), 100 U/mL penicillin, 100 µg/mL streptomycin, 2 mM L-glutamine, 2 U/mL Epo, SCF in Chinese Hamster Ovary cell conditioned medium, monothioglycerol and 0.5 µg/mL puromycin (Merck, Darmstadt, Germany) to select cells expressing the GATA1-ER fusion gene. This cell line was used to edit mutant *Sf3b1*[K700E] and isogenic *Sf3b1*[WT] using CRISPR-Cas9 strategy[11].

For some experiments, cells were arrested in G1/S phase with 0.6 µg/mL aphidicolin (APH, Merck, Darmstadt, Germany, cat no. 38966-21-1) or with 0.2 mM hydroxyurea (HU, Merck, cat no. H8627), or treated with 0.1 to 1 µM histone deacetylase inhibitor SAHA/vorinostat (HDACi, Merck, cat no. #SML0061) for 20 h.

### RNA-sequencing
RNA-seq data from a first cohort of BM MNC from 27 lower-risk (LR)-MDS (21*SF3B1*[MUT] and 6 *SF3B1*[WT] (4 *SRSF2*[MUT] and 2 *SF*[WT]) previously published[2] were re-analyzed. RNA-sequencing of two additional cohorts was performed: one cohort of BM MNC from 185 LR-MDS (74 *SF3B1*[MUT], 30 *SRSF2*[MUT], 11 *U2AF1*[MUT], 70 triple-negative samples) and one cohort of basoE and/or polyE obtained from 13 MDS (8 *SF3B1*[MUT] and 5 *SF3B1*[WT]). RNA integrity (RNA integrity number ⩾ 7.0) was checked on the Agilent Fragment Analyzer (Agilent, Santa Clara, CA)

and quantities were determined using Qubit (Invitrogen, Waltham, CA). 50-100 ng of total RNA sample was used for poly-A mRNA selection using oligo(dT) beads and subjected to thermal fragmentation. For BM MNC, MuLV Reverse Transcriptase (Invitrogen) was used for cDNA synthesis. Libraries were constructed using the TruSeq Stranded mRNA Sample Preparation Kit (Illumina, San Diego, CA) and sequenced on an Illumina HiSeq 2500 platform using a 100-bp paired-end sequencing strategy. For RNA-sequencing of erythroblasts, fragmented mRNA samples were subjected to cDNA synthesis, converted into double stranded DNA using SureSelect Automated Strand Specific RNA Library Preparation Kit. Libraries were bar-coded, and subjected to 100-bp paired-end sequencing on Novaseq-6000 sequencer (Illumina, San Diego, CA). For RNA-sequencing of murine G1E-ER4 erythroblasts, libraries were constructed using the TruSeq Stranded mRNA Sample Preparation Kit (Illumina) and sequenced on an Illumina NextSeq500 platform (Illumina) using a 75-bp paired-end sequencing strategy.

### Bioinformatic analysis of RNA-sequencing
FASTQ files were mapped using STAR (v2.7.9) to align the reads against the human reference genome GRCh37 (UCSC version hg19) downloaded from the GENCODE project website. Reference genome and annotations are available on Gencode (https://www.gencodegenes.org/ human/release_19.html)[73]. Read count normalizations and groups comparisons were performed using DESeq2 (v1.30.1), with the Wald test for significance testing. Genes with low counts were filtered, if at least half the samples have less than 20 normalized reads. Outliers removal with Cook's distance was left at default. For differential expression study, the results obtained after DESeq2 comparison were selected for further analysis and filtered at Benjamini-Hochberg (BH)-adjusted *P* value < 0.05 and log2(FC) > |1|[74]. To identify differentially expressed splicing events, we used KisSplice (v2.6.2), KisSplice2refgenome (v2.0.7) and KissDE (v1.15.3)[75,76], and filtered results at BH-adjusted *P* value < 0.05 and ΔPSI > |0.10|. Biological processes associated with genes that were differentially expressed or spliced were determined using the Gene Ontology enrichment or over-representation analyses and Gene Set Eenrichment analysis with reference to specific gene sets (Supplementary Data 7).

### Sample preparation and mass spectrometry analysis
Sample preparation was done using the FASP procedure[77]. Briefly, cells were solubilized in 100 µL of Tris/HCl 100 mM pH8.5 buffer containing 2% sodium dodecyl sulfate (SDS). Total protein amounts were quantified using BiCinchoninic acid Assay (BCA; Thermo Fischer Scientific). Then, 10 mM TCEP and 40 mM chloroacetamide were added and the samples were boiled for 5 min. 50 µg of proteins were sampled and treated with urea to remove SDS. After urea removal, proteins were digested overnight with 1 µg of sequencing-grade modified trypsin

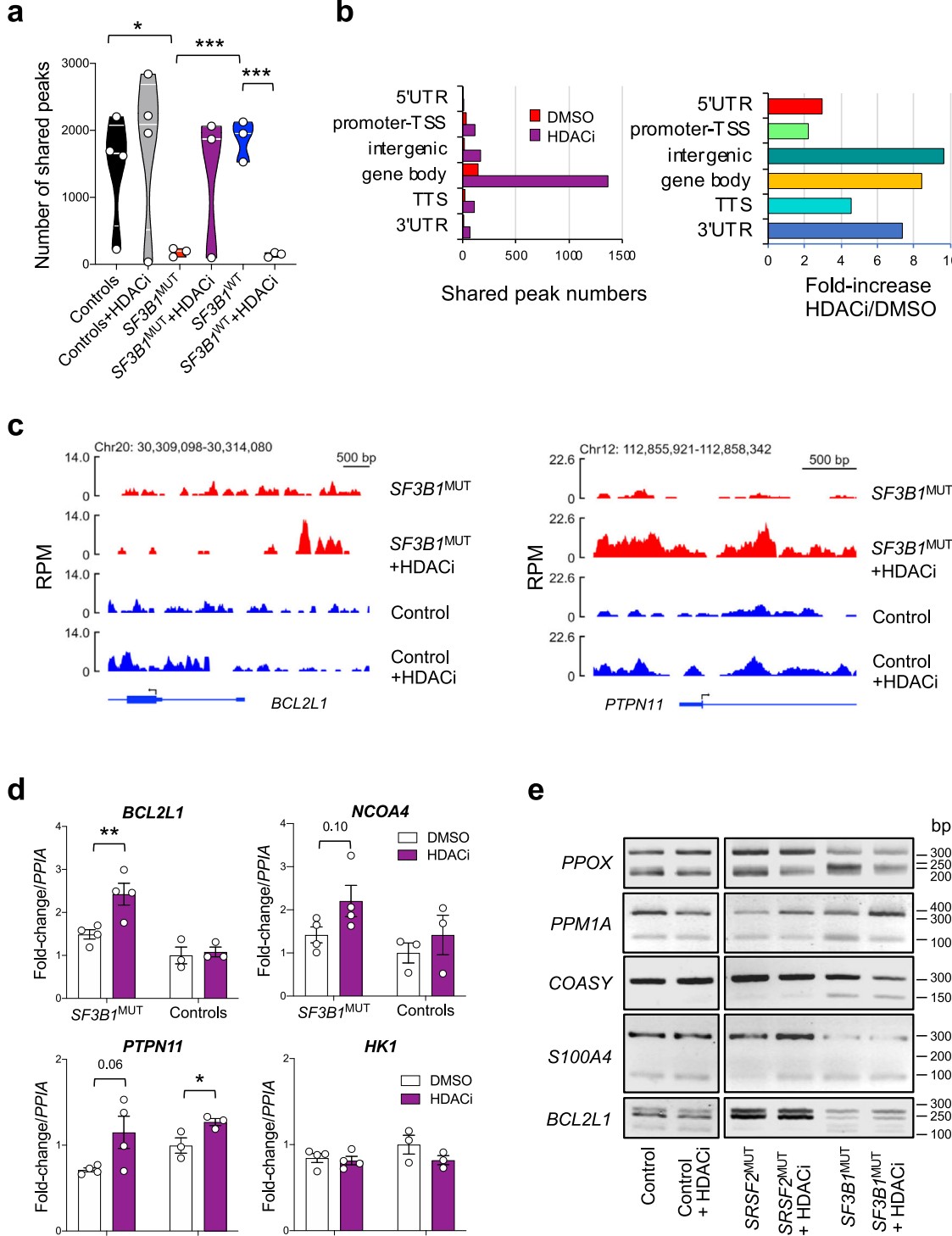

**Fig. 6 | Targeting of R-loops by HDACi may change gene expression without modifying the pattern of spliced isoforms. a** Violin plots with medians and quartiles representing R-loops as shared peak numbers by DRIP-seq of 3 *SF3B1*MUT, 3 *SF3B1*WT (2 *SRSF2*MUT, 1 *TET2/NRAS*MUT) and 4 control erythroblasts ± HDACi at 0.5 µM for 20 h. Two-sided unpaired t-test. **b** R-loop annotation to gene features in *SF3B1*MUT samples. Left panel: Shared peak numbers. Right panel: Fold-increase of peak number between HDACi and DMSO conditions. **c** R-loop profiles

near *BCL2L1* and *PTPN11* promoter. RPM: reads par million. **d** Quantification of *BCL2L1, NCOA4, PTPN11* and *HK1* transcripts by RT-qPCR in 4 *SF3B1*MUT and 3 control samples. Mean quantities normalized to *PPIA* ± SD. Two-sided unpaired t-test. **e** Expression of *PPOX, PPM1A, COASY, S100A4*, and *BCL2L1* transcript isoforms by RT-PCR representative of 4 *SF3B1*MUT, 6 *SF3B1*WT and 2 control samples. **** *P* < 0.0001, *** *P* < 0.001, ** *P* < 0.01, * *P* < 0.05. Source data are provided as a Source Data file.

(Promega, Madison, Wi) in 50 mM Tris/HCl pH8.5 buffer. Peptides were recovered by filtration, desalted on C18 reverse phase StageTips and dried. They were then separated in 5 fractions by strong cationic exchange StageTips and analyzed using an Orbitrap Fusion mass

spectrometer (Thermo Fisher Scientific). Peptides from each fraction were separated on a C18 reverse phase column (2 µm particle size, 100 A pore size, 75 µm inner diameter, 25 cm length) with a 170 min gradient starting from 99% of solvent A containing 0.1%

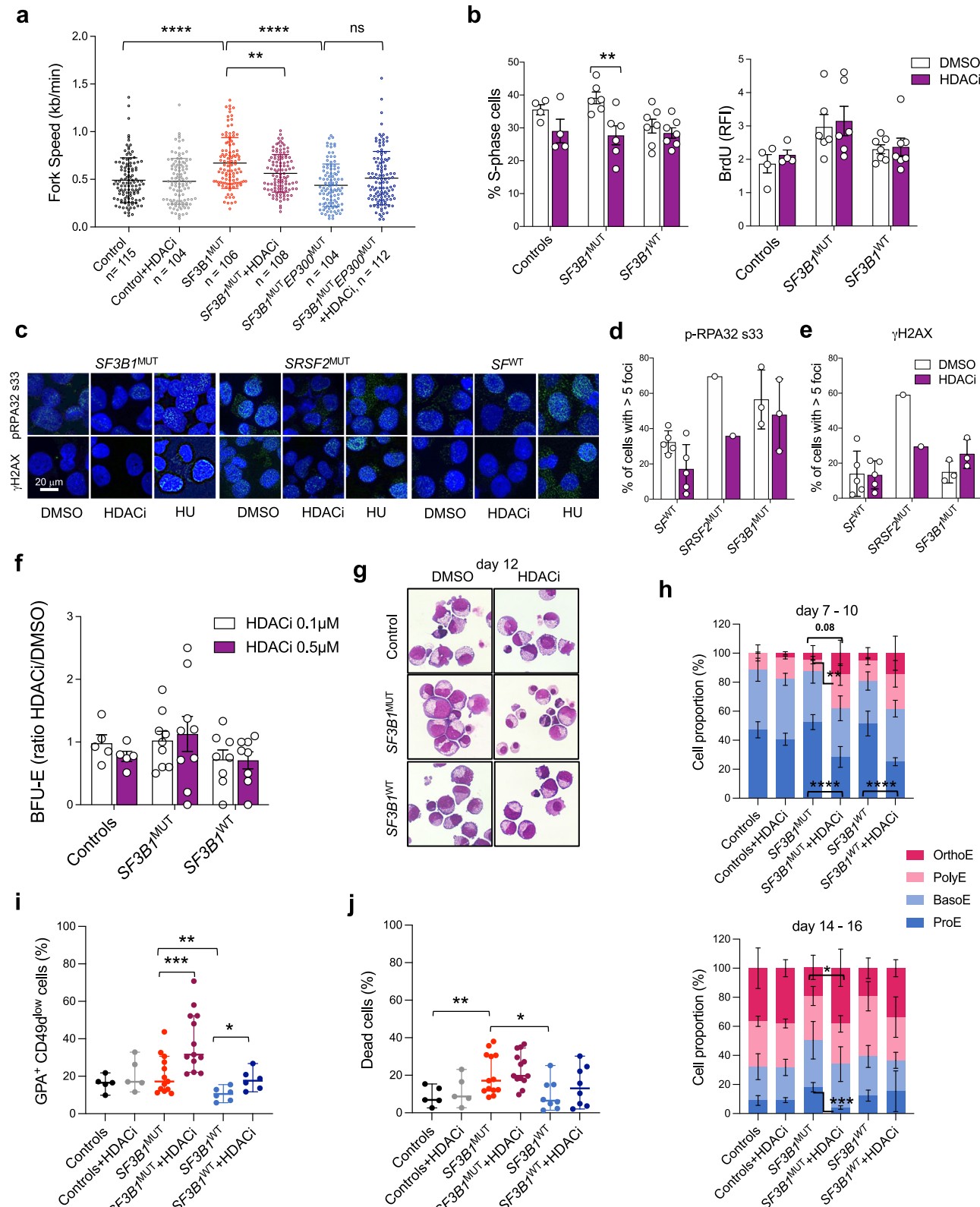

formic acid in milliQ-H$_2$O and ending in 55% of solvent B containing 80% acetonitrile and 0.085% formic acid in milliQ-H$_2$O. The mass spectrometer acquired data throughout the elution process. The MS1 scans spanned from 350 to 1500 Th with 1.10$^6$ Automatic Gain Control (AGC) target, 60 ms maximum ion injection time (MIIT) and resolution of 60 000. MS Spectra were recorded in profile mode. High

energy Collision Dissociation (HCD) fragmentations were performed from the most abundant ions in top speed mode for 3 seconds with a dynamic exclusion time of 30 s. Precursor selection window was set at 1.6Th. HCD Normalized Collision Energy was set at 30% and MS/MS scan resolution was set at 30000 with AGC target 1.10$^5$ within 60 ms MIIT.

**Fig. 7 | HDACi improves the differentiation of human *SF3B1*^MUT erythroblasts.**
**a** Scatter plot of fork speed measurement (kb/min) by DNA combing in 2
*SF3B1*^MUT (1 *SF3B1/DNMT3A*, 1 *SF3B1/TET2/EP3OO*) and 1 control ± HDACi 0.2 μM
for 20 h expressed in means ± SD. Two-sided Mann-Whitney test. **b** Cell cycle
analysis by BrdU incorporation in 6 *SF3B1*^MUT, 7 *SF3B1*^WT and 4 controls. Left panel:
Mean percentages ± SD of S-phase cells. Right panel: Mean BrdU RFIs ± SD in
S-phase. Two-sided paired t-test. *SF3B1*^MUT versus *SF3B1*^MUT + HDACi, $P = 0.002$.
**c**−**e** Immunofluorescence images of p-RPA32s33 and γH2AX representative of 3
*SF3B1*^MUT, 6 *SF3B1*^WT MDS (1 *SRSF2*^MUT, 3 *SF*^WT, 2 w/o mutation) at day 11. Nuclei
labelling with DAPI. Magnification X100 (scale: 20 μm). **d**, **e** Mean percentages ±

SD of positive cells with > 5 intranuclear foci. **f**. Burst forming unit-erythroid
(BFU-E) colony assays in 9 *SF3B1*^MUT, 8 *SF3B1*^WT and 5 controls. Mean ratios
between HDACi and DMSO conditions ± SD. **g** May-Grünwald-Giemsa-stained
cytospins (d12). **h** Proportions of erythroid precursors in 7 *SF3B1*^MUT, 3 *SF3B1*^WT
and 4 controls at d7-10 and d14-16. Means ± SEM and 2-way ANOVA multiple
comparisons for *q* values. **i** Scatter plots showing differentiation by flow cyto-
metry expressed as mean percentages ± SD of GPA⁺CD49d^low cells. **j** Scatter plots
showing mean percentages ± SD of dead cells (FSC/SSC) at d12-14. **i**, **j** Two-sided
paired t-test for *P* values. **** $P < 0.0001$, *** $P < 0.001$, ** $P < 0.01$, * $P < 0.05$.
Source data are provided as a Source Data file.

## Proteomic data analysis

The mass spectrometry data were analyzed using Maxquant version
2.1.1.0[78]. The database used was a concatenation of Human sequences
from the Uniprot-Swissprot database (Uniprot, release 2022-05) and
the list of contaminant sequences from Maxquant. Cystein carbami-
domethylation was set as constant modification and acetylation of
protein N-terminus and oxidation of methionine were set as variable
modifications. Second peptide search and the "match between runs"
(MBR) options were allowed. False discovery rate (FDR) was kept
below 1% on both peptides and proteins. Label-free protein quantifi-
cation (LFQ) was done using both unique and razor peptides with at
least 2 peptides ratios required for LFQ.

Statistical analysis was done using "R". Among identified proteins,
those with at least 70% of values in at least one condition were selected.
Then, proteins with t-test *P* value < 0.05 were defined as significantly
differentially expressed, and proteins with 100% of valid values in one
condition and 0% of valid values in the other condition were defined as
"Appeared" or "Disappeared". Log2(LFQ intensity) matrix were filtered
before imputation. For heatmaps, Z-score were calculated after
imputation.

Functional analyses were generated through Ingenuity Pathway
Analysis (QIAGEN Inc., https://www.qiagenbioinformatics.com/
products/ingenuitypathway-analysis) version 76765 844 for each list
of differential proteins. Significantly over-represented biological terms
(Canonical pathways or Diseases and functions) were identified with a
right-tailed Fisher's Exact test that calculates an overlap *P* value
determining the probability that each term associated with our lists of
differential proteins was due to chance alone. The z-score is a statistical
measure of correlation between relationship direction and experi-
mental protein expression. Its calculation assessed the activation
(positive z-score) or repression (negative one) of each term. To be
considered significant the z-score has to be greater than 2 in absolute
value. Overlap of selected pathways and functions was designed using
Cytoscape version 3.9.1.

## DRIP-qPCR and DRIP-sequencing

DRIP-qPCR or DRIP-seq was performed on human CD34-derived proE/
basoE from 6 MDS with *SF3B1* mutation, 6 MDS without *SF3B1* muta-
tion and 4 healthy controls[79]. Briefly, $8 \times 10^6$ cells were harvested and
processed for genomic DNA extraction by SDS/ Proteinase K treatment
at 37 °C followed by phenol-chloroform extraction and ethanol pre-
cipitation. 8 μg of genomic DNA was fragmented using HindIII, EcoRI,
BsrGI, XbaI, and SspI. Then DNA pre-treated or not with 16 UI/mL
RNase H1 (New Engl Biolabs, Ipswitch, MA) overnight was immuno-
precipitated using S9.6 antibody at 40 μg/mL (Kerafast, Shirley, MA or
ATCC mouse hybridoma) overnight at 4 °C (Supplementary Meth-
ods 1). Quality control of immunoprecipitation was performed by
qPCR, at R-loop-positive loci (*RPL13A*, *CALM3*, *TFPT*) and R-loop-
negative loci (*EGR1*, *SNRP1*). DRIP-qPCR was also performed at specific
loci (*ABCC5*, *IREB2*, *TCIRG1*, *TMX2*) (Supplementary Method 2).

Percentage of input expected between 1-15%

$$\%\text{input} = 100 \times 2^{(\text{Ct input corrected} - \text{Ct DRIPed DNA})} \text{ where Ct (cycle threshold) input (corrected)} = (\text{Ct input} - \log 2(10))$$

and fold enrichment (expected between 20-300)

$$\text{Fold enrichment} = [2^{(\text{Ct input (positive locus corrected)} - \text{Ct DRIPed (DNA positive locus)})}]/$$
$$[2^{(\text{Ct input (negative locus corrected)} - \text{Ct DRIPed DNA (negative locus)})}]$$

were calculated to assess the immunoprecipitation efficiency and
specificity, respectively. For DRIP-seq, good quality DRIPed DNA
samples (with and without RNase H1-treatment) were sonicated to get
an average length of 200-300 bp. Then samples were subjected to end-
repair, dATP tailing, adaptor ligation and library indexing. Lastly, the
libraries were cleaned up using AMPure beads, and amplified. After
checking library quality on Agilent Bioanalyzer using Agilent High
Sensitivity DNA 1000 kit, libraries were sequenced on Illumina Nova-
Seq instruments.

## DRIP-seq bioinformatic analysis

100 bp paired-end reads were trimmed and filtered using fastp
(v0.23.4) to remove low quality reads, low complexity sequences, as
well as polyG tails (corresponding to no signal in the Illumina two-color
systems, in NovaSeq data). Reads were mapped to the human refer-
ence genome (GENCODE, GRCh37; https://www.gencodegenes.org/
human/release_19.html) with Bowtie2 (v2.3.5.1), using parameters – no-
discordant and – no-mixed. Resulting SAM files were piped through
Samblaster (v0.1.24) to remove duplicate reads, then through Sam-
tools (v1.10) to generate sorted (by coordinates) BAM files, with a cut-
off for MAPQ score of 10. Peaks were called for each replicate with the
MACS algorithm (MACS3) in broad mode, with the input DNA as
control as well as the same sample treated with RNase H1, at *q*
value < 0.1. Resulting peaks were analysed by groups of biological
replicates with MSPC (v5.5.0)[80]. Firstly, by categorizing them as either
background, weak, or stringent (with both cut-off on *P* value at 1e-4
and 1e-8). Weak peaks were rescued if stringent in other biological
replicates. Peaks were then confirmed or discarded based on the
combined stringency test supported by enough replicates and if their
combined stringency, using Fisher's combined probability test, satis-
fies the threshold of 1e-8. Confirmed peaks are qualified as true posi-
tive if they pass the Benjamini-Hochberg (BH) multiple testing
correction at level 0.05.

To identify differentially expressed R-loops, we intersected the
peaks from MACS3 calling with the 5.5 million of restriction fragments
generated before S9.6 immunoprecipitation, using the resulting
matrix as a reference frame for featureCounts (v2.0.0). Normalization
and differential R-loop expression analysis between wild-type, mutant
and control samples was performed with DESeq2, using BH-adjusted *P*
value < 0.05 and log2 (FC) > |1|. Bedtools (v2.27.1) intersect was used
for overlap analyses.

## May-Grunwald Giemsa staining and flow cytometry

Erythroid differentiation of human primary erythroblasts and murine
G1E-ER4 clones was followed by May-Grünwald Giemsa staining of
cytospins. Cell viability was assessed by the scatter profile (FSC/SSC)
using flow cytometry. For erythroid differentiation, $5 \times 10^4$ cells are
washed in 1X phosphate buffered saline (PBS) supplemented with 2%
FCS and incubated for 20 min at 4 °C with fluorescent antibodies to

GPA (CD235a), CD49d and CD71 for human primary cells (Beckman Coulter, Brea, CA) or Kit (CD117) and Ter-119 for murine cells (BD Biosciences, Franklin Lane, NJ). Analysis was performed on LSRFortessa apparatus (BD Biosciences) with Kaluza software (Beckman Coulter) (Supplementary Methods 3).

## Cell cycle analysis and live cell imaging

Cell cycle was analysed by double labelling using a fluorescent anti-BrdU antibody and 7-aminoactinomycine D (7-AAD, BD Biosciences). Cells were incubated with 10 μM BrdU for 30 min at 37 °C (BD Biosciences). Then cells were pelleted, fixed in 500 μL 70% ethanol for 20 min at room temperature and washed in 1X PBS supplemented with 0.5% bovine serum albumin (BSA). Cell pellet is resuspended in 2 M chlorhydric acid for 20 min at room temperature and washed in 1X PBS. Acid was neutralized by a 0.1 M borate solution pH 8.5 for 2 min (Borax $Na_2B_4O_7$, Merck). Cells were washed 3 times and transferred to 96-well plates for incubation with 10 μL anti-BrdU antibody or isotype control for 20 min at room temperature either FITC-anti-BrdU (BD Biosciences, clone 3D4) or APC-anti-BrdU (BD Biosciences, clone 3D4) at a final concentration of 0.5 μg/mL. Cells were washed 3 times and incubated with 50 μL of a 10 μg/mL 7-AAD and 50 μg/mL RNase A (Macherey-Nagel, Hœrdt, France) solution for 30 min at room temperature. Finally, cells were transferred into Eppendorf vials containing 100 μL 1X PBS and analysed on BD Accuri C6 flow cytometer using CFlow Plus software (BD Biosciences) (Supplementary Method 4). The proliferation capacities of G1E-ER4 cells treated or not with β-estradiol at $10^{-7}$M (Merck) were measured using IncuCyte live-cell imaging system (Essen Instruments, Ann Arbor, MI). 48-well plates were coated with 0.01% Poly-L-lysine sterile-filtered solution (Merck). Cells were seeded at 30,000 cells/well and monitored over time with 9 images per well every 3 hours for 72 h. Results were expressed as cell confluence (in % of occupied space).

## DNA fiber combing

DNA replication was analyzed by pulse labelling with fluorescent thymidine analogs of human erythroblasts. Cells were first labelled with 20 μM iododesoxyuridine (IdU) for 30 min at 37 °C and then with 100 μM chlorodesoxyuridine (CldU) for 30 min at 37 °C. DNA replication was blocked by the addition of an excess of thymidine (300 μM) on ice and cells were washed in 1X PBS and counted. DNA combing was performed after DNA extraction in 1% low melting agarose plugs (0.3 × $10^6$ cells in 90 μL per plug). Briefly, low melting agarose was maintained at 45 °C. Cells were resuspended at 6.66 × $10^6$ in one volume of 1X PBS, mixed in the same volume of 2% agarose and distributed in plug mold. Then proteins in agarose plugs were digested with 1 mg/mL proteinase K in 0.25 M EDTA pH 8/ 1% SDS at 42 °C for 48 h. Finally, plugs were washed 3 times in 10 mM Tris-HCl pH 8/1 mM EDTA (TE), and agarose was eliminated by digestion using β-agarase for 48 h. For DNA fiber stretching, extracted DNA was resuspended in 0.25 M 2-(N-morpholino)-ethanosulfonic acid pH 5.5 in Eppendorf vials for 30 min at 65 °C, 30 min at room temperature and 2 weeks at 4 °C. Before combing, vials were placed at room temperature for 30 min. DNA was then placed in FiberComb reservoir (Genomic Vision, Bagneux, France) and stretched on silane treated coverslips. Coverslips were sticked on glass slides and incubated for 2 h at 60 °C and stored at -20 °C. For hybridization, DNA was denatured in a 1 N NaOH solution, rinsed in cold 1X PBS and dehydrated in 70%, 85% and 100% ethanol. Aspecific sites were blocked for 30 min at 37 °C. DNA was first incubated with primary anti-IdU (BD Biosciences, cat no. 347580, dilution: 1/50) mouse antibody and anti-CldU (Abcam, cat no. Ab 6326, dilution: 1/50) rat antibody, and then with fluorescent secondary antibodies. Finally, DNA was labelled with a primary anti-ssDNA antibody (mouse), and with a first fluorescent secondary anti-mouse antibody (goat), and with a second fluorescent secondary anti-goat antibody (donkey). Slides were mounted in VectaShield medium (Vector Laboratories, Burlingame, CA). All antibodies

and reagents are described in Supplementary Methods 5. Fiber length and symmetry of a minimum of 200 fibers per group were measured. Fork symmetry was expressed as the IdU/CldU ratio. The speed of the replication fork was calculated by the ratio $(d_I + d_{Cl}) / (t_I + t_{Cl})$, where $d_I$ and $t_I$ represent respectively the measured distance (in kb) and labelling time (in min) for IdU incorporation, and $d_{Cl}$ and $t_{Cl}$ denote the corresponding parameters for CldU incorporation.

## Immunofluorescence experiments

Immunofluorescence was used to detect phospho-RPA32 (p-RPA32, serine 33 and serine 4/8), phospho-H2AX (γ-H2AX, serine 139) and 53BP1 nuclear foci. Cells treated with 5 mM hydroxyurea for 3 h were used as positive controls (Merck). Briefly, cytospins were prepared with $10^5$ cells in 1X PBS, fixed in 1X PBS/2% paraformaldehyde (Santa Cruz Biotechnologies, Santa Cruz, CA) for 20 min at room temperature and washed. Cells were permeabilized in 1X PBS/0.5% Triton X-100 for 10 min and washed in cold 1X PBS. Saturation was performed using 1X PBS/3% BSA (Euromedex, Souffelweyersheim, France) for 30 min and cells were incubated with primary antibodies for 1 h at 37 °C (Supplementary Methods 6). After washing, cells were incubated with secondary antibody for 30 min at 37 °C in the dark. DNA was counterstained with 4′,6-diamidino-2-phenylindole (DAPI) for 5 min and cytospins were rinsed in 1X PBS and mounted with Fluoromount-G (Clinisciences, Nanterre, France). Analysis was conducted using an inverted DMI600 microscope at 100X magnification (Leica, Wetzlar, Germany). Images were analyzed using the ImageJ software (NIH, Bethesda, MD).

## Western blot analysis

Cell lysates were solubilized for 5 minutes at 95 °C in Laemmli buffer (65 mM Tris [pH 6.8], 20% glycerol, 5% β-mercaptoethanol, 0.01% bromophenol blue, and 2% sodium dodecyl sulfate [SDS]) Proteins were separated by SDS–polyacrylamide gel electrophoresis and transferred to a nitrocellulose membrane (VWR, Radnor, PA). Membranes were blocked with 5% of dry milk in TBS-T buffer (10 mM Tris-HCl, pH 7.5, 150 mM NaCl, 0.15% Tween 20) for 1 h and incubated in specific antibody overnight at 4 °C (Supplementary Methods 7). Membranes were washed in TBS-T buffer and incubated for 1 h at room temperature with secondary horseradish peroxidase (HRP)-linked antibody (horse anti-mouse HRP-linked antibody 7076 S or goat anti-rabbit HRP-linked antibody 7074 V, Cell Signaling, Danvers, MA). Enzyme activity was visualized by an ECL-based detection system (VWR, Radnor, PA). Blot imaging was performed on the Fujifilm LAS-3000 Imager (Fujifilm, Tokyo, Japan) and images were analysed using the Multi Gauge software (Fujifilm).

## Targeted LC-MS metabolomics analyses

$3.10^5$ $Sf3b1^{K700E/+}$ or $Sf3b1^{+/+}$ G1E-ER4 cells were collected after 0 h or 24 h of β-estradiol treatment and with or without 0.2 mM HU for 16 h (Merck). For metabolomics analysis, extraction was performed in 30 μL of 50% methanol, 30% acetonitrile (ACN) and 20% water. After centrifugation at 16,000 g for 15 min at 4 °C, supernatants were collected and stored at −80 °C until analysis. LC/MS analyses were conducted on a QExactive Plus Orbitrap mass spectrometer equipped with an Ion Max source and a HESI II probe coupled to a Dionex UltiMate 3000 uHPLC system (Thermo). Samples (5 μL) were injected onto a ZIC-pHILIC column with a guard column (Millipore) for LC separation in a gradient of buffer A (20 mM ammonium carbonate, 0.1% ammonium hydroxide pH 9.2), and buffer B (ACN) with a flow rate of 0.200 μL. $min^{-1}$ as follows: 0–20 min, linear gradient from 80% to 20% of buffer B; 20–20.5 min, linear gradient from 20% to 80% of buffer B; 20.5–28 min, 80% buffer B. The mass spectrometer was operated in full scan, polarity switching mode with the spray voltage set to 2.5 kV and the heated capillary held at 320 °C. The sheath gas flow was set to 20 units, the auxiliary gas flow to 5 units and the sweep gas flow to 0

units. The metabolites were detected across a mass range of 75–1000 m/z at a resolution of 35,000 (at 200 m/z) with the automatic gain control target at $10^6$ and the maximum injection time at 250 ms. Lock masses were used to ensure mass accuracy below 5 ppm. Data were acquired with Thermo Xcalibur software (Thermo Fisher Scientific). The peak areas of metabolites were determined using Thermo TraceFinder software (Thermo Fisher Scientific), identified by the exact mass of each singly charged ion and by the known retention time on the HPLC column. Each metabolite was quantified as the area under the curve and results were expressed as arbitrary unit (A.U).

### RT-PCR and PCR on colonies
For RT-PCR, RNA was extracted using RNAeasy Mini Kit (Qiagen) and retrotranscribed with the Maxima First Strand cDNA synthesis kit (Thermo Fischer Scientific). cDNA was amplified using Phire Hot Start II DNA polymerase (Thermo Fischer Scientific). Amplicons were analysed by electrophoresis on a 2% agarose gel. For fluorescent RT-PCR, the PCR was performed using the same primers except that the forward primers were labelled with 6-carboxyfluorescein (6-FAM) at 5' end. For capillary electrophoresis, 1 µL of diluted PCR product was added to 0.2 µL of GeneScan 500 ROX dye standard and 18 µL of RNAse-free water. After denaturation for 5 min at 95 °C, fragments were separated using the 3730xl DNA analyzer and analyses was performed using GeneMapper Software 5 (Thermo Fischer Scientific). For qPCR, the primers were used with the SYBRGreen Master Mix (Meridian Bioscience) in a LightCycler480 (Roche). Expression levels were normalized to actin (*ACT*) and cyclophilin A (*PPIA*) expression using geometric averaging, and analyzed using ΔΔCt method. Primers for RT-PCR, fluorescent RT-PCR and qPCR are listed in Supplementary Methods 8.

For the detection of mutants by PCR, colonies were picked and transferred in 50 µL of lysis buffer (KCl 50 mM, Tris HCl pH8 10 mM, 0.4%NP-40, 0.4% Tween-20, 0.2 mg/mL proteinase K). After 1h incubation at 55 °C, DNA was treated for 10 min at 95 °C. Samples (5 µL) was used for 40 cycles of PCR cycles (30 s at 95 °C, 30 s at 60 °C, 1 min at 72 °C) using specific primers (Supplementary Data 2). Amplicons were analysed by gel electrophoresis.

### Statistical analysis
For quantitative variables, values were expressed as median and interquartile range (IQR) or means and standard error of the mean (SEM) and compared using the Student t-test or non parametric Mann-Whitney or Kruskal-Wallis tests. Chi-squared or Fisher exact tests were used to compare categorical variables. For transcript quantification, the Mann-Whitney test was used to assign a statistical significance for each group comparison. $P$ values < 0.05 were considered significant (JMP version 10.0.2, SAS Institute Inc, Cary, NC).

### Reporting summary
Further information on research design is available in the Nature Portfolio Reporting Summary linked to this article.

## Data availability
The RNA-sequencing and DRIP-sequencing data generated in this study have been deposited in the NCBI's Gene Expression Omnibus (GEO) database. Accession codes are provided below. The processed RNA-sequencing and DRIP-sequencing data are available in the Supplementary information as Supplementary Data indicated for each of them. RNA-seq BM MNC cohort of 27 cases (Supplementary Data 1): GSE220525, RNA-seq of human basophilic erythroblasts and poly-chromatophilic erythroblasts (Supplementary Data 2): GSE220523. DRIP-seq of human basophilic erythroblasts (Supplementary Data 4): GSE220271. RNA-seq of mouse G1EER erythroblasts (Supplementary Data 5): GSE220516. RNA-seq data of the BM MNC cohort of 185 MDS patients (Supplementary Fig. 1b) are available at GSE220518 under

restricted access since these data are considered sensitive personal data according to the European Union General Data Protection Regulation (GDPR) and thus cannot be shared with third-parties without prior approval. Access can only be granted for research purposes. An application must be sent to michaela.fontenay@inserm.fr. The proteomic data are available on ProteomeXchange Consortium via the PRIDE partner repository: Human erythroblast proteome (Supplementary Data 3) and mouse erythroblast proteome (Supplementary Data 5): PXD038700. Source data are provided with this paper.

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

## Acknowledgements

The authors dedicate this work to the memory of their colleague Dr Angelos Constantinou whose discoveries inspired this work. The authors acknowledge Pr Seishi Ogawa (Kyoto University, Japan) for very helpful discussion and comments on the manuscript, Dr Evelyne Lauret for support and discussion, Institut Cochin, Paris; Dr Sarah Lambert (Institut Curie, Orsay), Dr Jean-Charles Cadoret (Institut Jacques Monod, Paris), Dr Valeria Naim (Institut Gustave Roussy, Villejuif), Pr Raphaël Itzykson (Institut de Recherche Saint-Louis, Paris) for discussion. The authors thank Dr Carole Almire, Laboratory of Hematology, Cochin Hospital, Dr Emilie-Fleur Gautier, Platform Proteom'IC, Institut Cochin, Dr Franck Letourneur, Platform Genom'IC, Institut Cochin, Ms Stella Hartono, Department of Molecular and Cellular Biology and Genome Center, University of California, Davis, Dr. Ivan Nemazanyy, Platform for Metabolic Analyses (INSERM US24/CNRS UAR 3633, Necker, Paris, France) for their expertise, Ms Katherine Wetmore, Ms Ania Alik, Mrs Angélique Marcon, Mrs Marlène Dejean, Ms Camille Knops, Ms Laïla Zaroili, Ms Alice Rousseau (Laboratory of Hematology, Cochin Hospital, Paris) for technical help. **Grants**: This work was funded by the Institut National du Cancer INCa PLBio 2015-129 (M.F., M-H. S., A. C.), the Fondation pour la Recherche Médicale Equipe labellisée FRM202003010191 (M.F.), the Laboratoire d'Excellence GR-Ex and the MDS-RIGHT European Union's Horizon 2020 research and innovation programme under grant 634789 (P.F., M.F.).

## Author contributions

T.R., S.Bo., A.L. and R.M.M. contributed equally to this study. M.F., M-H.S., A.C., F.C., B.M., B.P. designed the study. D.R., C.L., B.F., A.L., N.D., M.W., R.M. developed the methods. C.L., T.R., S.Bo., R.M.M., B.F., A.L-P, A.L., S.Ba., M.DC., D.C-G., F.L., A.T., C.F. performed the experiments. C.F., M.T., L.A., E.C., P.F., S.R., T.C., S.P., L.W., D.B., N.C., O.K. acquired and managed patient clinical and biological data. M.L., M.LG., N.D. provided facilities. M.W., S.A., R.M. provided tools. D.R., I.B., A.H. performed computational analyses. D.R., C.L., A.L., B.M., S.Bo., B.F., F.C., D.C-G., M.F. analyzed the data and performed biostatistical analyses. M.F., D.R. wrote the manuscript. All authors reviewed the manuscript.

## Competing interests

The authors declare no competing interests.

## Additional information

[1]Université Paris Cité, Centre National de la Recherche Scientifique, Institut National de la Santé et de la Recherche Médicale, Institut Cochin, Paris, France. [2]Equipe labellisée par la Fondation pour la Recherche Médicale, Paris, France. [3]Laboratoire d'excellence du Globule Rouge GR-Ex, Université Paris Cité, Paris, France. [4]Assistance Publique-Hôpitaux de Paris.Centre-Université Paris Cité, Hôpital Cochin, Laboratory of Hematology, Paris, France. [5]Université Paris Cité, CNRS, Institut Jacques Monod, Paris, France. [6]Department of Molecular and Cellular Biology and Genome Center, University of California, Davis, CA, USA. [7]Platform Proteom'IC, Université Paris Cité, Institut Cochin, Paris, France. [8]Institut Curie, PSL Research University, Sorbonne University, INSERM U830, DNA repair and uveal melanoma, Equipe labellisée par la Ligue Nationale contre le Cancer, Paris, France. [9]Assistance Publique-Hôpitaux de Paris.Centre-Université Paris Cité, Hôpital Cochin, Clinical Department of Hematology, Paris, France. [10]Department of Hematology, Centre Hospitalier Universitaire, Université de Grenoble Alpes, Grenoble, France. [11]Laboratory of Hematology, Université Côte d'Azur, Centre Hospitalier Universitaire, Nice, France. [12]Clinical Department of Hematology, Université Côte d'Azur, Centre Hospitalier Universitaire, Nice, France. [13]Assistance Publique-Hôpitaux de Paris.Nord-Université Paris Cité, Saint-Louis Hospital, Laboratory of Hematology, Paris, France. [14]Assistance Publique-Hôpitaux de Paris.Nord-Université Paris Cité, Saint-Louis Hospital, Service Hématologie Séniors, Paris, France. [15]Institut Curie, Paris Sciences Lettres Research University, Sorbonne University, INSERM U934, UMR3215 Paris, France. [16]Institut Gustave Roussy, INSERM 1287, Université Paris Saclay, Villejuif, France. [17]Institut de Génétique Humaine, Centre National de la Recherche Scientifique, Université de Montpellier, Montpellier, France. [18]These authors contributed equally: David Rombaut, Carine Lefèvre. [19]Deceased: Angelos Constantinou. ✉e-mail: michaela.fontenay@inserm.fr

