## [Peer Review File · Nature Communications]

Accelerated DNA replication fork speed due to Loss of R-Loops in Myelodysplastic Syndromes with SF3B1 MutationReviewers' Comments:

Reviewer #1:

Remarks to the Author:

This study aims to decipher the molecular basis of myelodysplastic syndromes (MDS) associated with mutations in the SF3B1 splicing factor, which are characterized by ineffective erythropoiesis. The authors demonstrate that SF3B1-mutated erythroid precursors exhibit characteristic transcriptomic and proteomic signatures that correlate with reduced retention of introns found in SF3B1-mutated MDS. Moreover, they show that erythroblasts from SF3B1-mutated MDS patients display a dramatically reduced R-loop formation in gene bodies as compared to MDS erythroblast with wild-type SF3B1. Interestingly, SF3B1-mutated MDS erythroblasts were also found to exhibit elevated rate of replication fork progression and to accumulate Ser4/8 phosphorylated RPA2 but not other DNA damage markers such as γ -H2AX or 53BP1 foci. Finally, the authors show that histone deacetylase inhibition increased R-loop formation in murine erythroblasts carrying a Sf3b1-K700E mutation and improved erythroid cell differentiation. Overall, this manuscript contains a significant amount of interesting and high-quality data, however, some of the conclusions drawn by the authors from these data are questionable.

Specific comments:

1. Based on the data shown in Fig. 3, the authors conclude that SF3B1 mutation induces loss of R-loops in erythroid cells. To verify this hypothesis, the levels of R-loops in SF3B1-mutated MDS cells and the control cells derived from healthy individuals should be compared at a genome-wide scale. Could R-loop levels in the SF3B1-wild type MDS cells used in DRIP-Seq experiments be elevated due to mutations in other factors preventing R-loop formation? In fact, the DNA combing data presented Fig. 4E-G show an elevated frequency of fork stalling in these cells, which can be caused by R-loops. It should be also noted that the S9.6 dot blot data in Fig. 6H indicate that Sf3b1-K700E mutation does not significantly increase R-loop levels in murine erythroblasts.
2. The effect of HDAC inhibition on R-loop levels in the SF3B1-mutated MDS erythroblasts should be also tested by S9.6 dot blot. Data from three independent experiments must be shown. This also applies for the analysis conducted with murine cells (Fig. 6H, data from only one experiment are shown). Moreover, samples treated with RNase H should be also present on S9.6 dot blot to demonstrate that the signal detected corresponds to RNA:DNA hybrids.
3. The authors conclude that the elevated fork speed in SF3B1-mutated MDS erythroblasts is due to reduced R-loop levels. This is even stated in the title of the manuscript: "DNA replication stress due to loss of R-loops in Myelodysplastic Syndromes with SF3B1 mutation". However, such statement is not supported by the data presented in the manuscript. Can the authors design an experiment to test this hypothesis? If they confirm that HDAC inhibition with SAHA restores normal R-loop levels in SF3B1-mutant cells, they could then test by DNA combing whether SAHA can reduce fork velocity in these cells.
4. Fig. 4F shows that compared to control erythroblasts, SF3B1-mutated MDS erythroblasts do not exhibit an asymmetric fork progression (an indicative of fork stalling). In contrast, analysis of DNA replication patterns in Fig. 4G shows that SF3B1-mutated MDS erythroblasts exhibited an elevated frequency of fork stalling events (green tract only events) as compared to control erythroblasts. How do the authors explain this discrepancy?
5. Given the fact that SF3B1-mutated erythroblasts showed increased fork stalling compared to control erythroblasts, could it be that the accelerated replication fork velocity observed in these cells is not caused by loss of R-loops?

6. The observation of elevated levels of RPA2 phosphorylation at Ser4/8 (pRPA S4/8) in SF3B1-mutant cells led the authors to the conclusion that these cells accumulate ssDNA gaps. However, it should be noted that pRPA2 S4/8 is not a specific ssDNA marker. pRPA2 S4/8 is produced by DNA-PK in response to DNA double-strand breaks generated, for example, by replication fork collapse. Of note, it has been shown that accelerated replication fork progression upon PARP inhibition induces DNA damage and genomic instability (doi: 10.1038/s41586-018-0261-5). I recommend the authors reading this paper. To determine ssDNA levels in SF3B1-mutant cells, the author can perform immunofluorescence staining with an anti-ssDNA antibody or quantify RPA foci using an anti-RPA2 antibody. They can also analyze the levels of RPA2 phosphorylation at Ser33 which is mediated by ATR in response to ssDNA.

Minor comments:

1. In Methods, it should be described how replication fork velocity was calculated from the DNA combing data.
2. Legend for Fig. 4E: Replace "Histogram" with "Scatter plot".
3. Fig.3F: Data for samples treated with RNase H should be added.
4. What SF3B1 mutation is present in the SF3B1-mutant MDS cells used in this study?
5. In 2019 (Bondu et al., *Sci. Transl. Med.* 11), the authors published data from transcriptome analysis of the same set of cell lines as those used in this manuscript [BM MNCs isolated from 27 patients in this cohort, including 21 with SF3B1(MUT) MDS, 6 with SF3B1(WT) MDS]. Why did they repeat these experiments?
6. In several cases, the type of statistical test used is not described in figure legends (e.g. Fig. 3F, Fig.4B,C, Fig. 5D-E). Is Mann-Whitney test appropriate for statistical analysis of the data in Fig. 4I-K? There is only one data point for U2AF1(MUT) cells in these graphs.

Reviewer #2:

Remarks to the Author:

The manuscript by Rombaut et al., describes characterization of erythroid differentiation models of MDS and the effects of SF3B1 mutation in this setting. In contrast to previous literature the authors describe a reduction in R-loops associated with SF3B1 mutations using genome-wide analysis. These reduced R-loops are correlated with increased replication fork speed and evidence of DNA replication stress but not DNA damage. The authors conduct experiments to support the value of therapeutic R-loops restoration by vorinostat treatment to promote erythroid differentiation. This is an interesting manuscript as it synthesizes a number of unique observations into a novel perspective on SF3B1 and R-loops. Nevertheless, the authors could do more to make connections between their various analyses to strengthen their model. Below are my major suggestions for improvement.

Gene expression changes:

1. At several places in the manuscript the authors describe differential gene or protein expression experiments that link SF3B1 mutations to altered expression of factors involved in genome maintenance, DNA replication and repair. It seems like these factors are both up and down regulated. Are these changes likely to cause a coherent cell biological phenotype? Or are the functional enrichments random? Are there other features of these genes (aside from biological function) that would better explain the effects of SF3B1mut?
2. Can the authors connect the reported changes in protein expression for DNA damage checkpoint proteins (Figure 6D) with any of the phenotypes reported? Its unclear what functional links are being suggested between changed protein levels and replication stress phenotypes?
3. Specifically, the authors (on page 13) seem to suggest that low levels of FANCD2 could be responsible for the fast fork speed and RPA32 phosphorylation observed in the SF3B1 mutant cells. If so they should confirm that FANCD2 protein is significantly reduced by western blot or another

orthogonal assay, and then explain their model in light of the publications stating that FANCD2 depletion increases R-loops (not decreases). I was confused by the discussion linking low FANCD2 to the observed phenotypes since there are multiple independent reports starting in 2015 suggesting that low FANCD2 increases R-loops (the opposite of what the authors see with SF3B1mut).

Additional analyses, clarifications or experiments

1. The whole proteome analysis in Figure 2 is interesting but seems like more analysis could be done. For example, are specific types of splicing changes associated with proteome changes? Can mutant proteins be detected? If it simply reflects the RNA-seq it does not seem particularly useful - could additional analyses allow the authors to say more about the proteome changes?

2. Additional DNA combing data could give more insight to whether R-loops are really a major barrier to fork speed in erythroblasts. For example, treatment with transcription inhibitors or RNaseH should equalize forkspeed in the SF3B1WT and mut samples if the authors are correct. Fork speed can also be regulated by other signals, including changes to the machinery for fork reversal. Doing the appropriate controls with transcription inhibitors or RNaseH would help the authors better describe potential mechanisms.

3. Why did the authors monitor RPA32 s4/8, which is deposited by DNAPK, rather than ser33, which is deposited by ATR?

4. The increase in fork speed and RPA phosphorylation but lack of DNA damage is somewhat surprising. Have the authors considered enhancing fork speed with PARP inhibitors (i.e. as in PMID: 29950726)? If changes to PARP expression could partly explain the increase rate it could explain the mechanism.

5. The effects of SAHA on R-loops are not that well characterized. Do the authors think that expression of a subset of genes is the key mechanism behind the observed rescue of differentiation in SAHA treated cells? Is expression analysis available in this model to support rescue changes to specific decreased R-loop target genes? The use of R-loop restoration for therapy is a really innovative approach of the manuscript. Do the authors think that alternative R-loop modulators would also restore differentiation? Understanding the mechanisms of SAHA changes to gene expression, or the use of alternative R-loop inducers to rescue differentiation (e.g. sub-lethal doses of camptothecin or other agents), would help to understand this.

Need to address literature context:

1. The lack of DNA damage as marked by gH2AX or 53BP1 in SF3B1 mutant cells is surprising given the literature. The authors should attempt to explain this discrepancy. The authors seem to assign some importance to changes in gene expression of DNA repair/replication factors in Figure 1/2 but do not see increased DNA damage (as others have). Are there functional consequences to changes in genome stability gene expression in SF3B1mut or not? If so, what are the consequences?

2. The authors do not adequately address the previous findings of the Boulton and Savage labs (e.g. PMID: 32076118 and PMID: 35027467), which suggest that SF3B1 mutations actually increase R-loops as a cause of replication stress and drug sensitivity. I recognize the clear methodological advantages of DRIP-seq done here as a means to characterize R-loops, but given that there are multiple independent publications that oppose the findings here, the authors should directly address them and highlight advantages of this study.

Clarifications to the overall model:

1. I was a bit confused by the model for why R-loop accumulation is decreased in SF3B1-mut cells. The authors cite a correlation between increased intron excision and lower R-loops. Can the authors say more about WHY SF3B1mut would increase intron excision? Is splicing flux faster? This seems

inconsistent with previous literature focused on SF3B1 which found alternative 3'SS selection, skipped exons and other features of the transcriptome. A statement on Page 9, saying "this suggests that transcription initiation is less active in SF3B1mut cells." was additionally confusing. How would increased intron excision be linked to reduced transcription initiation? Would SF3B1 even be recruited to transcription initiation sites, or do the authors envision an indirect effect? The reduced R-loop observation is one of the most surprising findings of the manuscript and the authors should do more to put it in context, contrast with the literature, and try to explain the mechanism.

Reviewer #3:

Remarks to the Author:

In this manuscript Rombaut et al study the cause and impact of aberrant R-loop formation in SF3B1 mutant MDS.

Previous studies have identified increased R-loop formation in SRSF2 and U2AF1 mutant MDS that results in replication stress and sensitivity to ATR inhibitors.

In contrast to these finding, Rombaut et al show that SF3B1 mutations results in reduced R-loop formation in part due to increased splicing of normally detained/retained introns. This process specifically impairs erythropoietic differentiation that is in part driven by regulation of gene expression via intron retention and non-sense mediated decay.

The authors show in detail that SF3B1 mutations result in accelerated transcription as evident by increased replication fork speed, loss of R-loops with "exposure" of cells to single-stranded DNA and replication stress and compromised differentiation. They identify a signature related to HDACi induced signatures and determine that HDAC inhibitors slow transcription speed and restore R-loop formation and thereby erythropoietic differentiation.

The authors' conclusions are based on robust multi-omic analysis (RNAseq, DRIPseq, proteome, etc.) of primary SF3B1 mutant, wildtype and SRSF2 or U2AF1 mutant MDS erythroblasts, isogenic human cell lines and Sf3b1 mutant murine erythroblasts. The authors convincingly show decreased R-loop formation, enhanced replication fork speed, increased replication stress, and impaired differentiation of SF3B1 mutant cells. Interestingly, they identify a signature in the proteome that includes targets of clinically approved HDAC inhibitors and treatment of mutant cells indeed partially rescues erythroid differentiation.

Overall, the authors' findings are very well documented and mechanisms are analyzed in detail. The finding that HDAC inhibition reduces the aberrantly accelerated replication fork speed, increases R-loop formation and improves erythropoietic differentiation is of interest and highly relevant.

The rescue of differentiation and replication stress by R-loop induction via HDACi however, while a major novel point of this manuscript, is insufficiently proven. The murine cells may replicate some aspects of SF3B1 mutant MDS but the rescue effect of HDACi should be shown in human SF3B1 mutant human MDS erythroblast in contrast to SF3B1 WT and SF mutant human MDS erythroblasts. In addition, restoration of intron-retention - R-loop formation - gene expression should be shown at least for select key targets

Major Comments:

Figures 3B is the key point of the paper: loss of R-loops in SF3B1 mutant cells. However, the next most important point, loss of intron retention as a mechanism of loss of R-loops is not as well supported: In figure 3D only a minority of lost R-loops are explained by loss of retrained introns. 63% of the peaks lost are located in gene bodies, purportedly mainly in introns. But 95% of peaks lost in Fig 3D are not explained by RRI suggesting that the effect of lost R-loops is mainly driven by another mechanism. maybe add something in the discuss or limitations of the study.

What are "consensus" R-loops (Figure 3G). What are the R-loops reported here? Can a motif analysis

be provided?

Figure 6H: since increase in R-loop formation and the resulting rescue of erythroid differentiation is a major point, the authors should provide additional data to confirm this point.

Such experiments should include S9.6 immunofluorescence, DRIP-PCR of targets whose expression (up or down) is not restored, etc.. RNAseq and DRIPseq of HDACi treated samples could provide such confirmation. Also, while the murine cells may replicate some aspects of human disease and human cells, the fact remains that Sf3b1 mutant mice do not develop MDS and remain a poor model for SF3B1 mutant MDS. The authors should validate the effectiveness of HDACi in human SF3B1 mutant MDS.

Minor Comments:

The highlight #1 should be reconsidered: it seems that reduction of retained introns coincides with loss of R-loops. Figure 3D suggests that the reduction of retained introns is mainly explained by loss of R-loops but not vice versa. 95% of R-loops lost are not explained by reduction of retained introns. How do the authors reconcile this aspect?

The authors state that in contrast to SF3B1 mutations that cause replication stress by accelerated replication fork speed and exposure to ssDNA that in turn is ameliorated by HDAC inhibitors. They should better explain how this is distinct from the erythroid differentiation defect shown in U2AF1 mutant MDS. Is the HDAC effect not "simply" due to rescued intron retention rather than specifically via R-loop formation? U2AF1-mut increases R-loops (Chen Mol Cell 2018) and impairs erythroid differentiation (Yip JCI 2017). This should be further discussed.

Figure 1B – green/red should be avoided to accommodate individuals with color blindness. The highlight in green of "terms of interest" seems arbitrary/serving the purpose of the paper; it could be better achieved by highlight text in bold.

In Supp Fig 1A the authors mention 189 MDS samples analyzed by RNA-seq. In the methods the total number of samples is 100, and in the paragraph describing figures 1 and 1S the authors describe 21+6 samples. Please reconcile and provide data sets or references in a table.

Add stats to Figure 2B.

The authors should generate a supp figure where they split the GO analysis for up- and down-regulated genes. For example, considering their result in Figure 4, one would expect to see enrichment in DNA replication for the upregulated genes.

Genes named in Figure 2I are highly selected. There seem to be many more significantly changed genes that should be spelled out to provide more unbiased representation that can still highly DNA repair.

For Figure 3C the authors could show distribution of R-loops relative to exons/introns

In Figure 3F the SF3B1 WT samples with wide variability between samples include a U2AF1 mutant sample. This seems odd as U2AF1 mutations aberrantly induce R-loop formation. The U2AF1 mutant sample should be identified and if it significantly contributes to the differences between SF3B1 mut and wt it should be replaced by a SF WT sample.

Figure 3I: how do the authors explain the decreased expression in SF3B1 mutant cells in those genes that have R-loops at the TSS in SF3B1 WT cells? The last sentence of the paragraph should be explained and justified mechanistically (Together, this suggests that transcription initiation is less active in SF3B1MUT cells.).

Figure 4: sample numbers in main text, figure legend and figure do not correspond?

Figure 4 I,J,K: the bar for the U2AF1 mutant sample should be removed. It is a single point.

In Supple Fig. 6 was a WT donor template not used for WT cells? Show sequencing validation of heterozygous K700E mutations.

Figure S8 suggests regulation of targets in the same direction (up/down) in Sf3b1 mut and by the HDAC inhibitor vorinostat. If HDACi restore R-loops one would expect opposite regulation of the described key targets.

REVIEWER COMMENTS

Reviewer #1 (Remarks to the Author):

This study aims to decipher the molecular basis of myelodysplastic syndromes (MDS) associated with mutations in the SF3B1 splicing factor, which are characterized by ineffective erythropoiesis. The authors demonstrate that SF3B1-mutated erythroid precursors exhibit characteristic transcriptomic and proteomic signatures that correlate with reduced retention of introns found in SF3B1-mutated MDS. Moreover, they show that erythroblasts from SF3B1-mutated MDS patients display a dramatically reduced R-loop formation in gene bodies as compared to MDS erythroblast with wild-type SF3B1. Interestingly, SF3B1-mutated MDS erythroblasts were also found to exhibit elevated rate of replication fork progression and to accumulate Ser4/8 phosphorylated RPA2 but not other DNA damage markers such as gamma-H2AX or 53BP1 foci. Finally, the authors show that histone deacetylase inhibition increased R-loop formation in murine erythroblasts carrying a Sf3b1-K700E mutation and improved erythroid cell differentiation. Overall, this manuscript contains a significant amount of interesting and high-quality data, however, some of the conclusions drawn by the authors from these data are questionable.

Specific comments:

Answer: We do thank the reviewer for his/her sound evaluation of our work.

1. Based on the data shown in Fig. 3, the authors conclude that SF3B1 mutation induces loss of R-loops in erythroid cells.

1. a - To verify this hypothesis, the levels of R-loops in SF3B1-mutated MDS cells and the control cells derived from healthy individuals should be compared at a genome-wide scale.

Answer: To verify the level of R-loops genome-wide, DRIP-seq experiments on human CD34+-deriving erythroblasts at ProE/basophilic stage were completed. We have included a total of 15 patients and healthy controls for DRIP-seq experiments. We used RNase H1 (RNH1) pretreatment to remove R-loops before immunoprecipitation with S9.6 antibody, for each sample. Including the inputs sequencing (once per patient) and HDACi treatment for 10 samples, we obtained 65 DRIP-seq that we included in a unique bioinformatic analysis.

The current dataset comprised

- 5 samples with *SF3B1* mutation: 3 cases with *SF3B1* K700E, one case with *SF3B1* H662C mutation and one case with *SF3B1* K666R mutation
- 4 healthy controls
- 6 MDS *SF3B1*^{WT} with distinct genotypes including 3 cases with a pattern of mutations usually characteristic of patients with diseases that could evolve to chronic myelomonocytic leukemia (CMML), denoted CMML-like in the table below.

	WHO	mutations
1	MDS-LB	DNMT3A P904Q (33%)
2	MDS-LB	TET2 splice 35%, TP53 p.Phe134Cys 4%
3	MDS-LB	No mutation (NGS panel 40 genes)
4	CMML-like	SRSF2 49%, TET2 5%
5	CMML-like	TET2 Lys1692Trpfs 94%, NRAS Gly12Asp 39%
6	CMML-like	SRSF2 P95H 38%, SETBP1 4%, TET2 p.Thr1399fs 31%, TET2 p.Arg1359Cys 28%, CUX1 5%, JAK2 2%

Quality control: Before sequencing, DRIP-qPCR were performed in order to assess the presence of R-loops at positive loci like *RPL13A*, *CALM3* and *TFPT* and the absence of R-loops at negative loci like *EGR1* or *SNRPN*.

We considered the peaks shared by all the samples of a given subgroup (MDS *SF3BI*^{MUT}, MDS *SF3BI*^{WT} or controls), further named **shared** peaks instead of consensus peaks in the initial version of the manuscript.

The comparison of the numbers of shared peaks between *SF3BI*^{MUT} samples and controls demonstrated that *SF3BI*^{MUT} samples had significantly less R-loops (median 197, range:44-216) compared to controls (median: 1650; range: 214-2209) confirming that erythroid cells with a *SF3BI* mutation do have less R-loops compared to healthy controls (Unpaired t-test; P-val=0.011).

The comparison of the number of shared peaks between *SF3BI*^{MUT} and *SF3BI*^{WT} samples also demonstrated less R-loops in *SF3BI*^{MUT} compared to *SF3BI*^{WT} (median: 2040; range: 1528-6618; P=0.015) (Figure 1).

These results are shown in Figure 3A of the manuscript and in the text.

Page 7: “The number of shared peaks in *SF3BI*^{MUT} erythroblasts was significantly decreased compared to controls or to *SF3BI*^{WT}”

Figure 1 (3A in ms): Violin plots showing the variations of the number of shared peaks (shared by samples of the same group) obtained by S9/6 immunoprecipitation of the genomic DNA. Unpaired t-test for P-values. * P<0.05.

1. b - Could R-loop levels in the SF3B1-wild type MDS cells used in DRIP-Seq experiments be elevated due to mutations in other factors preventing R-loop formation? In fact, the DNA combing data presented Fig. 4E-G show an elevated frequency of fork stalling in these cells, which can be caused by R-loops.

Answer: We agree with the reviewer that the genotype can influence the formation of R-loops. In addition to splicing factors *SRSF2*, *U2AF1* (Chen et al, PMID 29395063), several publications reported the increase of R-loop formation in the context of *DDX41*, *NRAS* or *HRAS* deletion/mutations (PMID: 34916496, 32913330, 27725641). Of note, the contribution of *HRAS* remains controversial depending on cell type and state as H-RAS mutated in pre-senescent cells was demonstrated to induce topoisomerase TOP1 overexpression and R-loop resolution (PMID: 36170822). Interestingly, the group of Dr A Rao reported that the deletion of *Tet2* and *Tet3* in murine mature B cells increased the formation of R-loops (PMID: 34937926).

In this study, the sequencing of putative regulators (*NRAS*, *KRAS*, *DDX41*, *RAD21*) of R-loops was performed. Only two patients with splicing factor mutation *U2AF1* or *SF3B1* had one mutation in *DDX41* or *RAD21* gene, respectively. None of these samples was available for combing or DRIP-seq experiments.

Due to the limited availability of fresh human samples, the difficulties inherent to the expansion of erythroblasts and to the realization of DRIP-seq experiments, we were not able to explore a broad range of distinct genotypes. However, to get further insights into the heterogeneity of *SF3B1*^{WT} samples, we analyzed *SRSF2*-mutated samples and samples with mutations with epigenetic factors *TET2* or *DNMT3A*. Looking at shared R-loops within this group, the highest numbers were observed in samples with epigenetic mutations.

1. c - It should be also noted that the S9.6 dot blot data in Fig. 6H indicate that Sf3b1-K700E mutation does not significantly increase R-loop levels in murine erythroblasts.

Answer: According to the results we obtained by DRIP-seq in human primary erythroblasts, the amount of R-loops was expected to be lower when *Sf3b1* gene is mutated in G1E-ER4 murine proerythroblastic clones compared to the isogenic wild-type clones. As underlined by this reviewer, the dot blot experiments shown in the initial version of the manuscript did not confirm this expectation.

These experiments using S9.6 antibody hybridization on the genomic DNA dotted onto a nitrocellulose membrane are thought to reveal

- R-loops formed during RNA polymerase I (at rDNA repeating unit), RNA polymerase III (tRNA) and RNA polymerase II-dependent transcription,
- R-loops at centromeres known to regulate mitosis
- TERRA R-loops at chromosome ends or telomeres
- and R-loops at mitochondrial DNA (mtDNA)

Examples from our DRIP-seq experiments show that R-loops at mtDNA or rDNA repeating unit are high, very stable at least partially resistant to RNH1 treatment compared to RNA pol II –dependent gene CALM3, which R-loops are disrupted by RNH1 pre-treatment (Figure 2).

We thought to remove R-loops at rDNA by inhibiting RNA pol I transcription using specific inhibitors such as CX-5461. We already used this inhibitor in primary erythroblasts to demonstrate that very active ribosome biogenesis collapses at the boundary between basophilic and polychromatophilic erythroblasts allowing final erythroid differentiation (Le Goff S, Boussaid I et al, Blood 2020). But considering that other constitutive R-loops at centromeres, telomeres or mitochondrial DNA would remain in place, this treatment would be insufficient to clear the picture and to allow seeing differences in R-loops related to RNA pol II-dependent transcription between samples and according to their mutational status. Therefore, we believe that S9.6 dot blot is a technique not contributive to our study and we decided to remove these experiments from the revised manuscript.

Figure 2: Visualization of R-loops in the Genome Browser IGV. Panels on the right show the peaks after pre-treatment with RNH1. Upper panel: Mitochondrial DNA (peaks reaching a maximum scale of 957 w/o RNH1 and 310 with RNH1), Middle panel: Ribosomal DNA (peaks reaching a maximum scale of 1354 w/o RNH1 and 1542 with RNH1). Bottom panel: RNAPII-dependent gene CALM3 (peaks reach a scale of 8 w/o RNH1 and around 3 with RNH1).

2. The effect of HDAC inhibition on R-loop levels in the SF3B1-mutated MDS erythroblasts should be also tested by S9.6 dot blot. Data from three independent experiments must be shown.

- This also applies for the analysis conducted with murine cells (Fig. 6H, data from only one experiment are shown).

Answer: We have performed S9.6 dot blot in human samples that are presented below (Figure 3). For the reasons given in our answer to question 1-c., we did not wish to extend further S9.6 dot blot experiments in human or murine samples, but in place we provided DRIP-qPCR at specific loci and DRIP-seq experiments in human samples in which the effect of HDAC inhibition on R-loop levels was tested.

- Moreover, samples treated with RNase H should be also present on S9.6 dot blot to demonstrate that the signal detected corresponds to RNA:DNA hybrids.

Answer: The dot blot experiments in murine G1E-ER4 Cripsr-Cas9 clones were performed with appropriate controls such as (i) RNH1 pretreatment that clearly diminished the S9.6 signal and (ii) antibody to dsDNA as a dot blot loading control (Figure 3). These results together with the 3 experiments performed using patient samples (Figure 4) are shown to the reviewer but not included in the revised manuscript.

Figure 3: Results of the S9.6 dot blot experiment #1 performed in G1E-ER4 clones including the RNase H1 (RNH) pretreatment as a control of S9.6 antibody specificity (A) and dsDNA antibody dot blot as a loading control (B) used to normalize the quantification (C).

Figure 3 (cont'd): Results of the S9.6 dot blot experiment #2 performed in G1E-ER4 clones including the RNase H1 (RNH) pretreatment as a control of S9.6 antibody specificity (A) and dsDNA antibody dot blot as a loading control (B) used to normalize the quantification (C).

Figure 4: Three dot blot experiments of MDS patient and control DNA samples. The RNase H1 (RNH) pretreatment as a control of S9.6 antibody specificity and dsDNA antibody dot blot as a loading control used to normalize the quantification (right panel) are shown. RNH efficiently disrupt R-loops, but note that normalized S9.6 signals were heterogeneous between samples of a given group.

3-The authors conclude that the elevated fork speed in *SF3B1*-mutated MDS erythroblasts is due to reduced R-loop levels. This is even stated in the title of the manuscript: “DNA replication stress due to loss of R-loops in Myelodysplastic Syndromes with *SF3B1* mutation”. However, such statement is not supported by the data presented in the manuscript.

- Can the authors design an experiment to test this hypothesis?

Answer: We thank the reviewer for this important question.

Numerous publications have shown that the fork velocity slows down when the fork encounters obstacles such as R-loops, leading eventually to fork collapse and double strand breaks, a source of genome instability.

Conversely, the replication fork speed can be accelerated following overexpression of oncogenes (PMID: 20660370), downregulation of mRNA biogenesis (PMID: 24896180), or PARP1 inhibition (PMID: 19103807; 29950726). Moreover, it has been shown that the activation of oncogenic HRAS in pre-senescent cells induces the acceleration of replication forks, and DNA damage revealed by γ H2X and 53BP1 foci. This phenotype is associated with an increased expression of topoisomerase 1 (TOP1) which is known to resolve unwanted R-loops and to participate in the maintenance of genome stability (PMID: 36170822).

Senataxin (SETX) and THO are other R-loop modulators. Of note, SETX and several THOC proteins were significantly upregulated in our *SF3B1*-mutated samples, while TOP1 was heavily expressed even if it was not differentially expressed between *SF3B1*^{MUT} and *SF3B1*^{WT} erythroblasts. Therefore, it could be interesting in the future to investigate the relative contribution of these different actors, including TOP1, SETX and THO complex to the regulation of R-loop formation in *SF3B1*^{MUT} and *SF3B1*^{WT} cells.

To address the comment, we took advantage of the role for histone deacetylase SIN3A, a protein interacting with the THO complex, in R-loop resolution and of the effect of HDAC inhibitors trichostatinA or SAHA/vorinostat in preventing R-loop resolution (PMID: 29074626). Vorinostat has already been used to prevent the leukemic progression of patients with high risk MDS with limited side effects. In our study, we report that IRR-transcripts deregulated in *SF3B1*^{MUT} erythroblasts were over-represented in the positive regulation of histone deacetylation (Fig. 2J).

For the revised version, we now provide experiments in which R-loops are modulated in order to assess the consequences on replication stress.

To further support our interpretation that accelerated fork speed is linked with loss of R-loops, we treated primary erythroblasts with vorinostat. First, we establish the concentrations of vorinostat maintaining the primary cells viable and we incubated them for 20h with 0.5 μ M of vorinostat (already used elsewhere in human primary erythroblasts to induce γ -globin gene by Dr D Higgs' team (PMID: 31406232). Then, treated and untreated *SF3B1*^{MUT}, *SF3B1*^{WT} and healthy control erythroblasts were harvested for DRIP-seq and DNA combing.

We obtained DRIP-seq data with or w/o HDACi in

- 4 healthy controls
- 3 MDS with *SF3B1* mutation (1 *SF3B1* K700E, 1 *SF3B1* H662C, 1 *SF3B1* K666R)
- 3 MDS *SF3B1* wildtype (1 *SRSF2*, 1 *SRSF2*/*biTET2*, 1 *NRAS*/*biTET2*)

HDACi increased R-loop quantities in 2/3 *SF3B1*^{MUT} samples, and in controls. In the 2 responding *SF3B1*^{MUT} patients, the level of R-loops reached the level of control samples. By contrast, HDACi systematically decreased the quantity of R-loops in *SF3B1*^{WT} (Figure 5).

Figure 5: Violin plots of the numbers of shared peaks obtained by DRIP-seq of 3 *SF3B1*^{MUT}, 3 *SF3B1*^{WT} (1 *SRSF2*, 1 *SRSF2* *biTET2*, 1 *biTET2*/*NRAS*) and 3 controls. Medians (quartiles) are indicated.

- If they confirm that HDAC inhibition with SAHA restores normal R-loop levels in *SF3B1*-mutant cells, they could then test by DNA combing whether SAHA can reduce fork velocity in these cells.

Answer: As we demonstrated that HDACi can restore normal R-loop levels in *SF3B1*^{MUT} cells, BrdU labelling and DNA combing experiments were performed to test the effect of HDACi on DNA synthesis and fork velocity.

BrdU was performed in 6 MDS patients with *SF3B1* mutation and 7 MDS patients without *SF3B1* mutation and 4 healthy controls. The results show that *SF3B1*^{MUT} samples exhibited an increase percentage of S-phase cells compared to *SF3B1*^{WT} samples. HDACi specifically and significantly decreased the percentage of S-phase cells in *SF3B1*^{MUT} samples, but neither in *SF3B1*^{WT} MDS nor controls.

By contrast, the intensity of BrdU in S-phase was not affected. This suggests that HDACi reduced the accumulation of S-phase cells. As shown previously by Dr Y Pommier (PMID: 20460513), vorinostat did not reduce dNTP incorporation or alter thymidylate synthase or ribonucleotide reductase function. Overall, HDACi did not abrogate cell cycle progression.

To investigate the impact on fork velocity, we performed DNA combing on three samples of proerythroblasts incubated or not with 0.5 μ M HDACi (vorinostat) for 20h including 1 age-matched healthy control, 1 MDS with *SF3B1/DNMT3A* mutations also analyzed concomitantly by DRIP-seq, and 1 MDS with *SF3B1/TET2/EP300* mutations.

As shown in the **figure 6**, we confirmed that fork speed was significantly increased in MDS with *SF3B1* and *DNMT3A* mutations and back to normal rate with pretreatment by vorinostat. By contrast, the second case of MDS with *SF3B1/TET2* and histone acetyltransferase *EP300* mutation had a normal velocity of replication fork insensitive to HDACi as well as the control. Whether *EP300* mutation is involved in the resistance to vorinostat remains to be determined.

Consistent with our hypothesis, we found that the quantity of R-loops in the *SF3B1/DNMT3A*-mutated sample was lower than in the control, and increased upon vorinostat treatment.

Altogether these data indicate that the modulation of R-loop landscape may impact on the S-phase of cell cycle and replication fork velocity. Conversely, the lack of R-loops facilitates fork progression and accelerates its velocity.

Figure 6: DNA combing. Scatter plots showing fork speed (as means +/- SD). Mann-Whitney for *P*-value.

4. Fig. 4F shows that compared to control erythroblasts, *SF3B1*-mutated MDS erythroblasts do not exhibit an asymmetric fork progression (an indicative of fork stalling).

In contrast, analysis of DNA replication patterns in Fig. 4G shows that *SF3B1*-mutated MDS erythroblasts exhibited an elevated frequency of fork stalling events (green tract only events) as compared to control erythroblasts

How do the authors explain this discrepancy?

Answer: We thank the reviewer for this comment. Our understanding of the pattern of replication shown in Figure 4F of the initial version was that green tracts may represent events of termination occurring during IdU pulse and not necessarily fork stalling. A green track flanked of single stranded DNA fibers is a mark of first label termination.

We understand from the reviewer's comment that our analysis can be interpreted in different ways. We thought it best to delete it and keep only the fiber data. The interpretation of the latter is not open to discussion and shows an increase in fork velocity without fork asymmetry in *SF3B1*^{MUT} cells.

5. Given the fact that SF3B1-mutated erythroblasts showed increased fork stalling compared to control erythroblasts, could it be that the accelerated replication fork velocity observed in these cells is not caused by loss of R-loops?

Answer: We observed that accelerated replication fork velocity was associated with preserved fork symmetry in *SF3B1*^{MUT} samples compared to controls suggesting that the progression of divergent forks did not encounter excessive obstacles. We cannot exclude that the loss of R-loops is not the sole cause of accelerated fork velocity in *SF3B1*^{MUT} cells. However, we observed a strong correlation between fork velocity and R-loop signals, and a diminution of fork velocity when R-loops were restored by HDACi treatment, we propose to comment this result in the discussion page 16: “accelerated fork velocity was observed when R-loops are lost”, and further discuss alternative causes to accelerated fork velocity.

“Alternative causes of increased fork velocity are related to overexpression of oncogenes (PMID: 20660370), downregulation of mRNA biogenesis (PMID: 24896180), or PARP1 inhibition (PMID: 19103807; 29950726). Moreover, activation of oncogenic HRAS in pre-senescent cells induces fork acceleration by inducing overexpression of topoisomerase 1 (TOP1), which is known to resolve unwanted R-loops (PMID: 36170822). TOP1 was heavily expressed in *SF3B1*^{MUT} and *SF3B1*^{WT} erythroblasts, while senataxin and THOC proteins, other R-loop modulators (PMID: 24746923) were specifically upregulated in *SF3B1*^{MUT} erythroblasts and could contribute to fork velocity. “

6. The observation of elevated levels of RPA2 phosphorylation at Ser4/8 (pRPA S4/8) in SF3B1-mutant cells led the authors to the conclusion that these cells accumulate ssDNA gaps. However, it should be noted that pRPA2 S4/8 is not a specific ssDNA marker. pRPA2 S4/8 is produced by DNA-PK in response to DNA double-strand breaks generated, for example, by replication fork collapse.

Of note, it has been shown that accelerated replication fork progression upon PARP inhibition induces DNA damage and genomic instability (doi: 10.1038/s41586-018-0261-5). I recommend the authors reading this paper.

To determine ssDNA levels in SF3B1-mutant cells, the author can perform immunofluorescence staining with an anti-ssDNA antibody or quantify RPA foci using an anti-

RPA2 antibody. They can also analyze the levels of RPA2 phosphorylation at Ser33 which is mediated by ATR in response to ssDNA.

Answer:

We do thank this reviewer for his/her very helpful comments. We were aware of the remarkable paper from Dr Jiri Bartek's team showing that inhibition of PARP led to increased fork speed and proposing a role for PARP/p21/p53 pathway in the regulation of fork speed (PMID: 29950726).

Looking at our transcriptomic data in *SF3BI*^{MUT} basophilic erythroblasts (suppl table 2), we observed that, the expression of CDKN1A/p21 transcript was significantly increased compared to *SF3BI*^{WT} erythroblasts:

ENSG00000124762	CDKN1A	Log2FC 1,78	P-val 0,00006	BH-adj P-val 0,00481
-----------------	--------	-------------	---------------	----------------------

At protein level, PARP1 was significantly increased in *SF3BI*^{MUT} basoE (suppl table 4), while p21 was not detected. Considering the current literature (PMID: 29950726), an increased expression of PARP1 (even if this does not tell about its function) and an increased expression of CDKN1A/p21 gene could hardly be causative of the observed phenotype. The contribution of this pathway should be investigated deeply in the future.

We agree with the reviewer that phosphorylation of RPA32 at s4/8 is not a specific marker of ssDNA gaps, while RPA2 phosphorylation at Ser33 is mediated by ATR in response to ssDNA. Recent work from the team of Dr E Rothenberg indicated that ATR continuously monitors the levels of RPA recruited on ssDNA within the replisomes in basal conditions of replication, by using single-molecule imaging to visualize the replisome at nanoscale. Furthermore, RPA accumulation is independent of Chk1 but requires ATR-RPA contacts. Finally, ATR-mediated phosphorylation of Ser33 at RPA2 may facilitate RPA's removal from the forks (Yin Y et al, Mol Cell 2021 PMID: 34473946).

Considering that the RPA protein is detectable at the replisome in basal conditions, we reasoned that, showing changes in the quantities of total RPA in the replication stress conditions associated with *SF3BI* mutation in comparison to controls could be challenging with confocal microscopy.

Thus, as suggested by the reviewer and also requested by reviewer 3, we performed immunofluorescence labeling with pRPA32 s33 antibody and results are added to the revised manuscript.

In figure 4G-I of the revised ms (figure 7 below), p-RPA32 s33 foci were detected more frequently in control and *SF*^{WT} erythroblasts compared to p-RPA32s4/8 foci. p-RPA32 s33 foci were abundantly detected in *SF3BI*^{MUT} and in other *SF*^{MUT} MDS indicating ssDNA exposure.

The presence of p-RPA32 s33 foci suggests that ATR, mobilized by ssDNA exposure may initiate the recruitment and phosphorylation of RPAs at serine 33, and that, in conditions of

replication stress driven by SF mutations, RPA (instead of being removed of forks) remains at forks and may undergo subsequent phosphorylation at serine4/8 by DNA-PK.

By contrast, γ H2AX foci formed only in *SRSF2*^{MUT} or *U2AF1*^{MUT} cells as shown by Chen et al, (PMID: 29395063), but not in *SF3B1*^{MUT} cells, suggesting that forks did not collapse when *SF3B1* gene was mutated. Further experiments at a molecular level would provide insights on which molecules decorate the forks when one splicing factor or the other is mutated.

Figure 7 (fig. 4G-I of the ms): Upper panel: Microphotographs of p-RPA32 s33 and p-RPA32 s4/8 immunofluorescence labelling in controls, *SF3B1*^{MUT}, *SRSF2*^{MUT}, *U2AF1*^{MUT}, *SF*^{WT} basophilic erythroblasts. Bottom panel: Histograms representing the percentages of positive cells (> 5 foci per nucleus) in controls, *SF3B1*^{MUT}, *SF*^{MUT}, *SF*^{WT} basophilic erythroblasts. Control cells treated with hydroxyurea are shown as positive controls.

Minor comments:

1. In Methods, it should be described how replication fork velocity was calculated from the DNA combing data.

Answer: The speed of the replication fork was calculated by the ratio of $(d_I + d_{CI}) / (t_I + t_{CI})$, where d_I and t_I represent, respectively, the measured distance (in kb) and labelling time (in min) for IdU incorporation, and d_{CI} and t_{CI} denote the corresponding parameters for CldU incorporation.

2. Legend for Fig. 4E: Replace “Histogram” with “Scatter plot”.

Answer: Scatter plot is now used instead of histogram.

3. Fig.3F: Data for samples treated with RNase H should be added.

Answer: RNase H treatment was not added in the main figure in the interest of space or to facilitate the reading. Instead, the detailed experiments including at least 2 S9.6 immunoprecipitation per sample and a pretreatment of each with RNaseH, performed in the Chédin’s lab are provided as Supplementary Fig. 6.

Figure 8 (Sup Fig. 6 in ms): Examples of DRIP-qPCR in 3 patient-derived erythroblast samples (2 MDS with *SF3B1* mutation, 1 MDS with a mutation in *ATM* gene). NT2 cell line was used as a positive control of DRIP. Upper panel: Bar chart of DRIP-qPCR (as percent input) for one negative control locus (corresponding to an intergenic region ~200kb downstream of *DNAJB1*) and two positive R-loop forming loci (*RPL13A* and *TFPT*). Each bar is the average of IPs performed.

Bottom panel: Bar chart of DRIP-qPCR (as percent input) for one negative control locus (corresponding to an intergenic region ~200kb downstream of *DNAJB1*) and 5 specific loci: 2 at genes with an intron retention reduction (*ABCC5*, *TCIRG1*) and 3 genes without IRR (*IFRD2*, *IREB2*, *TMX2*) in *SF3B1* mutant condition. Each bar is the average of IPs performed. Error bars represent standard error.

4. What SF3B1 mutation is present in the SF3B1-mutant MDS cells used in this study?

Answer: In this study we have included 143 cases with 68 MDS with *SF3B1* mutations that are divided as following:

aa	700	622	625	662	666	742	745	783	splice
n	39	6	7	8	4	1 (bi)	1	1	1

5. In 2019 (Bondu et al., Sci. Transl. Med. 11), the authors published data from transcriptome analysis of the same set of cell lines as those used in this manuscript [BM MNCs isolated from 27 patients in this cohort, including 21 with SF3B1(MUT) MDS, 6 with SF3B1(WT) MDS]. Why did they repeat these experiments?

Answer: We did not repeat the experiments but re-analyzed these data. To increase consistency, we also provided the readers with a new cohort of 189 lower-risk MDS presented in the supplementary data (Sup fig. 1) and with transcriptomic analysis of human erythroblasts at two stages of differentiation (Fig. 2D-J).

6. In several cases, the type of statistical test used is not described in figure legends (e.g. Fig. 3F, Fig.4B,C, Fig. 5D-E). Is Mann-Whitney test appropriate for statistical analysis of the data in Fig. 4I-K? There is only one data point for U2AF1(MUT) cells in these graphs.

Answer: We apologize for the lack of information in the figure legends regarding the statistical tests we used. We have now corrected the figure legends. We used a parametric test ie. unpaired t-test instead of Mann-Whitney test when appropriate. Notably, we have also corrected a mistake: the test used for the comparison of DRIP-qPCR data in figure 3F was an unpaired t-test and not a Mann-Whitney.

In figure 4 H-K of the ms (former Fig. 4I-K and figure 9 below), we considered *U2AF1*^{MUT} and *SRSF2*^{MUT} cases in a unique group. Of note, there was no *U2AF1*^{MUT} sample in p-RPA32 s33 IF experiments. We performed 2 by 2 comparisons using unpaired t-tests.

Figure 9: (Figure 4H-K of the revised manuscript): Quantification of DNA damage either ssDNA exposure or double strand breaks in primary human erythroblasts from MDS with *SF3B1* mutation with *SRSF2* or *U2AF1* mutation (SF^{MUT}), MDS without splicing factor mutation (SF^{WT}) and age-matched healthy controls. Hydroxyurea (HU) treated cells were used as positive control. Unpaired t-test for P-values.

Reviewer #2 (Remarks to the Author):

The manuscript by Rombaut et al., describes characterization of erythroid differentiation models of MDS and the effects of SF3B1 mutation in this setting. In contrast to previous literature the authors describe a reduction in R-loops associated with SF3B1 mutations using genome-wide analysis. These reduced R-loops are correlated with increased replication fork speed and evidence of DNA replication stress but not DNA damage. The authors conduct experiments to support the value of therapeutic R-loops restoration by vorinostat treatment to promote erythroid differentiation. This is an interesting manuscript as it synthesizes a number of unique observations into a novel perspective on SF3B1 and R-loops. Nevertheless, the authors could do more to make connections between their various analyses to strengthen their model. Below are my major suggestions for improvement.

Gene expression changes:

1. At several places in the manuscript the authors describe differential gene or protein expression experiments that link SF3B1 mutations to altered expression of factors involved in genome maintenance, DNA replication and repair. It seems like these factors are both up and down regulated. Are these changes likely to cause a coherent cell biological phenotype? Or are the functional enrichments random? Are there other features of these genes (aside from biological function) that would better explain the effects of SF3B1mut?

Answer: To better characterize these pathways at transcriptome level, we have added separated analyses of the up- and down-regulated genes to the comparison of *SF3B1*^{MUT} versus *SF3B1*^{WT} bone marrow mononuclear cells (Supplementary Fig. 1A), and of the up-regulated genes to the comparison of *SF3B1*^{MUT} versus *SF3B1*^{WT} basophilic or polychromatophilic erythroblasts (Supplementary Fig. 3A). This shows that up-regulated genes were over-represented in the pathways of interest.

In addition, we performed a GSEA of the up- and down-regulated genesets generated in human basophilic and polychromatophilic erythroblasts using publicly available gene signatures.

GOBP_DNA REPAIR
KEGG_NUCLEOTIDE_EXCISION_REPAIR
KEGG_BASE_EXCISION_REPAIR
REACTOME_HOMOLOGOUS_RECOMBINATION
REACTOME_NON_HOMOLOGOUS_END_OINING
KEGG_MISMATCH_REPAIR
PID_FANCONI_PATHWAY
KEGG_DNA_REPLICATION
KEGG_CELL_CYCLE

The lists of genes used for GSEA is provided as a Supplementary Table 9.

In basophilic erythroblasts, DNA replication, Base Excision Repair and G1/S phase checkpoint pathways were significantly deregulated with a trend for Nucleotide excision Repair pathway (see in the table below). By contrast, GOBP_DNA repair, Homologous recombination, Non-Homologous End-Joining, Mismatch Repair, and Cell Cycle were not. In polychromatophilic erythroblasts, none of these pathways were significantly deregulated.

Figure 10. GSEA of transcriptome from *SF3B1*^{MUT} versus *SF3B1*^{WT} basophilic erythroblasts.

These plots are now added in the main figure 2. Accordingly, we reformulated the presentation of the results in the text. The GSEA results with statistics are summarized in the tables below.

For human basophilic erythroblasts

	GS follow link to MSigDB	GS DETAILS	SIZE	ES	NES	NOM p-val	FDR q-val	FWER p-val	RANK AT MAX	LEADING EDGE
1	KEGG DNA REPLICATION	Details ...	35	-0.55	-1.81	0.000	0.019	0.010	6356	tags=40%, list=19%, signal=49%
2	REACTOME BER	Details ...	43	-0.48	-1.66	0.005	0.011	0.005	7097	tags=49%, list=21%, signal=62%
3	G1 - S PHASES CHECKPOINT	Details ...	65	-0.39	-1.47	0.017	0.120	0.177	8685	tags=52%, list=26%, signal=71%
4	KEGG NUCLEOTIDE EXCISION REPAIR	Details ...	43	-0.38	-1.31	0.080	0.239	0.418	6356	tags=35%, list=19%, signal=43%
5	KEGG MISMATCH REPAIR	Details ...	22	-0.36	-1.07	0.370	0.583	0.812	8697	tags=36%, list=26%, signal=49%
6	CELL CYCLE	Details ...	122	-0.25	-1.05	0.332	0.542	0.847	5490	tags=26%, list=17%, signal=31%
7	GOBP_DNA REPAIR (598)	Details ...	536	-0.19	-0.96	0.710	0.688	0.942	6061	tags=21%, list=18%, signal=26%
8	REACTOME HR	Details ...	65	-0.27	-1.05	0.351	0.377	0.401	6356	tags=26%, list=19%, signal=32%
9	REACTOME NHEJ	Details ...	30	0.42	1.26	0.166	0.137	0.228	8654	tags=53%, list=26%, signal=72%

For human polychromatophilic erythroblasts

	GS follow link to MSigDB	GS DETAILS	SIZE	ES	NES	NOM p-val	FDR q-val	FWER p-val	RANK AT MAX	LEADING EDGE
1	KEGG NHEJ	Details ...	10	-0.50	-1.25	0.201	1.000	0.597	1443	tags=20%, list=5%, signal=21%
2	CELL CYCLE	Details ...	122	-0.25	-1.08	0.259	1.000	0.873	8082	tags=37%, list=30%, signal=52%
3	KEGG DNA REPLICATION	Details ...	34	-0.28	-0.96	0.511	1.000	0.972	11658	tags=56%, list=43%, signal=98%
4	GOBP_DNA REPAIR (598)	Details ...	532	-0.18	-0.94	0.712	1.000	0.978	8625	tags=34%, list=32%, signal=48%
5	KEGG HR	Details ...	24	-0.29	-0.91	0.588	1.000	0.984	10516	tags=58%, list=39%, signal=95%
6	KEGG NUCLEOTIDE EXCISION REPAIR	Details ...	42	-0.25	-0.91	0.605	1.000	0.984	7130	tags=31%, list=26%, signal=42%
7	KEGG BASE EXCISION REPAIR	Details ...	33	-0.26	-0.88	0.630	0.939	0.991	7736	tags=45%, list=29%, signal=64%
8	G1 - S PHASES CHECKPOINT	Details ...	64	-0.20	-0.76	0.884	1.000	0.999	4278	tags=17%, list=16%, signal=20%
9	ATR BRCA1 PATHWAY	Details ...	20	-0.23	-0.69	0.878	1.000	1.000	1865	tags=10%, list=7%, signal=11%
10	KEGG MISMATCH REPAIR	Details ...	21	-0.21	-0.64	0.937	0.955	1.000	8083	tags=29%, list=30%, signal=41%

	GS follow link to MSigDB	GS DETAILS	SIZE	ES	NES	NOM p-val	FDR q-val	FWER p-val	RANK AT MAX	LEADING EDGE
1	G1 PHASE CHK PROTEINS IN CELL CYCLE CHECKPOINT CONTROL	Details ...	27	0.39	1.22	0.183	0.330	0.659	2595	tags=19%, list=10%, signal=20%
2	DNA DAMAGE CHECKPOINT G2 CELL CYCLE	Details ...	23	0.21	0.64	0.934	0.954	1.000	4029	tags=17%, list=15%, signal=20%

We also analyzed the expression of the proteins corresponding to the genesets used for GSEA of transcripts. In the proteome of basophilic and polychromatophilic erythroblasts, 36 proteins of DNA repair pathway were up or downregulated, and several of them involved in Base Excision Repair (BER, n=6), Nucleotide Excision Repair (NER, n=6) and DNA replication (n=4). Few of them were involved in homologous recombination (n=2), NHEJ (n=2), mismatch repair (n=1) or Fanconi pathway (n=1) suggesting that these latter pathways were not engaged.

Protein	GOBP_DNA_REPAIR (598)	KEGG_NER (44)	KEGG_BER (90)	REACTOME_NHEJ (68)	KEGG_MISMATCH_REPAIR (16)	PID_FANCONI_PATHWAY (47)	KEGG_DNA_REPLICATION (36)
AP5Z1	Red						
ARID1A							
ARID1B	Blue						
BRCC3	Blue						
CHAF1B	Red						
CHD1L							
DDB1	Blue						
DTX3L	Blue						
EP400	Blue						
ERCC2	Red						
FEN1	Blue		Blue	Blue			Blue
FTO	Blue						
GTF2H1	Red						
GTF2H4	Red						
HLTF	Blue						
LIG1	Blue		Blue			Blue	Blue
LIG3	Blue						
MDC1	Blue						
PARP1	Red		Red				
PDS5A	Blue						
PNKP	Blue		Blue				
POLA1	Blue						Blue
POLE	Blue			Blue			Blue
PSME4	Red						
RECQL	Blue						
RRM1	Blue						
SETX	Red						
SF3B5	Red						
SMARCC1	Red			Red			
SMC5	Blue						
SMC6	Blue						
SUPT7L	Red						
TNKS1BP1	Red						
TRIM28	Red						
UBE2V2	Red						
XRCC3	Red			Red			Red
TCEA1	Red						
POLR2H	Red						
MAP2K2	Red						
ORC1							Red

Figure 11. Up- and down-regulated proteins in *SF3B1*^{MUT} versus *SF3B1*^{WT} basophilic erythroblasts.

DNA replication and BER pathways share common actors like LIG1, POLE, FEN1 (PMID: 31320249). FEN1 (5'FLAP-endonuclease and gap endonuclease) that possesses 5'-exonuclease and gap-endonuclease activities participate in Okazaki fragment maturation, stalled replication fork rescue, and base excision repair. POLE is replicating the leading strand of the fork with high fidelity. LIG1 is an ATP-dependent DNA ligase involved in DNA replication and base excision repair. Furthermore, PARP1 has been involved in the recruitment of LIG3 to promote Okazaki fragment maturation (PMID 29983321). The deregulated expression of these proteins is consistent with the observation that, in the context of a chronic DNA replication stress triggered by *SF3B1* mutation, protective mechanisms such as the BER pathway could be engaged to make this replication stress tolerable and maintain genome stability. These hypotheses will be addressed experimentally in the future.

The figure 11 is shown as Supplementary Fig. 4C to clarify Ingenuity Pathway Analysis provided in Fig. 2L.

2. Can the authors connect the reported changes in protein expression for DNA damage checkpoint proteins (Figure 6D) with any of the phenotypes reported? Its unclear what functional links are being suggested between changed protein levels and replication stress phenotypes?

Answer: We thank the reviewer for this comment that helps us to clarify the connection between deregulated protein pathways and the phenotype.

As shown in our answer to the first question of this reviewer, we have clarified the types of DNA repair pathways affected by the presence of *SF3B1* mutation in human cells at transcript level with GSEA and at protein level with IPA. These two analyses were consistent, highlighting the deregulation of DNA replication together with base excision repair and nucleotide excision repair both reported playing a role in DNA replication stress.

The murine G1E-ER4 cells was engineered to express *Sf3b1* mutation using the CRISPR-cas9 strategy. Functional studies in these cells showed features comparable to primary human cells such as increased proliferation capacities, increased DNA synthesis in BrdU experiments and increased p-RPA32 foci. Transcriptomic analysis of proE (without estradiol) and basoE (post induction of GATA1 with estradiol) showed a high proportion of intron retention reductions in genes over-represented in DNA repair and cellular response to DNA damage (Fig. 5A, B). As mentioned in the text (page 9),

“despite the substantial species specificity of RNA splicing, the deregulated pathways associated with differential splicing events in murine cells appeared similar to those identified in human cells”.

Furthermore, the Ingenuity Pathway Analysis of mouse G1E-ER4 proerythroblast proteome identified 11 terms significantly deregulated in murine cells (Fig. 5C of the ms), 10 of them being also deregulated in human cells (fig. 2L), assessing the consistency of the murine cell line model. Interestingly, 4 of the 15 proteins (Lig1, Lig3, Pnkp, Parp1) involved in BER and DNA replication pathways specifically deregulated in human *SF3B1*^{MUT} erythroblasts (Fig. 11) were also similarly deregulated in G1E- ER4 *Sf3b1*^{K700E/+} cells. This is consistent with the onset of a DNA replication stress as suggested by the increase of DNA synthesis and p-RPA32 foci as well as reported in human cells.

We provide the reviewer with DNA combing experiments showing that G1E- ER4 *Sf3b1*^{K700E/+} before and after induction of GATA1 with estradiol have a higher fork speed than isogenic G1E- ER4 *Sf3b1*^{+/+} cells.

Figure 12: DNA combing. Scatter plots showing preliminary results of fork speed (as means +/- SD). Mann-Whitney for *P*-value

Finally, one term appeared specifically in murine cells: *DNA damage checkpoint of cells*. The five proteins involved in *DNA damage checkpoint of cells*, Atm, Gmnn, Tp53bp1 (down regulated in *Sf3b1*^{K700E/+} murine cells) and Prkcd, Chk1 (up-regulated in *Sf3b1*^{K700E/+} murine cells) were not deregulated in human *SF3B1*^{MUT} cells. Although Chk1 might be upregulated in murine *Sf3b1*^{K700E/+} cell line, we did not observe Chk1 activation in these cells. Thus, we cannot exclude that the murine *Sf3b1*^{K700E/+} cell line behaves differently from human primary cells.

Taken together these results suggest that *Sf3b1*^{K700E/+} cell line endures a chronic replication stress that triggers a DNA repair response possibly involving BER pathway. Additional experiments are required to address this hypothesis in the future.

3. Specifically, the authors (on page 13) seem to suggest that low levels of FANCD2 could be responsible for the fast fork speed and RPA32 phosphorylation observed in the SF3B1 mutant cells. If so they should confirm that FANCD2 protein is significantly reduced by western blot or another orthogonal assay, and then explain their model in light of the publications stating that FANCD2 depletion increases R-loops (not decreases). I was confused by the discussion linking low FANCD2 to the observed phenotypes since there are multiple independent reports starting in 2015 suggesting that low FANCD2 increases R-loops (the opposite of what the authors see with SF3B1mut).

Answer: We agree with the referee that several papers in the literature showed that FANCD2 depletion may provoke R-loop formation and that, for instance, FANCD2 could be recruited to R-loops and suppressed R-loops by recruiting RNA processing factors (Okamoto et al, FEBS J 2019; PMID 30431240).

Our proteomic analysis by mass spec provided an absolute quantification of proteins expressed in copy number compared to histones. Compared to *SF*^{WT} samples, FANCD2 absolute quantity

was diminished in *SF3B1*^{MUT} basophilic erythroblast samples by 30% and in *SF3B1*^{MUT} polychromatophilic erythroblasts by 40%.

As suggested, we performed Western blot analysis of FANCD2 on additional samples harvested from erythroid cell cultures. The results showed that expression of FANCD2 in *SF3B1*^{MUT} erythroblasts, was similar to controls. By contrast, FANCD2 appeared decreased in *SRSF2*^{MUT} or *U2AF1*^{MUT} erythroblasts which is consistent with augmented R-loops (Fig. 13).

Figure 13: Expression of FANCD2 in erythroblasts by mass spec expressed as copy number per cell (left panel) and by Western blot (right panel). Two loading controls are shown.

Because, FANCD2 under-expression in *SF3B1*^{MUT} erythroblasts was not confirmed, we remove the part of discussion dedicated to FANCD2.

Additional analyses, clarifications or experiments

1. The whole proteome analysis in Figure 2 is interesting but seems like more analysis could be done. For example, are specific types of splicing changes associated with proteome changes? Can mutant proteins be detected? If it simply reflects the RNA-seq it does not seem particularly useful - could additional analyses allow the authors to say more about the proteome changes?

Answer: As suggested by the reviewer, we have performed analyses of correlation between RNA expression and protein expression. Starting from the normalized expression level of ~21,000 transcripts obtained by RNA-seq at each stage of erythroid differentiation in mutated condition, we compared the expression of the transcripts corresponding to the 5,401 proteins and 4,862 proteins that we obtained by mass spec from ProE/basophilic erythroblasts and from polychromatophilic erythroblasts, respectively.

As expected from previous results from our team, the correlation between transcripts and proteins expression at different stages of erythropoiesis is weak (Gautier et al, Cell Rep 2016, PMID: 27452463 & Blood Adv 2022, PMID: 32282884). A bi-parametric scatter plot of the

whole transcripts and proteins indicated by their fold-change of expression between $SF3B1^{MUT}$ and $SF3B1^{WT}$ basoE (grey dots) demonstrated a poor correlation. Examples of transcripts with intron retention gain in mutant cells (left) or reduction in mutant cells (right) were superimposed. Again, the correlation between IR-transcript and protein expression changes was weak (Fig. 14).

Figure 14: Correlation between transcripts and proteins changes expressed in log₂ (FC) in $SF3B1^{MUT}$ compared to $SF3B1^{WT}$ basoE (grey dots). Red dots on left panel represent IR-transcripts up-regulated in $SF3B1^{MUT}$ basoE. Blue dots on right panel represent IR-transcripts down-regulated in $SF3B1^{MUT}$ basoE.

We also interrogated the correlation between several IR-transcripts and corresponding proteins, individually. In our analysis, few IRR-transcripts (*BCS1L*, *CBS*, *FGFR1OP*, *NPL*, *PPOX*, *SEPT2*, *SEPT8*, *TPP2*) had a deregulated expression of the corresponding protein in $SF3B1^{MUT}$ cells. All of these proteins were significantly down-regulated.

There is a limitation to this analysis linked to the performance of mass spec that detected ~5,000 proteins per sample among the 10,000 proteins of a cell. However, these results suggest that major post-transcriptional regulations of gene expression could occur in these cells.

Mutant protein can be detected. We have already published a mutant protein of erythroferrone (Bondu, Alary et al. *Sci Transl Med.* 2019; PMID: 31292266) following a candidate-protein approach using specific proteomic methods of detection and identification. Here, classical proteomic methods were used to identify proteins listed in the Uniprot-Swissprot database. Our datasets could be reanalyzed to search for mutant proteins using an in-house dedicated pipeline.

2. Additional DNA combing data could give more insight to whether R-loops are really a major barrier to fork speed in erythroblasts. For example, treatment with transcription inhibitors or RNaseH should equalize forkspeed in the $SF3B1^{WT}$ and mut samples if the authors are correct. Fork speed can also be regulated by other signals, including changes to the machinery for fork reversal. Doing the appropriate controls with transcription inhibitors or RNaseH would help the authors better describe potential mechanisms.

Answer: We fully agree with the reviewer comment that RNaseH1 and transcription inhibitors can help deciphering the role of R-loops on fork speed.

We addressed this point using an HDACi (used in clinic). We showed that it causes increased levels of R-loop signal as well as reduced fork speed in a patient with *SF3B1* mutation. These data further support a relationship between fork speed and R-loops levels in *SF3B1*^{MUT} cells. In addition, it is clinically relevant as shown on the differentiation of the cells (Fig. 15).

Figure 15 (Fig. 6F of the ms): DNA combing before and after treatment with HDACi 0.2 μ M for 20h of one control, one *SF3B1*^{MUT} and one *SF3B1/EP300*^{MUT} erythroblast samples.

3. Why did the authors monitor RPA32 s4/8, which is deposited by DNAPK, rather than ser33, which is deposited by ATR?

Answer: We agree with the reviewer that RPA2 phosphorylation at Ser33 is mediated by ATR in response to ssDNA, while p-RPA32s4/8 deposited by DNA-PK is not a specific marker of ssDNA gaps. Recent work from the team of Dr E Rothenberg indicated that ATR continuously monitors the levels of RPA recruited on ssDNA within the replisomes in basal conditions of replication, by using single-molecule imaging to visualize the replisome at nanoscale. Furthermore, ATR-mediated phosphorylation of Ser33 at RPA2 may facilitate RPA's removal from the forks (Yin Y et al, Mol Cell 2021 PMID: 34473946).

Thus, as suggested by the reviewer and also requested by reviewer 1, we performed immunofluorescence labeling with pRPA32 s33 antibody and results were added to the revised manuscript.

In figure 4G-I of the revised ms (Fig.15 below), p-RPA32 s33 foci were detected more frequently in control and *SF*^{WT} erythroblasts compared to p-RPA32s4/8 foci. p-RPA32 s33 foci were abundantly detected in *SF3B1*^{MUT} and in other *SF*^{MUT} MDS indicating ssDNA exposure.

The presence of p-RPA32 foci suggests that ATR, mobilized by ssDNA exposure may initiate the recruitment and phosphorylation of RPAs at serine 33, and that, in conditions of replication

stress driven by splicing factor mutations, RPA (instead of being removed of forks) remains at forks and may undergo subsequent phosphorylation at serine4/8 by DNA-PK. By contrast, γ H2AX form foci only in *SRSF2*^{MUT} or *U2AF1*^{MUT} cells, but not in *SF3B1*^{MUT} cells, suggesting that forks may collapse as suggested by Chen et al, (PMID: 29395063). Further experiments at a molecular level would provide insights on which molecules decorate the forks when one splicing factor or the other is mutated (Fig. 16).

Figure 16: (Figure 4G-I of the revised manuscript)): Upper panel: Microphotographs of p-RPA32 s33 and p-RPA32 s4/8 immunofluorescence labelling in controls, *SF3B1*^{MUT}, *SRSF2*^{MUT}, *U2AF1*^{MUT}, *SF*^{WT} basophilic erythroblasts. Bottom panel: Histograms representing the percentages of positive cells (> 5 foci per nucleus) in controls, *SF3B1*^{MUT}, *SF*^{MUT}, *SF*^{WT} basophilic erythroblasts. Control cells treated with hydroxyurea are shown as positive controls.

This marker together with γ H2AX has been used also to investigate whether the treatment with HDACi vorinostat may induce increase ssDNA exposure or induce DNA damage in these cells (Fig. 17).

Figure 17 (Fig. 6H-I of the revised ms): Immunofluorescence labelling of p-RPA32 s33 and γ H2AX was performed on erythroblasts from 3 *SF3B1*^{MUT}, 6 *SF3B1*^{WT} MDS (1 *SRSF2*^{MUT}, 3 *SF*^{WT}, 2 w/o mutation) \pm HDACi 0.5 μ M for 20h. Hydroxyurea (HU) 5 mM for 3h was used as positive control. Nuclei were labeled with DAPI. Representative experiments at day 11. **I.** Quantification of immunofluorescence signals of p-RPA32 s33 and γ H2AX. Cells with > 5 intranuclear foci were considered positive. Results are expressed as mean percentages \pm SD of positive cells.

4. The increase in fork speed and RPA phosphorylation but lack of DNA damage is somewhat surprising. Have the authors considered enhancing fork speed with PARP inhibitors (i.e. as in PMID: 29950726)? If changes to PARP expression could partly explain the increase rate it could explain the mechanism.

Answer: We thank the reviewer for this very interesting question. The work published in Nature in 2018 by the group of Dr Jiri Bartek demonstrated a role for the PARP/p21/p53 pathway in the regulation of fork speed. In this work, inhibition of PARP1 activity but not the depletion of PARP1 (in which case parylation is compensated by other PARP proteins), induced accelerated fork speed. Parylation and p21 are suppressors of fork speed. Thus, the depletion of p21 could accelerate fork velocity.

Looking at our transcriptomic data in basophilic and polychromatophilic erythroblasts (suppl table 2), we discovered that a PARPBP, but not PARP1 or other family members or CDKN1A/p21, was affected by two splicing events (exon skipping and alternative splicing donor). PARPBP alternative transcript was less expressed in *SF3B1*^{MUT} polychromatophilic erythroblasts compared to wild-type without affecting the total amount of the transcript. In basophilic erythroblasts, the expression of CDKN1A/p21 transcript was significantly increased:

ENSG00000124762 CDKN1A Log2FC 1,78 P-val 0,00006 BH-adj P-val 0,00481

At protein level, PARP1 is significantly increased in mutant basoE (Suppl Table 4), while p21 was not detected.

Considering the current literature (PMID: 29950726), increased expression of PARP1 (even if this does not tell about its function) and of CDKN1A/p21 gene is unlikely responsible for the phenotype of *SF3B1*^{MUT} erythroblasts.

A very recent work demonstrated the sensitivity of *SRSF2*^{MUT} or *U2AF1*^{MUT} leukemic cells to PARPi (Liu et al, Cancer Res Nov 2023; PMID: 37967363). It has been also shown by Dr R Natrajan (PMID: 37524790) that *SF3B1*^{MUT} K562 or MEL cell lines are sensitive to PARPi, because of a defective response to PARPi-induced replication stress leading to G₂/M cell cycle

arrest. Preliminary tests of olaparib on the viability of *Sf3b1*^{K700E/+} murine proerythroblasts showed their higher sensitivity to increasing concentrations of the drug compared to *Sf3b1*^{+/+} cells (Fig. 18).

Figure 18: Effects of PARP inhibitor Olaparib on the viability *Sf3b1*^{K700E/+} and *Sf3b1*^{+/+} murine proerythroblasts (n=3). Multiple t-test for q-value. * q<0.05.

Due to limited number of available human samples, it was not possible to investigate the effects of olaparib and vorinostat in parallel. In an attempt to rescue ineffective erythropoiesis of *SF3BI*^{MUT} bone marrows by providing alternative strategies to the eradication of clonal cells in low risk MDS-*SF3BI*, we concentrated our efforts in deciphering the pro-differentiating effect of HDACi.

5. The effects of SAHA on R-loops are not that well characterized. Do the authors think that expression of a subset of genes is the key mechanism behind the observed rescue of differentiation in SAHA treated cells? Is expression analysis available in this model to support rescue changes to specific decreased R-loop target genes? The use of R-loop restoration for therapy is a really innovative approach of the manuscript. Do the authors think that alternative R-loop modulators would also restore differentiation? Understanding the mechanisms of SAHA changes to gene expression, or the use of alternative R-loop inducers to rescue differentiation (e.g. sub-lethal doses of camptothecin or other agents), would help to understand this.

Answer: We thank the reviewer for these very interesting suggestions.

During the course of this study we collected several arguments providing a rationale for testing the effects of HDACi on *SF3BI*^{MUT} erythropoiesis:

- IRR-transcripts were found involved in the regulation of histone deacetylation by Gene Ontology analysis
- Histone deacetylase SIN3A is interacting with the THO complex which prevent the formation of co-transcriptional R-loops
- R-loop removers SETX and THOC6 are over-expressed in *SF3BI*^{MUT} basophilic erythroblasts and numerous R-loops are lost in these cells
- HDAC inhibitors trichostatinA and SAHA are reported as R-loop inducers.

We have found that a pretreatment with HDACi restored R-loops and improved erythropoiesis. On the basis of our findings that genes with R-loop loss at their promoter had a lower expression, we searched for genes with an R-loop loss near promoters and studied the effect of

HDACi on R-loop formation. Several genes like *BCL2L1*, *PTPN11*, *ARPC3*, *NCOA4* or *SUZ12* which R-loops were lost or decreased at their promoter in *SF3B1*^{MUT} samples gained R-loops after HDACi treatment. By contrast, some other genes like *HK1* which presented a large R-loop at promoter in *SF3B1*^{MUT} condition did not change its pattern upon HDACi treatment. At *SUZ12* locus, we observed a clear decrease of peaks in *SRSF2*^{MUT} or *NRAS/biTET2*^{MUT} erythroblasts.

Furthermore, *BCL2L1*, *PTPN11* and *NCOA4* expression by RT-qPCR tended towards a higher expression level after HDACi treatment. By contrast, *HK1* expression remained similar.

Interestingly, *BCL2L1* gene is a target of GATA1 and STAT5 playing a major role in erythroid cell maturation. Increasing its expression may contribute to the improvement of erythroid differentiation observed upon HDACi treatment.

We exclude a direct effect of HDACi on splicing. *BCL2L1* transcript was affected by an intron retention reduction in *SF3B1*^{MUT} cells. HDACi did not modify the expression of the short and long transcripts of *BCL2L1*.

Furthermore, HDACi vorinostat has already been using in a phase I clinical trial in MDS patients with limited efficacy but no anemia reported (PMID: 17962510) or added to azacitidine in a phase II clinical trial to prevent the progression to leukemia in unselected groups of MDS patients without significant clinical improvement possibly related to the cumulative toxicities of the two drugs (PMID: 27977052). Our study provides a rationale for drug repositioning trial in lower-risk MDS patients.

Low doses of other putative R-loop inducers such as camptothecin, or romidepsin have not been tested yet. Camptothecin is expected cytotoxic for primary erythroblasts. In vitro studies of romidepsin showed the induction of apoptosis of AML/MDS cell lines (PMID: 18245545) and the accumulation of R-loops (PMID: 34050002). We agree that it could be interesting to test the effects of romidepsin on erythroblasts.

Need to address literature context:

1. The lack of DNA damage as marked by γ H2AX or 53BP1 in *SF3B1* mutant cells is surprising given the literature. The authors should attempt to explain this discrepancy. The authors seems to assign some importance to changes in gene expression of DNA repair/replication factors in Figure 1/2 but do not see increased DNA damage (as others have). Are there functional consequences to changes in genome stability gene expression in *SF3B1*mut or not? If so, what are the consequences?

Answer:

We observed a mild replication stress in *SF3B1*-mutated erythroblasts compared to *SRSF2*^{MUT} or *U2AF1*^{MUT} erythroblasts. The ssDNA exposure that we report by the visualization of p-RPA32 s33 foci could be caused by accelerated fork velocity which can produce post-

replicative ssDNA gaps (Hashimoto Y. Nat Struct Mol Biol 2010 PMID: 20935632; Lossaint G. PMID: 23993743).

Looking more deeply into the DNA repair pathways globally deregulated we noticed by GSEA that BER and DNA replication were significantly affected while HR, MMR or NHEJ did not. Thus, *SF3B1*^{MUT} erythroblasts may endure a chronic replicative stress that is tolerable because of engagement of protective mechanisms such as BER. This observation is consistent with low rate of mutations and good prognosis of the MDS patients with *SF3B1* mutation (Fig. 19).

Figure 19: GSEA using KEGG DNA replication, reactome base excision repair and G1-S phases checkpoint of human *SF3B1*^{MUT} basophilic erythroblasts compared to *SF3B1*^{WT} erythroblasts.

Human basophilic erythroblasts

	GS follow link to MSigDB	GS DETAILS	SIZE	ES	NES	NOM p-val	FDR q-val	FWER p-val	RANK AT MAX	LEADING EDGE
1	KEGG DNA REPLICATION	Details ...	35	-0.55	-1.81	0.000	0.019	0.010	6356	tags=40%, list=19%, signal=49%
2	REACTOME BER	Details ...	43	-0.48	-1.66	0.005	0.011	0.005	7097	tags=49%, list=21%, signal=62%
3	G1 - S PHASES CHECKPOINT	Details ...	65	-0.39	-1.47	0.017	0.120	0.177	8685	tags=52%, list=26%, signal=71%
4	KEGG NUCLEOTIDE EXCISION REPAIR	Details ...	43	-0.38	-1.31	0.080	0.239	0.418	6356	tags=35%, list=19%, signal=43%
5	KEGG MISMATCH REPAIR	Details ...	22	-0.36	-1.07	0.370	0.583	0.812	8697	tags=36%, list=26%, signal=49%
6	CELL CYCLE	Details ...	122	-0.25	-1.05	0.332	0.542	0.847	5490	tags=26%, list=17%, signal=31%
7	GOBP_DNA REPAIR (598)	Details ...	536	-0.19	-0.96	0.710	0.688	0.942	6061	tags=21%, list=18%, signal=26%
8	REACTOME HR	Details ...	65	-0.27	-1.05	0.351	0.377	0.401	6356	tags=26%, list=19%, signal=32%
9	REACTOME NHEJ	Details ...	30	0.42	1.26	0.166	0.137	0.228	8654	tags=53%, list=26%, signal=72%

Human polychromatophilic erythroblasts

	GS follow link to MSigDB	GS DETAILS	SIZE	ES	NES	NOM p-val	FDR q-val	FWER p-val	RANK AT MAX	LEADING EDGE
1	KEGG NHEJ	Details ...	10	-0.50	-1.25	0.201	1.000	0.597	1443	tags=20%, list=5%, signal=21%
2	CELL CYCLE	Details ...	122	-0.25	-1.08	0.259	1.000	0.873	8082	tags=37%, list=30%, signal=52%
3	KEGG DNA REPLICATION	Details ...	34	-0.28	-0.96	0.511	1.000	0.972	11658	tags=56%, list=43%, signal=98%
4	GOBP_DNA REPAIR (598)	Details ...	532	-0.18	-0.94	0.712	1.000	0.978	8625	tags=34%, list=32%, signal=48%
5	KEGG HR	Details ...	24	-0.29	-0.91	0.588	1.000	0.984	10516	tags=58%, list=39%, signal=95%
6	KEGG NUCLEOTIDE EXCISION REPAIR	Details ...	42	-0.25	-0.91	0.605	1.000	0.984	7130	tags=31%, list=26%, signal=42%
7	KEGG BASE EXCISION REPAIR	Details ...	33	-0.26	-0.88	0.630	0.939	0.991	7736	tags=45%, list=29%, signal=64%
8	G1 - S PHASES CHECKPOINT	Details ...	64	-0.20	-0.76	0.884	1.000	0.999	4278	tags=17%, list=16%, signal=20%
9	ATR BRCA1 PATHWAY	Details ...	20	-0.23	-0.69	0.878	1.000	1.000	1865	tags=10%, list=7%, signal=11%
10	KEGG MISMATCH REPAIR	Details ...	21	-0.21	-0.64	0.937	0.955	1.000	8083	tags=29%, list=30%, signal=41%

	GS follow link to MSigDB	GS DETAILS	SIZE	ES	NES	NOM p-val	FDR q-val	FWER p-val	RANK AT MAX	LEADING EDGE
1	G1 PHASE CHK PROTEINS IN CELL CYCLE CHECKPOINT CONTROL	Details ...	27	0.39	1.22	0.183	0.330	0.659	2595	tags=19%, list=10%, signal=20%
2	DNA DAMAGE CHECKPOINT G2 CELL CYCLE	Details ...	23	0.21	0.64	0.934	0.954	1.000	4029	tags=17%, list=15%, signal=20%

2. The authors do not adequately address the previous findings of the Boultonwood and Savage labs (e.g. PMID: 32076118 and PMID: 35027467), which suggest that SF3B1 mutations actually increase R-loops as a cause of replication stress and drug sensitivity. I recognize the clear methodological advantages of DRIP-seq done here as a means to characterize R-loops, but given that there are multiple independent publications that oppose the findings here, the authors should directly address them and highlight advantages of this study.

Answer: We thank the reviewer to give us the opportunity to comment the data from our colleagues.

Several published papers from Dr Savage, Dr Boultonwood and Dr Nowak (PMID: 32076118; PMID: 35027467; PMID: 33054116) reported an increase of R-loops in various cell lines and patient-derived samples with splicing factor mutation. R-loop quantity in *SF3B1*^{MUT} cells was assessed by immunofluorescence experiments using S9.6 antibody that identifies RNA:DNA hybrids but also RNA:RNA hybrids which are much more abundant in the cell and in particular in the cytoplasm than RNA:DNA hybrids in the nucleus. In two papers, the authors looked at R-loops in cell line models expressing an oncogenic background and engineered to express the mutant *SF3B1*^{K700E} like the K562 cell line, a CML derived cell line containing a *TP53* mutation and at least in some clones *RECQL4*, *BRCA1*, *MLH1*, *BIRCC6* or *FANCC* mutations or the H2591 mesothelioma-derived line, the U2OS osteosarcoma-derived cell line. In these models, elevated R-loop levels are evidenced.

RNaseH1 that displays affinity for RNA:DNA hybrids but also for dsRNA (Pallan and Egli, Cell Cycle 2008 PMID: 18719385; Chedin et al. EMBO J; PMID: 33411340) reduced the signals. When transfected in K562 cell lines, RNaseH1 almost completely abolished the cytoplasmic and nuclear signals (Dr Boultonwood) which does not help for the assessment of signal specificity for RNA:DNA hybrids.

Interestingly, in two papers, primary CD34+ cells from patients with various types of MDS were also studied showing the increase of S9.6 signals in SF-mutated samples irrespective of the mutation. The specificity of the signals remained questionable due to RNA:RNA hybrid recognition.

By contrast using DRIP methods implicates working at DNA level abrogating the risk of catching dsRNA. Furthermore the specificity of DRIP peaks is confirmed by systematic in vitro pre-treatment with RNase H1 that disrupt the RNA:DNA signals. Compared to DRIP-seq, defective RNaseH1-based method (R-ChIP) mainly recognizes a subset of loci with a clear bias toward G-rich loci associated with promoter-proximal pausing of RNA polymerase II (Chen et al, 2017; PMID: 29395063). By contrast, DRIP-seq identifies R-loops at gene bodies, TTS and intergenic regions.

Clarifications to the overall model:

1. I was a bit confused by the model for why R-loop accumulation is decreased in SF3B1-mut cells. The authors cite a correlation between increased intron excision and lower R-loops. Can the authors say more about WHY SF3B1mut would increase intron excision? Is splicing flux

faster? This seems inconsistent with previous literature focused on SF3B1 which found alternative 3'SS selection, skipped exons and other features of the transcriptome. A statement on Page 9, saying "this suggests that transcription initiation is less active in SF3B1mut cells." was additionally confusing. How would increased intron excision be linked to reduced transcription initiation? Would SF3B1 even be recruited to transcription initiation sites, or do the authors envision an indirect effect? The reduced R-loop observation is one of the most surprising findings of the manuscript and the authors should do more to put it in context, contrast with the literature, and try to explain the mechanism.

Answer: The loss of intron excision has already been reported in previous publication from Dr S Ogawa and Dr M Cazzola appearing in Nat Comm in 2018 (PMID: 30194306). Why SF3B1 mutation would increase intron excision is still unknown. However, some elements are available for the discussion:

- Mutations in *SF3B1* gene do affect the HEAT domains that align in the 3-dimensional structure of the complex as a platform interacting with the template RNA.
- Mutations are gain-of-function not abrogating the interaction with the template RNA
- It has been proposed that the increase of transcripts with excised intron was not due to the degradation of transcripts with intron retention and suggested that SF3B1 mutant increases intron excision,
- However, data appearing in BiorXiv <https://doi.org/10.1101/2023.02.25.530019> using omics methods suggest that mutant *SF3B1* does not increase co-transcriptional splicing, but rather impairs transcription elongation. It is not clear whether a subset or all transcripts are targeted by impaired transcription elongation.

We agree that it is rather unlikely that increased intron excision is linked to reduced transcription initiation. However, SF3B1 has been identified in R-loop interactome at gene promoter (Mosler et al. 2021; PMID: 34916496). Thus, mutant SF3B1 could notably modify the kinetics of transcription elongation, as reported (BiorXiv 2023; <https://doi.org/10.1101/2023.02.25.530019>). It is still unclear whether impaired RNAPII transcription elongation is due to intensive recruitment of spliceosome for intron excision consistent with frequent IRR or, conversely, to impaired spliceosome assembly.

We rephrased the discussion p. 15: "Since the SF3B1 protein is detected in the vicinity of R-loops at promoters together with U2AF1 and SRSF2, the SF3B1 mutation could notably modify the kinetics of transcription elongation, as generally reported for splicing factors (BiorXiv <https://doi.org/10.1101/2023.02.25.530019>).

Reviewer #3 (Remarks to the Author):

In this manuscript Rombaut et al study the cause and impact of aberrant R-loop formation in SF3B1 mutant MDS.

Previous studies have identified increased R-loop formation in SRSF2 and U2AF1 mutant MDS that results in replication stress and sensitivity to ATR inhibitors. In contrast to these finding, Rombaut et al show that SF3B1 mutations results in reduced R-loop formation in part due to increased splicing of normally detained/retained introns. This process specifically impairs erythropoietic differentiation that is in part driven by regulation of gene expression via intron retention and non-sense mediated decay.

The authors show in detail that SF3B1 mutations result in accelerated transcription as evident by increased replication fork speed, loss of R-loops with “exposure” of cells to single-stranded DNA and replication stress and compromised differentiation. They identify a signature related to HDACi induced signatures and determine that HDAC inhibitors slow transcription speed and restore R-loop formation and thereby erythropoietic differentiation.

The authors' conclusions are based on robust multi-omic analysis (RNAseq, DRIPseq, proteome, etc.) of primary SF3B1 mutant, wildtype and SRSF2 or U2AF1 mutant MDS erythroblasts, isogenic human cell lines and Sf3b1 mutant murine erythroblasts. The authors convincingly show decreased R-loop formation, enhanced replication fork speed, increased replication stress, and impaired differentiation of SF3B1 mutant cells. Interestingly, they identify a signature in the proteome that includes targets of clinically approved HDAC inhibitors and treatment of mutant cells indeed partially rescues erythroid differentiation. Overall, the authors' findings are very well documented and mechanisms are analyzed in detail. The finding that HDAC inhibition reduces the aberrantly accelerated replication fork speed, increases R-loop formation and improves erythropoietic differentiation is of interest and highly relevant.

The rescue of differentiation and replication stress by R-loop induction via HDACi however, while a major novel point of this manuscript, is insufficiently proven. The murine cells may replicate some aspects of SF3B1 mutant MDS but the rescue effect of HDACi should be shown in human SF3B1 mut human MDS erythroblast in contrast to SF3B1 WT and SF mutant human MDS erythroblasts. In addition, restoration of intron-retention - R-loop formation - gene expression should be shown at least for select key targets

Answer:

We thank the reviewer for his/her time spent to evaluate our manuscript.

Major Comments:

1-Figures 3B is the key point of the paper: loss of R-loops in SF3B1 mutant cells. However, the next most important point, loss of intron retention as a mechanism of loss of R-loops is not as well supported: In figure 3D only a minority of lost R-loops are explained by loss of retained introns. 63% of the peaks lost are located in gene bodies, purportedly mainly in introns. But 95% of peaks lost in Fig 3D are not explained by RRI suggesting that the effect of lost R-loops is mainly driven by another mechanism. maybe add something in the discuss or limitations of the study.

Answer: We do agree that the integrative analysis of splicing events related to SF3B1 and R-loops showed that only part of the R-loop loss coincides with intron retention reduction (IRR).

As suggested by the reviewer, we have added to the discussion several elements:

1. Dr F Chédin, co-author of the manuscript has described two types of R-loops:
 - type I short R-loops at promoter characterized by high G-content and epigenetic regulation.
 - type II long R-loops on gene body which G-content appears lower (Ginno et al, 2012)DRIP-seq can detect both of them (Chédin et al, EMBO J 2021). However, finding the limits of the type II R-loops can be challenging making difficult the assignation to introns. Therefore, we quantified R-loops within restriction fragments generated during the experiment.
2. We cannot exclude that sample heterogeneity could blur the picture. For instance, in RNA-seq data, the set of genes targeted by an IRR is well conserved. However, the intensity of the loss is variable. Similarly, the R-loop landscape is conserved at some loci, but modified in pathological conditions (unscheduled R-loops) again with variable intensity.
3. Finally looking for the localization of R-loops across the genome pointed out a great number of R-loop losses at gene promoter/TSS that may result from a distinct mechanism.

This is commented in the discussion p.15: Increased intron excision occurring when *SF3B1* is mutated may suppress R-loops along the gene bodies. We report here that most of IRR overlap with lost R-loops. However, because R-loops were lost also in promoter-TSS, intergenic regions and TTS, **IRR is not the unique mechanism for R-loop loss in these cells.**

2-What are "consensus" R-loops (Figure 3G). What are the R-loops reported here? Can a motif analysis be provided?

Answer: We used the term of consensus R-loops to refer to the R-loops shared by all the samples of a given genetic group. We understand that this term is not appropriate and we replace it by shared.

3-Figure 6H: since increase in R-loop formation and the resulting rescue of erythroid differentiation is a major point, the authors should provide additional data to confirm this point. Such experiments should include S9.6 immunofluorescence, DRIP-PCR of targets whose expression (up or down) is not restored, etc.. RNAseq and DRIPseq of HDACi treated samples could provide such confirmation. Also, while the murine cells may replicate some aspects of human disease and human cells, the fact remains that Sf3b1 mutant mice do not develop MDS and remain a poor model for SF3B1 mutant MDS. The authors should validate the effectiveness of HDACi in human SF3B1 mutant MDS.

Answer: We thank the reviewer for this important comment. At first, we considered the murine cell line as a training model for assessing the effect of HDACi vorinostat. It allowed to determine the non-cytotoxic concentration to be used on primary cells. However, we do believe that it may not recapitulate all the features of human disease.

As requested, we performed DRIP-seq and RNA expression analyses of human erythroblasts before and after treatment with HDACi and we hope that these experiments will improve the understanding of the effect of HDAC inhibitor on R-loop formation, intron retention and gene expression.

For these tasks, we have studied 39 additional samples derived from 15 cases of MDS with *SF3B1* mutation, 7 cases with *SRSF2* (5) or *U2AF1* (2) mutation, 9 MDS without splicing factor mutation and 8 healthy controls.

Erythroid cultures were derived from bone marrow CD34⁺ progenitors and HDAC inhibitor vorinostat treatment was performed between d9 and d11 (proerythroblastic stage) for 20h.

To verify the level of R-loops genome-wide, DRIP-seq experiments on human CD34⁺-deriving erythroblasts at ProE/basophilic stage were completed. We have included a total of 15 patients and healthy controls for DRIP-seq experiments. We used RNase H1 (RNH1) pretreatment to remove R-loops before immunoprecipitation with S9.6 antibody, for each sample. DRIP-seq experiments were performed in 10 cases \pm HDACi 0.5 μ M and for each RNase H1 treatment was done (n=40). The input (one per individual) was also sequenced.

Importantly, to ensure the robustness of the results, we run the bioinformatic analysis of all the 65 DRIP-seq at the same time.

The current dataset comprised

- 5 samples with *SF3B1* mutation: 3 cases with *SF3B1* K700E, one case with *SF3B1* H662C mutation and one case with *SF3B1* K666R mutation
- 4 healthy controls

- 6 MDS *SF3B1*^{WT} with distinct genotypes including 3 cases with a pattern of mutations usually characteristic of patients who could evolve to chronic myelomonocytic leukemia (CMML), denoted CMML-like in the table below.

	WHO	mutations
1	MDS-LB	DNMT3A P904Q (33%)
2	MDS-LB	TET2 splice 35%, TP53 p.Phe134Cys 4%
3	MDS-LB	No mutation (NGS panel 40 genes)
4	CMML-like	SRSF2 49%, TET2 5%
5	CMML-like	TET2 Lys1692Trpfs 94%, NRAS Gly12Asp 39%
6	CMML-like	SRSF2 P95H 38%, SETBP1 4%, TET2 p.Thr1399fs 31%, TET2 p.Arg1359Cys 28%, CUX1 5%, JAK2 2%

Our main findings of HDACi treatment are the following:

- HDACi rescues the R-loops of *SF3B1*-mutated samples.
- HDACi slows down the fork speed of *SF3B1*-mutated samples. However, in one case of MDS with *SF3B1*, *TET2* and a frameshift in the gene encoding histone acetyltransferase *EP300*, the effect on fork speed was not observed.
- HDACi does not impact on intron retention but it could rescue gene expression, based on the analysis of some genes.
- HDACi does not impair the expansion of erythroid precursors at least in vitro, but improves their final maturation.

Figure 20: DRIP-seq in 10 samples with or without pre-treatment with HDACi 0.5 μ M 20h. Flow cytometry for erythroid differentiation upon HDACi.

Minor Comments:

1. The highlight #1 should be reconsidered: it seems that reduction of retained introns coincides

with loss of R-loops. Figure 3D suggests that the reduction of retained introns is mainly explained by loss of R-loops but not vice versa. 95% of R-loops lost are not explained by reduction of retained introns.

Answer: We have modified Highlight 1 accordingly
Reduction of intron retention coincides with the loss of R-loops.

2. How do the authors reconcile this aspect?

Answer: In this study we provide a genome-wide picture of R-loops in human primary cells thanks to the use of DRIP-seq. This technique allowed us to show that R-loop loss occurs not only at gene body, but also at promoter/TSS and in lesser extend at 3'UTR and TTS. The loss of R-loops at promoter is massive and unlikely related to the reduction of intron retention.

We have added this comment to the discussion p.15: Increased intron excision occurring when *SF3B1* is mutated may suppress R-loops along the gene bodies. We report here that most of IRR overlap with lost R-loops. However, because numbers of R-loops were lost also in promoter-TSS, intergenic regions and TTS, **IRR is not the unique mechanism for R-loop loss in these cells.**

3. The authors state that in contrast to *SF3B1* mutations that cause replication stress by accelerated replication fork speed and exposure to ssDNA that in turn is ameliorated by HDAC inhibitors. They should better explain how this is distinct from the erythroid differentiation defect shown in *U2AF1* mutant MDS. Is the HDAC effect not “simply” due to rescued intron retention rather than specifically via R-loop formation? *U2AF1*-mut increases R-loops (Chen Mol Cell 2018) and impairs erythroid differentiation (Yip JCI 2017). This should be further discussed.

Answer: Thank you for this comment and for citing the excellent description of *U2AF1*-mutated erythropoiesis and granulopoiesis provided by the team of Dr J Boulwood. According to Yip et al (PMID: 28436936), *U2AF1*^{S34F} mutant impaired expansion of immature erythroid precursors with an increase of apoptosis. By contrast, *SF3B1* mutant impaired more specifically mature erythroid precursors resulting in the expansion of immature erythroblasts.

The analysis of splicing events may provide insights into this default. In fact, intron retention is crucial for the final maturation of erythroblasts to specify the transcriptome toward the production of hemoglobin. IR-transcripts are detained in the nucleus or eliminated. The loss of intron retention, which is abundant in *SF3B1*-mutated MDS, not in *U2AF1*-mutated MDS, could participate in the maintenance of immature and still proliferating erythroid cells.

R-loop loss may produce a tolerable DNA replication stress allowing the expansion of immature *SF3B1*-mutated erythroblasts. By contrast, increased R-loops in *U2AF1*^{MUT} erythroblasts is a source of replication fork collapse, DNA damage and cell death which may explain a more severe impairment of erythropoiesis in this context.

In the present study we showed that HDACi induced the final maturation of erythroblasts without a deleterious effect on cell growth. As requested by the reviewer, we checked whether this effect was related to a rescue of intron retention.

We investigated the intron retention of 5 genes *PPOX*, *COASY*, *PPM1A*, *S100A4* and *BCL2L1* in 4 *SF3B1*^{MUT}, 6 *SF3B1*^{WT} and 2 control samples. As shown in Fig. 21, we did not observe any significant modification of the expression of the intronless and IR-transcripts of these genes in *SF3B1*^{MUT} samples (see figure 6D). HDACi did not rescue the intron retention reductions.

Figure 21 (Fig. 6E of the ms): Impact of HDACi on intron retention. RT-PCR representative of 4 *SF3B1*^{MUT}, 6 *SF3B1*^{WT} and 2 control samples.

By contrast, HDACi restored the formation of R-loops at a level comparable to the control samples. In particular, increased expression of the GATA1 and STAT5 target gene *BCL2L1* was linked to the restoration of an R-loop near its promoter. This may participate in rescuing mature erythroblasts from apoptosis (Fig. 22).

Figure 22: Restoration of R-loop near *BCL2L1* promoter and increased expression of *BCL2L1* by RT-qPCR.

This comment is added in the discussion p.15: Importantly, we identified *BCL2L1*, a GATA1 and STAT5 target gene, which expression increased, without splicing changes, when R-loops formed near its promoter under HDACi treatment.

4. Figure 1B – green/red should be avoided to accommodate individuals with color blindness. The highlight in green of “terms of interest” seems arbitrary/serving the purpose of the paper; it could be better achieved by highlight text in bold.

Answer: We thank the reviewer for this recommendation. The figure 1B has been modified accordingly

5. In Supp Fig 1A the authors mention 189 MDS samples analyzed by RNA-seq. In the methods the total number of samples is 100, and in the paragraph describing figures 1 and 1S the authors describe 21+6 samples. Please reconcile and provide data sets or references in a table.

Answer: We have loaded all the datasets in GEO. We used three cohorts:

- the first one published in Sc Transl Med has been re-analyzed for this paper (21 MDS with *SF3B1* mutation and 6 MDS without *SF3B1* mutation). Results are provided in supplementary table 1.
- the second cohort (in GEO upon request to the corresponding author) is a large cohort of 189 lower risk MDS patients that enabled us to compare splicing events according to the type of splicing gene mutation (*SF3B1*, *SRSF2*, *U2AF1*).
- the third cohort contained erythroblastic samples from 17 MDS patients (11 *SF3B1*^{MUT}, 6 *SF*^{WT}). Erythroid cells were derived from bone marrow CD34+ progenitors and processed for RNA-seq. Results are presented in supplementary table 2.

The detailed description of the 3 cohorts are now included in the Mat and Methods part.

6. Add stats to Figure 2B.

Answer: A new version of Fig. 2B is now presented including the data from additional cultures performed during the time of manuscript revision and statistics.

7. The authors should generate a supp figure where they split the GO analysis for up- and down-regulated genes. For example, considering their result in Figure 4, one would expect to see enrichment in DNA replication for the upregulated genes.

Answer: As requested by the reviewer, we have separated the up and down regulated genes. As expected, the upregulated genes contributed to changes in pathways while the downregulated genes did not (Suppl Fig. 1A). We have added the barplot of GO terms related to upregulated genes in baso E and polyE as a Suppl figure 3A. No significant GO terms were obtained with the set of downregulated genes.

Figure 23 (Suppl Fig. 1A of the ms): Gene Ontology (GO) analysis of upregulated and down-regulated genes in *SF3B1*^{MUT} bone marrow mononuclear cells (BM MNC) compared to *SF3B1*^{WT} BM MNC.

Figure 24 (Suppl Fig. 3A of the ms): Gene Ontology (GO) analysis of upregulated genes in *SF3B1*^{MUT} erythroblasts compared to *SF3B1*^{WT} erythroblasts. Upper panel: basophilic erythroblasts (basoE). Bottom panel: polychromatophilic erythroblasts (polyE). No significant GO terms appeared in the downregulated genesets

To clarify the contribution of gene expression levels and pathways to the phenotype, GSEA was performed to decipher which DNA repair pathways were involved (Fig. 25).

Figure 25 (Fig. 2G of the ms): GSEA of up-regulated and down-regulated genes in the transcriptome of basophilic erythroblasts with *SF3B1* mutation showing their enrichment in DNA replication, base excision repair (BER), and G1/S phase checkpoint.

8. Genes named in Figure 2I are highly selected. There seem to be many more significantly changed genes that should be spelled out to provide more unbiased representation that can still highly DNA repair.

Answer: The new representations of volcano plots of proteins which were up and down in *SF3B1*-mutated proE/basoE or polyE highlight the most deregulated proteins in terms of fold-change and/or significance.

Figure 26 (Fig. 2K): Volcano plots represented differentially expressed proteins in the proteome of *SF3B1*^{MUT} proE/BasoE or polyE compared to *SF3B1*^{WT} cells.

9. For Figure 3C the authors could show distribution of R-loops relative to exons/introns

Answer: We thank the reviewer for this comment. We understand that the question is to compare R-loops overlapping exons to R-loops overlapping introns.

The method we used to identify R-loops across the genome is based on the recommendations of Dr F Chédin. It consists in delineating R-loops on restriction fragments generated during the DRIP-seq experiments. This allows to compare samples to samples.

As mentioned previously, R-loops covering gene bodies, known as type II R-loops are large, often overlapping intronic and exonic sequences. This is why we considered more reasonable to fuse intron and exon in gene body.

However, if we use the annotation by default, values for exons and introns can be produced separately but not confidently to our opinion. The repartition shows that R-loops at introns could be much more frequent than at exons.

10. In Figure 3F the SF3B1 WT samples with wide variability between samples include a U2AF1 mutant sample. This seems odd as U2AF1 mutations aberrantly induce R-loop formation. The U2AF1 mutant sample should be identified and if it significantly contributes to the differences between SF3B1 mut and wt it should be replaced by a SF WT sample.

Answer: The sample with U2AF1 mutation has also a heterozygous mutation in DDX41 gene, both U2AF1 mutation and DDX41 depletion have been reported to induce R-loops. As recommended by the reviewer, we have highlighted this case on the graph with a red dot. For this set of experiments, we did have additional samples to replace the U2AF1-mutated sample (Fig. 27).

Figure 27 (Fig. 3K of the ms): DRIP-qPCR analysis of 4 controls, 3 *SF3B1*^{WT} (2 *SF*^{WT} and 1 *U2AF1*^{MUT}) and 4 *SF3B1*^{MUT} erythroblastic samples. The enrichment signals normalized to the input show R-loops at specific loci of four genes including 2 genes with RI (*ABCC5*, *TCIRG1*) and 2 genes without RI (*IREB2*, *TMX2*) in *SF3B1*^{WT} erythroblasts, normalized to a no R-loop region downstream of *EGR1*. Two positive control genes (*RPL13A*, *TFPT*) are shown. Box plots represent means and standard errors of three or four samples and bars correspond to min and max values. The red dot indicates *U2AF1*^{MUT} sample also harboring a heterozygous *DDX41* mutation.

11. Figure 3I: how do the authors explain the decreased expression in *SF3B1* mutant cells in those genes that have R-loops at the TSS in *SF3B1* WT cells? The last sentence of the paragraph should be explained and justified mechanistically (Together, this suggests that transcription initiation is less active in *SF3B1*^{MUT} cells.).

Answer: We thank the reviewer for this comment.

As an example, we are now provided the quantification of *BCL2L1* gene before and after restoration of R-loop near its promoter by HDACi (Fig. 22, right panel).

R-loops assemble dynamically at transcription initiation and termination sites, where they contribute to the regulation of gene expression and transcription termination (PMID: 30735654; PMID: 23868195). Alternatively, transcription regulates the formation of R-loops.

SRSF2 has been implicated in the regulation of R-loop formation at promoter through a non-splicing function implicating the release of RNAPII from proximal pausing (PMID: 23663783). Since the *SF3B1* protein is detected in the vicinity of R-loops at promoters together with SRSF2 and *U2AF1* (Mosler et al. 2021; PMID: 34916496), the *SF3B1* mutation could notably modify the kinetics of transcription elongation, as generally reported for splicing factors (BiorXiv <https://doi.org/10.1101/2023.02.25.530019>). These elements have added to the discussion p.15.

12. Figure 4: sample numbers in main text, figure legend and figure do not correspond?

Answer: We thank the reviewer for carefully reading of our manuscript. We apologize for this mistake.

In the new figure 4I-K of the revised version, additional samples have been tested

- 4 controls
- 9 SF3B1 mut
- 7 SFmut other than SF3B1
- 3 SFwt

In Fig. 4H, p-RPA 32 s33 foci were investigated in additional samples

- 2 controls
- 6 SF3B1 mut
- 7 SFmut other than SF3B1
- 5 SF wt

This is clearly stated in the figure legend.

To avoid any confusion, we have removed the first sentence of the figure legend that recapitulated the total number of samples used for the experiments reported in figure 4.

13. Figure 4 I,J,K: the bar for the U2AF1 mutant sample should be removed. It is a single point.

Answer: We are providing a new figure in which *SRSF2*^{MUT} and *U2AF1*^{MUT} samples were put together as the phenotype we reported was similar (Fig. 28).

Fig. 28 (Fig. 4H-K of the ms): Histograms showing the results of immunofluorescence experiments for detection of p-RPA32 s33, p-RPA32 s4/8, γ H3AX, 53BP1 in indicated samples.

14. In Supple Fig. 6 was a WT donor template not used for WT cells? Show sequencing validation of heterozygous K700E mutations.

Answer: We did not introduce a WT donor template. Only one transfection with mutant donor template was performed.

To get WT clones, we have selected the clones that integrated the hygromycin selection cassette, but not the mutation during repair.

We provide the reviewer with the PCR fragment sequencing of the hygromycin-resistant clones.

Fig. 29: Chromatograms of PCR fragment sequencing showing the presence or absence of *Sf3b1*^{K700E} mutation in G1E-ER4 clones.

15. Figure S8 suggests regulation of targets in the same direction (up/down) in *Sf3b1* mut and by the HDAC inhibitor vorinostat. If HDACi restore R-loops one would expect opposite regulation of the described key targets.

Answer: The Ingenuity Pathway Analysis revealed vorinostat as an upstream regulator of the deregulated proteins shown in the figure based on known interactions with upregulated CASP2, and downregulated CRKL, SAMHD1, LSS, HTT, PLIN4, XIAP in the SF3B1 mutated condition.

However, there is no predictive effect of vorinostat on upregulated CDK9, BID, SIN3A and downregulated CREBBP.

This figure based on the IPA algorithm is not informative for SIN3A, the histone deacetylase interacting with the THO complex described as an R-loop preventing factor by the team of Dr A Aguilera.

Because this figure was drawn from the upstream regulator analysis in IPA of the murine proteome and we remove the proteome analysis of murine cell lines, we propose to remove it from the manuscript.

Reviewers' Comments:

Reviewer #1:

Remarks to the Author:

The authors adequately addressed the comments raised in my previous review.

Reviewer #2:

Remarks to the Author:

The authors provided a thorough and thoughtful reanalysis and discussion where requested and I think they have improved the manuscript. It will be interesting to see how this work is received. I have no further concerns.

Reviewer #3:

Remarks to the Author:

The authors have conducted extensive revision of the manuscript and addressed all critiques by this reviewer.